# Fixing the Loose Brake: Exponential-Tailed Stopping Time in Best Arm Identification

Kapilan Balagopalan [* 1]   Tuan Ngo Nguyen [* 1]   Yao Zhao [* 1]   Kwang-Sung Jun [1]

## Abstract

The best arm identification problem requires identifying the best alternative (i.e., arm) in active experimentation using the smallest number of experiments (i.e., arm pulls), which is crucial for cost-efficient and timely decision-making processes. We consider the fixed confidence setting where an algorithm repeatedly selects arms until it decides to stop and then returns the estimated best arm with a correctness guarantee. Since this stopping time is random, we desire its distribution to have light tails. Unfortunately, many existing studies focus on guarantees that hide the issue of allowing heavy tails or even not stopping at all. Indeed, we show that the never-stopping event can indeed happen for standard algorithms. Motivated by this, we make two theoretical contributions. First, we show that there exists an algorithm that attains a desirable exponential-tailed stopping time guarantee that is strictly better than the polynomial tail bound of Kalyanakrishnan et al. (2012) and the exponential guarantee obtained by uniform sampling. Our guarantee is more fundamental than existing ones in the sense that our guarantee implies that it achieves existing optimal guarantees up to logarithmic factors. Second, we show that there exists a meta algorithm that takes in any fixed confidence algorithm with a high probability stopping guarantee and turns it into one that enjoys an exponential-tailed stopping time with a matching instance-dependent complexity up to logarithmic factors. Our results imply that there might be much more to be desired for contemporary fixed confidence algorithms.

## 1. Introduction

The multi-armed bandit model serves a foundational framework for studying sequential decision-making under uncertainty. In this framework, a learner interacts with an environment by sequentially selecting among multiple alternatives (arms) and observing stochastic rewards, allowing for the rigorous analysis of fundamental trade-offs inherent in sequential decision-making. A core problem within this theoretical landscape is best arm identification, which focuses on identifying the arm associated with the highest mean reward, either with a pre-specified confidence level or within a pre-specified sampling budget, known as the fixed confidence setting and fixed budget setting, respectively.

While the fixed confidence setting has conceptual connections to applications such as A/B/../K testing, the focus of this work lies in the more general and theoretical side, adaptive fixed confidence best arm identification setting, where arm pulls are assigned adaptively based on the outcomes of previous pulls. The advantage of adaptive assignment is that it often results in significantly lower sample complexity compared to non-adaptive assignment.

Therefore, many works have proposed algorithms and proved bounds on how many samples they use until stopping (i.e., sample complexity) either with high probability or in expectation (Even-Dar et al., 2006; Karnin et al., 2013; Jamieson et al., 2014). However, most existing sample complexity guarantees do not sufficiently describe the behavior of the stopping time. For instance, an algorithm might be guaranteed to stop before $T^*$ samples with probability at least $1 - \delta$, but with probability up to $\delta$, the algorithm may never stop or stop only after a very long time, much larger than $T^*$. Algorithms with expected sample complexity guarantees will stop, but the tail of the distribution of the stopping time can be very thick. Thus, the realized stopping time can be significantly larger than the expected one, which is undesirable. These issues have been under-explored in the literature.

To address this gap, we push the limits of attainable guarantees for the tail decaying rate of stopping time distribution of fixed confidence best arm identification algorithms. To the best of our knowledge, aside from LUCB by (Kalyanakrishnan et al., 2012) with a polynomial-tailed stopping time bound, there has been no significant study on the tail of the

*Equal contribution [1]University of Arizona. Correspondence to: Kwang-Sung Jun <kjun@cs.arizona.edu>.

*Proceedings of the 42ⁿᵈ International Conference on Machine Learning*, Vancouver, Canada. PMLR 267, 2025. Copyright 2025 by the author(s).

stopping time distribution.[1] In particular, there are no results in the literature reporting an exponential tail bound for the stopping distribution except for the uniform sampling. However, uniform sampling is widely regarded as naive due to its inability to adaptively assign arm pulls.

In this paper, we make several key contributions towards strengthening guarantees for the stopping time distribution, which we summarize as follows:

- **Theoretical Evidence of a Limitation in Existing High-Probability Guarantees:** We demonstrate that both Successive Elimination (Even-Dar et al., 2006) and KL-LUCB (Tanczos et al., 2017), despite enjoying high-probability guarantees, they will not stop at all with a constant probability. To ourknowledge, this is the first such evidence in the literature for fixed confidence algorithms.
- **First Theoretical Work Establishing the Possibility of an Exponential-Tailed Stopping Time:** We present a fixed-confidence variant of Double Sequential Halving (FC-DSH), an algorithm that combines the Sequential Halving approach with the doubling trick and a carefully designed stopping rule. Our analysis shows that FC-DSH achieves an exponential-tailed stopping time with a matching instance-dependent complexity up to logarithmic factors. To our knowledge, this is the first such guarantee in the literature.
- **Introducing BrakeBooster: A Meta-Algorithm for Achieving Exponential-Tailed Stopping Time:** Motivated by our success with FC-DSH, we take a step further and propose a novel meta-algorithm approach. This approach takes in any fixed confidence algorithm that meets mild conditions and and turns it into an algorithm with an exponential tail guarantee for the stopping time.

Table 1 presents a comparison of popular FC-BAI algorithms, highlighting our theoretical contributions to the FC-DSH and BrakeBooster algorithms in achieving exponential-tail stopping time. We want to clarify that our objective is not to improve the computational complexity or practical performance of existing FC-BAI algorithms. Instead, our work demonstrates that an exponentially decaying tail with a matching instance-dependent complexity (up to logarithmic factors) for the stopping time distribution is achievable. Also, we focus on FC-DSH due to its simplicity and widespread adoption in practice (e.g., in Hyperband (Li et al., 2018)). The computational efficiency of FC-DSH's is not worse (orderwise) than other algorithms.

---

[1] While AT-LUCB (Jun & Nowak, 2016) showed a tail bound that decays exponentially, the bound is exponential in $\sqrt{t}$ rather than $t$, which does not fit Definition 1. Furthermore, the correctness is questionable, specifically, the paper's Lemma 7 and 8 is likely to be false.

## 2. Problem Definition and Preliminaries

We consider the standard $K$-armed bandit setting, where a learner sequentially selects one of $K$ arms where each arm $i \in [K]$ is associated with a reward distribution $\nu_i$ that is 1-sub-Gaussian (known) with mean $\mu_i$ (unknown). We assume that there exists a unique best arm, which is standard (Audibert et al., 2010). Without loss of generality, we assume that the arms are ordered in decreasing order of their mean rewards, i.e., $\mu_1 > \mu_2 \geq \cdots \geq \mu_K$. At each time step $t$, the learner selects an arm $A_t \in [K] := \{1, 2, \ldots, K\}$, then observes a reward $r_t \sim \nu_{A_t}$. We consider the fixed confidence setting, where the learner aims to identify the best arm with a pre-specified confidence level $\delta \in (0, 1)$ that is also called failure rate. The goal is to design an algorithm $\mathcal{A}$ that includes a *sampling rule* choosing $A_t$, a *stopping rule* that determines when to stop, and a *recommendation rule* that outputs the estimated best arm $J(\mathcal{A})$ when stopping. We denote by $\tau(\mathcal{A})$ the (random) stopping time of the algorithm $\mathcal{A}$, which is the arm pulls that $\mathcal{A}$ makes before the algorithm stops. We often omit the dependence on the algorithm $\mathcal{A}$ and use $\tau$ and $J$ when the algorithm being used is clear from context.

An algorithm for the fixed confidence setting is required to satisfy the following correctness result.

**Definition 2.1** ($\delta$-correct). A fixed confidence algorithm is said to be $\delta$-correct if it takes $\delta$ as input and satisfies

$$\mathbb{P}(\tau < \infty, J \neq 1) \leq \delta .$$

We call such an algorithm *$\delta$-correct*. Note that the condition $\tau(\mathcal{A}) < \infty$ above is necessary since $J_\tau(\mathcal{A})$ is undefined otherwise.

Furthermore, we desire the algorithm to stop as early as possible in addition to being $\delta$-correct. There are two criteria that have been popular in the literature: asymptotic expected sample complexity[2] and high probability sample complexity.

The asymptotic expected sample complexity (Chernoff, 1959; Garivier & Kaufmann, 2016; Shang et al., 2020; Qin et al., 2017) characterizes the asymptotic behavior of the stopping time as $\delta$ goes to 0 as follows:

**Definition 2.2** (Asymptotic expected sample complexity). A fixed confidence algorithm is said to have an asymptotic expected (AE) sample complexity of $T_\delta^*$ if it satisfies

$$\liminf_{\delta \to 0} \frac{\mathbb{E}[\tau]}{\ln(1/\delta)} \leq T_\delta^* ,$$

where $\tau$ depends on $\delta$.

The optimal guarantee has been well-understood for exponential family reward models (Garivier & Kaufmann, 2016). For example, Track-and-Stop (Garivier & Kaufmann, 2016)

---

[2] Recently, Jourdan & Degenne (2023) analyzed the non-asymptotic expected sample complexity.

*Table 1.* Comparison of guarantees.

| | Exponential-tailed stopping time | High probability sample complexity | Asymptotic expected sample complexity | Meta algorithm |
|---|---|---|---|---|
| Successive Elimination (Even-Dar et al., 2006) | ✗ | ✓ | ✗ | ✗ |
| LUCB (Kalyanakrishnan et al., 2012) | Unknown | ✓ | ✓ | ✗ |
| Track-and-Stop (Garivier & Kaufmann, 2016) | Unknown | Unknown | ✓ | ✗ |
| Top Two algorithms (Jourdan et al., 2022) | Unknown | Unknown | ✓ | ✗ |
| TTUCB (Jourdan & Degenne, 2023) | Unknown | Unknown | ✓ | ✗ |
| FC-DSH (our work) | ✓ | ✓ | ✓ | ✗ |
| BrakeBooster (our work) | ✓ | ✓ | ✓ | ✓ |

achieves the optimal asymptotic expected sample complexity.

However, the AE sample complexity has two limitations. First, the AE sample complexity does not tell us anything about the tail of $\tau$. In fact, $\tau$ can still be heavy-tailed as empirically observed by Jourdan et al. (2022, Figure 4, EB-TC) despite having a near-optimal AE sample complexity. Second, the AE sample complexity hides potentially bad behaviors of the algorithm in the non-asymptotic regime or with moderately small $\delta$. That is, the AE sample complexity hinges on the behavior of the algorithm when $\delta$ close enough to 0, in which case the algorithm would be running for a very long time. Indeed, one can observe that if $\mathbb{E}[\tau] \approx A \ln(1/\delta) + B$ for some $A$ and $B$ that are not dependent on $\delta$, the value of $\liminf_{\delta \to 0} \frac{\mathbb{E}[\tau]}{\ln(1/\delta)}$ will be independent of $B$ even if $B$ is very large.

On the other hand, high probability sample complexity guarantee (Even-Dar et al., 2006; Karnin et al., 2013; Jamieson et al., 2014; Jun et al., 2016; Tanczos et al., 2017), defined below, does not rely on the asymptotic behavior of the algorithm.

**Definition 2.3** (High probability sample complexity). A fixed confidence algorithm is said to have a sample complexity of $T_\delta^*$ if it takes $\delta \in (0, 1)$ as input and satisfies

$$\mathbb{P}(\tau \geq T_\delta^*) \leq \delta .$$

For example, Successive Elimination (Even-Dar et al., 2006) achieves a high probability sample complexity of $\tilde{\mathcal{O}}(H_1 \ln(1/\delta))$ where $H_1 := \sum_{i=2}^{K} \Delta_i^{-2}$ characterizes the instance-dependent complexity of the problem and $\tilde{\mathcal{O}}$ omits logarithmic factors except for $\ln(1/\delta)$. Despite being non-asymptotic, we find the high probability sample complexity weak and rather unnatural for the following reasons:

- First, we find it unnatural that $\delta$, the failure rate regarding the *correctness* of the output $J$, is also the

target failure rate with which we bound the *sample complexity*. In practice, one may desire to be loose on the correctness (large $\delta$) yet want to ensure that the stopping time is small with *very high* confidence (small $\delta$). We speculate that the high probability sample complexity is a mere byproduct of easy analysis.

- Second, more importantly, the guarantee above does not tell us about the shape of the tail of the stopping time. In particular, the high probability sample complexity guarantee does not exclude the possibility of never stopping even if the problem at hand is easy, which is a serious issue as it would imply that the practitioner will have to wait forever or forcefully stop the active experimentation procedure (See Figure 1). It also follows that the expected stopping time $\mathbb{E}[\tau]$ does not exist.

To further demonstrate the issue of having an extremely bad tail for the stopping time distribution, we provide a lower bound for the Successive Elimination algorithm (Even-Dar et al., 2006). Hereafter, all the proofs are deferred to the appendix unless stated otherwise.

**Theorem 2.4.** *For Successive Elimination, there exists an instance with a unique best arm where the algorithm never stops with a non-negligible probability of* $\Omega\left(\delta^{118}\right)$.

Although the probability bound established in Theorem 2.4 appears to be very small, it is just the looseness of the analysis, and in fact, the fraction that does not stop is quite nontrivial over a wide range of values of $\delta$. See Figure 2.

We show that the same is true for KL-LUCB (Tanczos et al., 2017).

**Theorem 2.5.** *For KL-LUCB, there exists an instance with a unique best arm where the algorithm never stops with a constant probability.*

*Remark* 2.6. One way to stop the algorithm from infinitely running is to allow $\varepsilon$-slack in the stopping condition as done

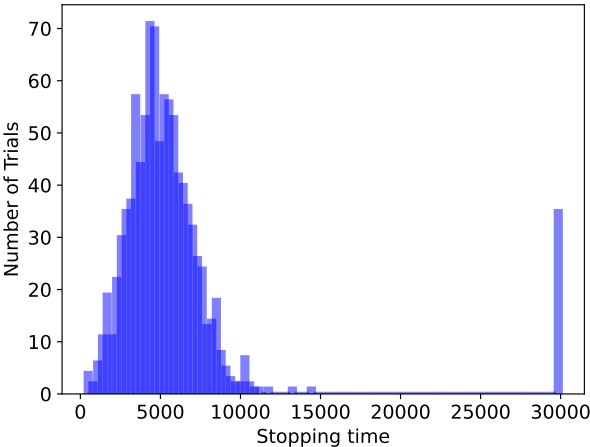

*Figure 1.* Historgram of stopping times of Successive Elimination (Even-Dar et al., 2006) out of 1000 independent trials on three arms with mean rewards of $\{1.0, 0.9, 0.9\}$ and Gaussian noise $\mathcal{N}(0, 1)$. We forcefully terminated the runs that do not stop until 30,000 time steps ($\delta = 0.01$). We have observed that all these runs have already eliminated the best arm, and thus we expect that many of them will never stop.

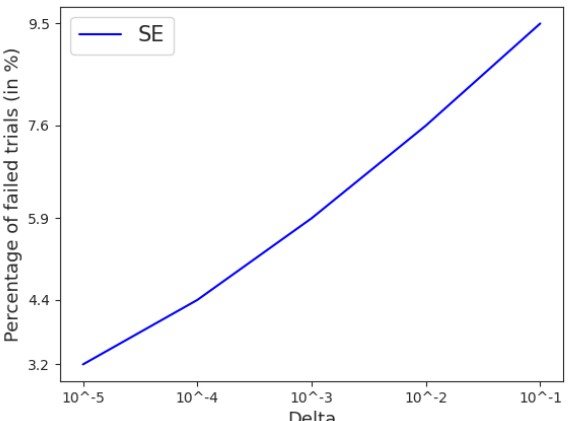

*Figure 2.* Percentages of trials that fails to terminate as a function of varying $\delta$ values over the range $[10^{-5}, \ldots, 10^{-1}]$. Results from 100K trials of Successive Elimination algorithm

in, e.g., Kalyanakrishnan et al. (2012). However, one can also extend the arguments in Theorem 2.4 and 2.5 to show that the stopping time can be as large as $\Theta(K/\varepsilon^2)$ even if the guaranteed high probability sample complexity is significantly smaller than $\Theta(K/\varepsilon^2)$, and this gap can be made arbitrarily large. Furthermore, we conducted experiments (see Appendix D), to compare the stopping time distribution before and after introducing $\varepsilon$-slack stopping condition for Successive Elimination (Even-Dar et al., 2006). The $\varepsilon$-slack variant achieves a weaker exponential-tail (similar to naive

uniform sampling). In this paper, we focus on the best arm identification to keep the discussion concise.

Meanwhile, the seminal work by Kalyanakrishnan et al. (2012) has proposed an algorithm called LUCB1 that satisfies the following polynomial tail guarantee on the stopping time, which is adapted for the best arm identification problem rather than the $\varepsilon$-optimal arm identification.

**Theorem 2.7** (Adapted from Kalyanakrishnan et al. (2012)). *Let* $T^* = \left\lceil 146 H_1 \ln\left(\frac{H_1}{\delta}\right) \right\rceil$. *For every* $T \geq T^*$, *the probability that LUCB1 has not terminated after $T$ samples is at most* $\frac{4\delta}{T^2}$.

This makes us wonder if it is possible to achieve an exponentially-decaying tail bound for the stopping time $\tau$, which we believe is important in practice as well. We formalize the desired property in the following definition where $x = \text{polylog}(T)$ means $x \leq a \log^b(T) + c$ for some absolute and positive constants $a$, $b$, and $c$.

**Definition 2.8** (($T_\delta, \kappa$)-exponential stopping tail). A fixed confidence algorithm is said to have a $(T_\delta, \kappa)$-exponential stopping tail if there exists a time step $T_\delta$ and a problem-dependent constant $\kappa > 0$ (but not dependent on $T$) such that for all $T \geq T_\delta$,

$$\mathbb{P}\left(\tau \geq T\right) \leq \exp\left(-\frac{T}{\kappa \cdot \text{polylog}(T)}\right). \qquad (1)$$

This property requires a tail bound for every large enough $T$, which reveals detailed information about the distribution function of $\tau$. Perhaps not surprising, this requirement is strictly stronger than the high probability sample complexity above and implies a few desirable properties regarding the stopping time $\tau$ as we summarize below.

**Proposition 2.9.** *If a fixed confidence algorithm $\mathcal{A}$ has $(T_\delta, \kappa)$-exponential stopping tail, then the following hold true:*

*(i)* $\mathbb{P}\left(\tau \geq T_\delta + \kappa \ln(1/\delta) \cdot \text{polylog}(\kappa \ln(1/\delta))\right) \leq \delta.$

*(ii)* $\mathbb{E}[\tau] \leq T_\delta + \kappa \cdot \text{polylog}(\kappa).$

*(iii)* $\mathbb{P}(\tau < \infty) = 1.$

*Proof.* For $(i)$, find $T$ that would make the RHS of (1) below $\delta$. For $(ii)$, use the identity $\mathbb{E}[X] = \sum_{x=0}^{\infty} \mathbb{P}(X > x)$. For $(iii)$, use the Borel–Cantelli lemma. $\square$

While the guarantees above can be individually satisfied by a specific criterion such as high probability or expected sample complexity, Proposition 2.9 shows that enjoying an exponential tail guarantee implies all three desirable properties simultaneously.

One can show that the naive uniform sampling algorithm that chooses arm $A_t = 1 + ((t - 1) \mod K)$ at time $t$ with a suitable stopping condition results in $(T_\delta, \kappa)$-exponential stopping tail with $T_\delta = \tilde{\Theta}(K\Delta_2^{-2} \ln(1/\delta))$

---

**Algorithm 1** FC-DSH

---

**Input**: A set of K arms, $\delta$
$T_1 = \lceil K \log_2 K \rceil$
$T_m = T_1 2^{m-1}, \forall m \geq 2$    // the sampling budget assigned to phase $m$
$L = \lceil \log_2(K) \rceil$    // the last stage in each phase
**for** $m = 1, 2, \ldots$ **do**
  // phase $m$
  Reset $\mathcal{A}_1 = [K]$
  **for** $\ell = 1, \ldots, L$ **do**
    // stage $\ell$
    Sample each arm $i \in \mathcal{A}_\ell$ for $N^{(m,\ell)}$ times where

$$N^{(m,\ell)} = \left\lfloor \frac{T_m}{K 2^{-\ell+1} \lceil \log_2(K) \rceil} \right\rfloor.$$

    Let $\mathcal{A}_{\ell+1}$ be the set of $\lceil \mathcal{A}_\ell/2 \rceil$ arms in $\mathcal{A}_\ell$ with the largest empirical means computed using samples from this stage only.
  **end for**
  Select $J_m$ as the only arm in $\mathcal{A}_L$.
  // stopping rule
  **if** $L_{J_m}^{(m)} \geq \max_{i \neq J_m} U_i^{(m)}$ **then**
    Stop and output $J_m$.
  **end if**
**end for**

---

and $\kappa = \Theta(K\Delta_2^{-1})$. However, by Proposition 2.9, this guarantee is converted to a high probability sample complexity of $\tilde{\mathcal{O}}(K\Delta_2^{-2} \ln(1/\delta))$, which is much worse than $\tilde{\mathcal{O}}(H_1 \ln(1/\delta))$ achieved by Successive Elimination.

The complexity of a best-arm identification problem is often characterized by instance-dependent quantities, such as $H_1 := \sum_{i=1}^{K} \Delta_i^{-2}$ and $H_2 = \max_{i=2}^{K} i\Delta_i^{-2}$ (Audibert et al., 2010). These two quantities are equivalent up to a logarithmic factor, as captured by the inequality:

$$H_2 \leq H_1 \leq \log(2K)H_2. \tag{2}$$

We thereby ask if it is possible to achieve $(\tilde{\Theta}(H_1 \ln(1/\delta)), \tilde{\mathcal{O}}(H_1))$-exponential stopping tail, or perhaps a looser form $(\tilde{\Theta}(H_2 \ln(1/\delta)), \tilde{\mathcal{O}}(H_2))$. We answer this question in the affirmative by proposing two algorithms in the following two sections.

## 3. Fixed-Confidence Doubling Sequential Halving (FC-DSH)

In this section, we introduce **FC-DSH**, a fixed-confidence variant of the Doubling Sequential Halving (DSH) algorithm, motivated by its fixed-budget counterpart (FB-DSH) proposed by Zhao et al. (2023). We show that FC-DSH satisfies the correctness guarantee and achieves an $(\tilde{\Theta}(H_2 \log(1/\delta)), \tilde{\mathcal{O}}(H_2))$-exponential stopping tail.

The DSH algorithm combines the Sequential Halving (SH) procedure (Karnin et al., 2013) with the doubling trick. It proceeds in iterative phases, with each phase running an independent instance of SH. The budget allocation follows a doubling schedule, starting with budget $T_1 = \lceil K \log_2(K) \rceil$ for phase 1 and doubling for subsequent phases, i.e., $T_m = 2^{m-1}T_1$, for phase $m \geq 2$.

FC-DSH fundamentally differs from its fixed-budget counterpart (FB-DSH) in how termination is handled. FB-DSH is provided with a pre-specified budget and terminates once this budget is exhausted. Conversely, FC-DSH operates with a given confidence level $\delta \in (0, 1)$ and terminates once a carefully designed stopping rule is satisfied. Our stopping rule is designed to ensure both $\delta$-correctness and the desired $(T_\delta, \kappa)$-exponential stopping tail guarantees. In what follows, we introduce additional notation and formally define our stopping rule.

During the phase $m$, for any arm $i \in [K]$, let $\ell_i$ denote the stage at which arm $i$ is eliminated. At this point, arm $i$ has been sampled $N^{(m,\ell_i)}$ times and has received rewards $\{r_{i,j}^{(m)}\}_{j=1}^{N^{(m,\ell_i)}}$. Based on these rewards, we define the empirical mean of arm $i$ as:

$$\hat{\mu}_i^{(m)} := \frac{1}{N^{(m,\ell_i)}} \sum_{j=1}^{N^{(m,\ell_i)}} r_{i,j}^{(m)}$$

and its confidence width as:

$$b_i^{(m)} := \sqrt{\frac{2}{N^{(m,\ell_i)}} \log\left(\frac{6K \lceil \log_2(K) \rceil m^2}{\delta}\right)}.$$

Using these quantities, we define the upper and lower confidence bounds for arm $i$ as:

$$U_i^{(m)} := \hat{\mu}_i^{(m)} + b_i^{(m)} \text{ and } L_i^{(m)} := \hat{\mu}_i^{(m)} - b_i^{(m)}.$$

At the end of phase $m$, the corresponding SH instance selects an arm, denoted by $J_m$. We formally define the stopping rule as:

$$L_{J_m}^{(m)} \geq \max_{i \neq J_m} U_i^{(m)}. \tag{3}$$

The rule ensures that, with high probability, the selected arm $J_m$ is statistically significantly better than all other arms. The full algorithm is presented in Algorithm 1.

The following theorems show that DSH satisfies the correctness guarantee and enjoys an exponential stopping tail.

**Theorem 3.1** (Correctness). *FC-DSH runs with confidence level $\delta$ is $\delta$-correct.*

**Theorem 3.2** (Exponential stopping tail). *FC-DSH enjoys $(\tilde{\Theta}(H_2 \log(1/\delta)), \tilde{\mathcal{O}}(H_2))$-exponential stopping tail.*

Theorem 3.2 establishes the existence of a fixed-confidence best-arm identification (FC-BAI) algorithm with an exponential stopping tail, characterized by $(\tilde{\Theta}(H_2 \log(1/\delta)), \tilde{\mathcal{O}}(H_2))$. While this bound is not tight,

it marks the first step toward demonstrating that such an exponential stopping tail is indeed achievable.

The formal proofs are deferred to Appendix B.1 and B.2 respectively. Here, we highlight the key ideas underlying the proof of Theorem 3.2. The main objective is to bound the probability that, for all phase $m$ such that $T_m \geq T_\delta$, FC-DSH fails to satisfy the stopping rule given by Equation (3). This event can be decomposed based on whether the selected arm $J_m$ is optimal:

$$\mathbb{P}\left(L_{J_m}^{(m)} < \max_{i \neq J_m} U_i^{(m)}\right)$$

$$= \mathbb{P}\left(L_{J_m}^{(m)} < \max_{i \neq J_m} U_i^{(m)}, J_m \neq 1\right)$$

$$+ \mathbb{P}\left(L_{J_m}^{(m)} < \max_{i \neq J_m} U_i^{(m)}, J_m = 1\right)$$

The first term implies the event where FC-DSH fails to identify the optimal arm; bounding this event, $\mathbb{P}(J_m \neq 1)$, is well-studied in BAI literature.

The second term, where the stopping rule is not met despite the optimal arm is being selected, is more challenging and requires a finer-grained analysis. To handle this case, we examine how long suboptimal arms survive in the elimination process during phase $m$.

Recall that, for any arm $i \in [K]$, $\ell_i$ denotes the actual stage at which arm $i$ is eliminated. We define the expected elimination stage $\ell_i^*$ to be the largest stage index satisfying $\Delta_i \leq \frac{1}{2}\Delta_{\frac{K}{4}} \cdot 2^{-\ell_i+1}$. If no such $\ell_i^*$ exists, it implies that $\Delta_i > \Delta_{\frac{K}{4}}$, indicating that arm $i$ has a sufficiently large suboptimal gap and should be eliminated early. For arms with a valid $\ell_i^*$, we further partition the analysis based on whether the actual elimination stage $\ell_i$ is greater than or less than $\ell_i^*$. These cases highlight the core technical contributions of our proof beyond standard BAI analysis. We provide detailed bounds for each case in Lemmas B.2, B.3, and B.4, respectively.

## 4. BrakeBooster: A meta algorithm approach

In this section, we propose a novel algorithm called Brake-Booster. This is a meta algorithm in the sense that it takes any fixed confidence best arm identification algorithm (denoted by $\mathcal{A}$) equipped with the standard guarantees on the correctness and stopping time as an input and converts it into one that enjoys an exponential stopping tail.

We denote by $\mathcal{A}(\delta_0)$ as algorithm $\mathcal{A}$ run with a target failure rate of $\delta_0$. We assume that $\mathcal{A}(\delta_0)$ is $\delta_0$-correct (Definition 2.1) and has a sample complexity guarantee of $T_{\delta_0}^*(\mathcal{A})$ (Definition 2.3). Note that BrakeBooster will not require $T_{\delta_0}^*$ as input, but only the existence.

Given such an algorithm $\mathcal{A}$, we are ready to describe Brake-Booster whose full pseudocode can be found in Algorithm 2.

---

**Algorithm 2** BrakeBooster

**Input:** base trial count $L_1$, base budget $T_1$, algorithm $\mathcal{A}$, base failure rate $\delta_0$

**for** $r = 1, 2, \ldots$ **do**
  **for** $c = 1, 2, \ldots, r$ **do**
    $L_{r,c} := r \cdot 2^{r-c}L_1$, $T_{r,c} := 2^{c-1}T_1$
    $J_{r,c} = $ BudgetedIdentification$(\mathcal{A}, L_{r,c}, T_{r,c}, \delta_0)$
    **if** $J_{r,c} \neq 0$ **then**
      **return** $J_{r,c}$
    **end if**
  **end for**
**end for**

---

The key idea of BrakeBooster is to repeatedly invoke our key subroutine BudgetedIdentification (Algorithm 3) with increasing trial count $L_{r,c}$ and budget $T_{r,c}$ until a stopping criterion is met where $(r, c)$ is an index of each invocation (called *stage*) with $r \in \mathbb{N}_+$ and $c \in [r]$. We defer the explanation on how we schedule $L_{r,c}$ and $T_{r,c}$ to the next paragraph. In stage $(r, c)$, we are given the number $L = L_{r,c}$ of trials, the sampling budget $T = T_{r,c}$, and the base failure rate $\delta_0$, and run $\mathcal{A}(\delta_0)$ repeatedly $L$ times with a sampling budget of $T$. Since $\mathcal{A}$ itself may not stop before $T$ time steps, we ensure the budget constraint by forcing $\mathcal{A}$ to stop (*forced-termination*) when it does not stop by itself (*self-termination*) after exhausting $T$ samples. For each trial $\ell \in [L]$, we collect the returned arm index $\hat{J}_\ell$, which is set to 0 if $\mathcal{A}$ was forced-terminated. If at least half of the trials were forced-terminated, then we have failed – we return 0 from BudgetedIdentification. Otherwise, we declare success and return the majority vote over $\{\hat{J}_\ell : \ell \in [L], \hat{J}_\ell \neq 0\}$ (i.e., majority votes over nonzero votes). In the latter case, BrakeBooster stops and outputs the majority vote as the final output $J_\tau(\mathcal{M})$. In the former case, we continue to the next stage with trial count $L_{r',c'}$ and $T_{r',c'}$ where $(r', c') = (r, c + 1)$ if $c \leq r - 1$ and $(r', c') = (r + 1, 1)$ if $c = r$.

The key in our algorithm is the particular scheduling of $L_{r,c}$ and $T_{r,c}$, which is inspired by Li et al. (2018) and can be viewed as a two-dimensional doubling trick. We visualize the schedule in Figure 4. Each dot in the figure represents the stage $(r, c)$, and the label below each dot represents the number of trials ($L_{r,c}$) and the budget assigned to each trial ($T_{r,c}$). For a fixed row $r$, each stage spends the same sampling budget of $L_{r,c}T_{r,c} = r2^{r-1}L_1T_1$. However, the assignment of $L_{r,c}$ halves with increasing $c$ and $T_{r,c}$ doubles with increasing $c$. For a fixed column $c$, as the row index increases, the total budget for each stage $L_{r,c}T_{r,c}$ doubles – in fact slightly more than double due to a technical reason that will become clear in the proof of Theorem 4.1. Both the number of trials and budget increase row-wise, meanwhile different combinations of trial size and budget are employed column-wise, keeping the effective total samples used constant throughout the same row.

---

**Algorithm 3** BudgetedIdentification

---

**Input:** algorithm $\mathcal{A}$, the number of trials $L$, sampling budget per trial $T$, base failure rate $\delta_0$

**for** $\ell = 1, 2..., L$ **do**

    Run algorithm $\mathcal{A}$ until it self-terminates or exhausts the sampling budget $T$.

    **if** $\mathcal{A}$ has self-terminated **then**

        $\hat{J}_\ell = J(\mathcal{A})$

    **else**

        $\hat{J}_\ell = 0$

    **end if**

**end for**

Count the votes: $\forall i \in \{0, 1, \dots, K\}, v_i = \sum_{\ell=1}^{L} \mathbb{1}\{\hat{J}_\ell = i\}$.

**if** $v_0 \geq \lfloor \frac{L}{2} \rfloor + 1$ **then**

    **return** $0$   // failure

**else**

    **return** $\arg\max_{i \in [K]} v_i$   // success

**end if**

---

Our 2D doubling trick extends the vanilla doubling trick introduced in DSH. The vanilla doubling trick works by iterating through an input parameter (e.g., budget in DSH) at an exponentially growing rate to eventually reach the optimal value with only a logarithmic cost. Similarly, our 2D doubling trick in Algorithm 2 requires two input parameters, necessitating the design of a doubling scheme for both. While the budget doubling mirrors DSH, the voting scheme enhances confidence through the independence of repeated trials. Additionally, our choice of $L_1$ ensures the minimal number of repeated trials needed to achieve a confidence certificate of $\delta$, as presented in the following theorem.

**Theoretical analysis.** We first show that BrakeBooster is $\delta$-correct.

**Theorem 4.1** (Correctness). *Let an algorithm $\mathcal{A}$ be $\delta_0$-correct and have a sample complexity of $T^*_{\delta_0}(\mathcal{A})$. Suppose we run BrakeBooster (Algorithm 2) denoted by $\mathcal{M}$ with input $\mathcal{A}$, $\delta$, $L_1 = \lceil \frac{4\log(1+\frac{2}{\delta})}{\log \frac{1}{4e\delta_0}} \rceil$, $T_1 \geq 1$, and $\delta_0 \leq \frac{1}{(2e)^2}$. Then,*

$$\mathbb{P}\left(\tau(\mathcal{M}) < \infty, J(\mathcal{M}) \neq 1\right) \leq \delta.$$

The proof is provided in the Appendix. We briefly outline the key ideas underlying the approach. The main novelty in the proof of Theorem 4.1 lies in establishing a $\delta$-correct guarantee by repeatedly invoking a $\delta_0$-correct algorithm within a voting-based framework. A central component of this argument is Lemma C.2, which demonstrates that the voting mechanism induces an exponentially decaying failure probability, provided that $\delta_0$ is not too large—for instance, when $\delta_0 > \frac{1}{(2e)^2}$. Another notable aspect of our analysis is the design of the trial budget schedule $L_{r,c}$. Instead of doubling the sample size in each round as in standard schemes (e.g., using $2^r$), we apply a slightly more aggressive growth rule,

namely $r2^r$. This multiplicative factor of $r$ in the exponent is crucial, as it enables the use of a converging geometric series in the analysis. However, this doubling trick only adds a logarithmic term in the final sample complexity. In summary, 1) the progression of trial counts $L$ along with the majority voting scheme ensures an exponentially decaying stopping time distribution, and 2) the per-trial budget schedule guarantees that the $\delta_0$-correctness condition of the black-box subroutine is satisfied at some point, when the budget exceeds the unknown $T^*_{\delta_0}(\mathcal{A})$.

Importantly, our framework permits the use of any black-box algorithm with a relatively large $\delta_0$, and allows one to tune the initial budget parameter $L_1$ to obtain a significantly stronger $\delta$-correct guarantee.

Furthermore, BrakeBooster enjoys an exponential stopping tail.

**Theorem 4.2** (Exponential stopping tail). *Let an algorithm $\mathcal{A}$ be $\delta_0$-correct and have a sample complexity of $T^*_{\delta_0}(\mathcal{A})$. Suppose we run BrakeBooster (Algorithm 2) with input $\mathcal{A}$, $\delta$, $L_1 = \lceil \frac{4\log(1+\frac{2}{\delta})}{\log \frac{1}{4e\delta_0}} \rceil$, $T_1 \geq 1$, and $\delta_0 \leq (\frac{1}{2e})^2$. Then, there exists $T_0 = \tilde{\Theta}((T^*_{\delta_0}(\mathcal{A}) + T_1) \cdot \ln(1/\delta))$ such that*

$$\forall T \geq T_0, \ \mathbb{P}\left(\tau(\mathcal{M}) \geq T\right) \leq \exp\left(-\frac{T}{T^*_{\delta_0}(\mathcal{A}) \cdot O(\log T)}\right).$$

*That is, if $T_1$ is an absolute constant, then BrakeBooster enjoys a $(\tilde{\Theta}(T^*_{\delta_0}(\mathcal{A}) \ln(1/\delta)), T^*_{\delta_0}(\mathcal{A}))$-exponential stopping tail.*

The theorem above literally delivers the promised guarantee – it takes in an algorithm with a high probability sample complexity guarantee $T^*_{\delta_0}(\mathcal{A})$ and turns it into the one that enjoys an exponential stopping tail without losing the same high probability sample complexity guarantee due to Proposition 2.9(i), up to polylog($T$) factors. For example, one can use Successive Elimination (Even-Dar et al., 2006) to obtain the following guarantee.

**Corollary 4.3.** *Suppose we take Successive Elimination algorithm (Even-Dar et al., 2006) as $\mathcal{A}$ and run Brake-Booster algorithm with $\delta$, $L_1 = \lceil \frac{4\log(1+\frac{2}{\delta})}{\log \frac{1}{4e\delta_0}} \rceil$, $T_1 \geq 1$, and $\delta_0 = \left(\frac{1}{2e}\right)^2$. Then, BrakeBooster enjoys a $\left(\tilde{\Theta}\left(H_1 \ln\left(1/\delta\right)\right), \tilde{\mathcal{O}}(H_1)\right)$-exponential stopping tail.*

Figure 3 shows that applying BrakeBooster on Successive Elimination helps to stop all the trials without sacrificing too many samples (the CDF curve of BrakeBooster+SE catches up with that of SE very fast because the crossover point is at the stopping time of $\sim 0.05 \times 10^6$).

# 5. Related Work

While research on best arm identification can be traced back to the seminal work of Chernoff (1959), algorithms with fixed confidence with the correctness and the sample com-

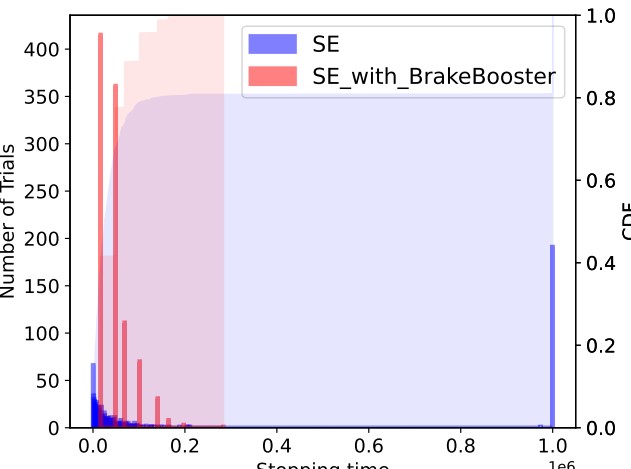

*Figure 3.* Historgram of stopping times of BrakeBooster applied on Successive Elimination (Even-Dar et al., 2006) vs bare Successive Elimination: Out of 1000 independent trials on four arms with mean rewards of $\{1.0, 0.9, 0.9, 0.9\}$ and Gaussian noise $\mathcal{N}(0, 1)$. We forcefully terminated the runs that do not stop until $1M$ time steps. ($\delta = 0.01$). CDF of stopping times are shaded with respective colors. (In this experiment, we employ a $1.2\times$ growth factor for both the per-trial budget and the number of trials, in contrast to the conventional doubling scheme.)

plexity guarantees first appeared in Even-Dar et al. (2006) where the authors proposed two influential algorithms: Median Elimination and Successive Elimination. The former shows a deterministic sample complexity of $\mathcal{O}\left(\frac{K}{\varepsilon^2} \log \frac{1}{\delta}\right)$ for finding an arm that is $\varepsilon$ close to the best arm with a probability of at least $1 - \delta$. This worst-case result is optimal up to a constant factor. The second algorithm, Successive Elimination, shows that, with a probability of at least $1 - \delta$, the sample complexity of identifying the best arm scales with an instance-dependent quantity $H_1 = \sum_{i=2}^{K} \frac{1}{\Delta_i^2}$, where $\Delta_i$ is the gap between the best arm and the $i$-th best arm. This result has had a significant influence on subsequent research on best arm identification. Since then, many algorithms have been proposed to improve the sample complexity of best arm identification in the fixed confidence setting. For instance, Kalyanakrishnan et al. (2012) propose the LUCB algorithm that extends Even-Dar et al. (2006) to the scenario where the algorithm is required to return the best $m$ arms instead of just the best arm. Karnin et al. (2013) and Jamieson et al. (2014) then propose algorithms with improved guarantees that turn problem-dependent logarithmic factors into doubly-logarithmic ones. Chen et al. (2017) have further improved both lower and upper bound guarantees on the high probability sample complexity. Garivier & Kaufmann (2016) propose the Track-and-Stop algorithm and Jourdan et al. (2022) propose Top Two algorithms that asymptotically matches the asymptotic lower bound for its sample complexity. Jourdan & Degenne (2023) propose a UCB-

based Top Two algorithm (TTUCB) which provides both asymptotic and non-asymptotic upper bounds on expected sample complexity. More recently, Jourdan et al. (2023) extend the Top Two framework with EB-TC$_{\varepsilon_0}$, where the objective is to identify an arm within $\varepsilon_0$ of the optimal arm. This algorithm achieves both asymptotically optimal and non-asymptotic guarantee in the $\varepsilon_0$-FC-BAI setting. However, notably, all these results focus on the expected or high-probability sample complexity, rather than the stopping time distribution. Therefore, these algorithms do not (yet) have a guarantee showing a light tail for the stopping time distribution, which does not exclude the possibility of running for a long time before stopping with a non-trivial probability.

Moreover, the fixed budget algorithm, in contrast, must return the best arm within a pre-specified budget. Audibert et al. (2010) propose the first fixed budget algorithm called Successive Rejects. They show that the probability of misidentifying the best arm scales with a problem-dependent quantity, $H_2 = \max_{i=2}^{K} \frac{i}{\Delta_i^2}$. Karnin et al. (2013) later improve this result by a logarithmic factor with an improved algorithm called Sequential Halving, which has been widely adopted in many applications including hyperparameter optimization (Li et al., 2018). A recent work by Zhao et al. (2023) builds upon Sequential Halving, addressing how to measure if the algorithm's output is good enough for any suboptimality gap $\varepsilon$, while $\varepsilon$ is free to be chosen after the algorithm finishes. Additionally, this work extends Sequential Halving to the challenging data-poor regime, where the number of samples is even smaller than the number of arms. While we leverage some fixed budget algorithms such as Sequential Halving, the main focus of this paper is the fixed confidence setting.

## 6. Conclusion

We have provided a new theoretical perspective on the behavior of the stopping time for the fixed confidence best arm identification algorithms, which inspires numerous open problems. First, both of our proposed algorithms introduce nontrivial extra constant or logarithmic factors in their sample complexity compared to the well-known optimal instance-dependent sample complexity that is achieved by existing algorithms (Garivier & Kaufmann, 2016). It would be interesting to investigate whether it is possible to develop novel algorithms that obtain instance-dependent optimality while attaining exponentially decaying tail bounds for the stopping time distribution. Second, BrakeBooster algorithm incurs a polylog($T$) term within the exponent. It would be interesting to investigate whether this term can be eliminated or, alternatively, to establish a lower bound that matches this dependence. Third, the resetting mechanism employed by our algorithms tends to be less practical. It would be interesting to investigate whether there exists a simple and/or elegant algorithm that can avoid the resetting mechanism

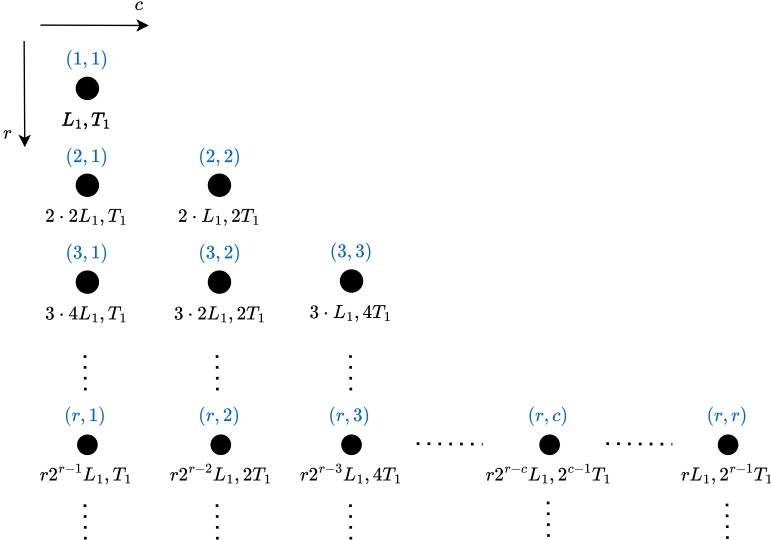

Figure 4. A diagram showing the progression of the stages in Algorithm 2 where each stage $(r, c)$ has a different trial count $L_{r,c} = r2^{r-c}L_1$ and a per-trial budget $T_{r,c} = 2^{c-1}T_1$.

and exhibit practical numerical performance. In particular, studying whether or not the recently proposed practical algorithms in Jourdan et al. (2022) achieve exponential tail bounds and, if not, developing remedies for them would be an interesting avenue of research. Finally, it would be interesting to attain similar exponential tail bounds for more complex settings such as combinatorial bandits or Markov decision processes.

## Impact Statement

This work is theoretical and not tied to a particular application that would have immediate negative impact.

## Acknowledgments

Kapilan Balagopalan, Tuan Ngo Nguyen, Yao Zhao, Kwang-Sung Jun were supported in part by the National Science Foundation under grant CCF-2327013 and Meta Platforms, Inc. We thank Emilie Kaufmann, Marc Jourdan, and Rémy Degenne for the discussion that sparked this project.

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

# Appendix

## Table of Contents

## A. Lower bounds

---

**Algorithm 4** Successive Elimination (Even-Dar et al., 2006)

---

   **Input**: $\delta$

   **Initialize** $t = 1$, $\mathcal{S} = [K]$, $\hat{\mu}_i = 0$ for $\forall i \in [K]$

   Sample each arm $i \in \mathcal{S}$ once

   **while** $|\mathcal{S}| > 1$ **do**

      Sample each arm $i \in \mathcal{S}$ once and update $\hat{\mu}_i$

      Set $S = S \setminus \left\{ i : \max_{j \in \mathcal{S}} \hat{\mu}_j - \hat{\mu}_i \geq \sqrt{\frac{2\ln\left(3.3t^2/\delta\right)}{t}} \right\}$

      $t = t + 1$

   **end while**

---

*Theorem 2.4* (Lower bound).  For Successive Elimination, there exists an instance with a unique best arm where the algorithm never stops with a constant probability.

*Proof.*  Consider an instance with $K = 3$ arms following Gaussian distributions $\mathcal{N}(1, 1)$, $\mathcal{N}(0.9, 1)$, and $\mathcal{N}(0.9, 1)$. The lower bound we show here is a consequence of the potential misbehave of the algorithm after it pulls each arm for once. Since all of the arms have same variance, we simply denote the any confidence width, with a probability $\delta$, of the sample mean after being pulled for $t$ times as $b_{\delta,t} = \sqrt{\frac{2\ln\left(3.3t^2/\delta\right)}{t}}$ by Theorem A.2. We define the following events:

- $E_1 := \left\{ \hat{\mu}_{1,1} \leq \mu_2 - 3b_{\delta,1} \right\}$.
- $E_2 := \left\{ \forall t > 0, \hat{\mu}_{2,t} \in \left( \mu_2 - b_{\delta,t}, \mu_2 + b_{\delta,t} \right) \right\}$.
- $E_3 := \left\{ \forall t > 0, \hat{\mu}_{3,t} \in \left( \mu_3 - b_{\delta,t}, \mu_3 + b_{\delta,t} \right) \right\}$.

We first note that the event $E_1 \cap E_2 \cap E_3$ implies that the algorithm eliminates the best arm after it pulls each arm for once and never stops. To see this, we have

$$\hat{\mu}_{1,1} \leq \mu_2 - 3b_{\delta,1} = \mu_2 - b_{\delta,1} - 2b_{\delta,1} \leq \hat{\mu}_{2,1} - 2b_{\delta,1}.$$

This implies that

$$\hat{\mu}_{1,1} + b_{\delta,1} \leq \hat{\mu}_{2,1} - b_{\delta,1},$$

which is the condition for the algorithm to eliminate the best arm. Next we lower bound the probabilities of the events $E_1$, $E_2$, and $E_3$. For the event $E_1$, by Theorem A.1,

$$\begin{aligned}
\mathbb{P}\left(E_1\right) &= \mathbb{P}\left(\hat{\mu}_{1,1} \leq \mu_2 - 3b_{\delta,1}\right) \\
&= \mathbb{P}\left(\mu_1 - \hat{\mu}_{1,1} > \Delta_2 + 3b_{\delta,1}\right) \\
&> \frac{1}{8\sqrt{\pi}} \exp\left(-\frac{7\left(\Delta_2 + 3b_{\delta,1}\right)^2}{2}\right) \\
&> \frac{1}{8\sqrt{\pi}} \exp\left(-\frac{7\left(4b_{\delta,1}\right)^2}{2}\right) \qquad\qquad (b_{\delta,1} \approx 1.94 > \Delta_2 \text{ for } \delta \leq 1/2)\\
&> \frac{1}{8\sqrt{\pi}} \left(\frac{\delta}{3.3}\right)^{118}.
\end{aligned}$$

By Theorem A.2, we have for $\delta \leq 1/2$,

$$\mathbb{P}\left(E_2 \cap E_3\right) = \mathbb{P}\left(E_2\right)\mathbb{P}\left(E_3\right) \geq (1-\delta)^2 \geq \frac{1}{4}.$$

Thus, we have

$$\mathbb{P}\left(E_1 \cap E_2 \cap E_3\right) = \mathbb{P}\left(E_1\right)\mathbb{P}\left(E_2\right)\mathbb{P}\left(E_3\right) > \frac{1}{32\sqrt{\pi}} \left(\frac{\delta}{3.3}\right)^{118}.$$

Thus with a constant probability, the algorithm never stops. $\qquad\square$

**Lemma A.1** (Anti-concentration inequality (Abramowitz & Stegun, 1968)). *For a Gaussian random variable $X \sim \mathcal{N}(\mu, \sigma^2)$ and any $z > 0$, we have*

$$\mathbb{P}\left(|X - \mu| > z\sigma\right) > \frac{1}{4\sqrt{\pi}} \exp\left(-\frac{7z^2}{2}\right).$$

**Lemma A.2** (Naive anytime confidence bound). *For a Gaussian random variable $X \sim \mathcal{N}(\mu, \sigma^2)$, the sample mean $\hat{\mu}_t$ satisfies*

$$\mathbb{P}\left(\forall t > 0, \ \hat{\mu}_t \in \left(\mu - \sqrt{\frac{2\sigma^2 \ln\left(3.3t^2/\delta\right)}{t}}, \ \mu + \sqrt{\frac{2\sigma^2 \ln\left(3.3t^2/\delta\right)}{t}}\right)\right) \geq 1 - \delta.$$

---

**Algorithm 5** KL-LUCB (Tanczos et al., 2017), adapted and simplified for sub-Gaussian distribution

---

**Input**: $\delta$
**Define**: $L_i\left(T_i(t), \delta\right) = \hat{\mu}_i - b_{\delta, T_i(t)}$ and $U_i\left(T_i(t), \delta\right) = \hat{\mu}_i + b_{\delta, T_i(t)}$
**Initialize**: Sample each arm once
**while** $L_{i^*}\left(T_{i^*}(t), \delta\right) < \max_{i \neq i^*} U_i(T_i(t), \delta)$ **do**
    Sample the following two arms:
    •    $i^* = \arg\max_{i \in [K]} \hat{\mu}_i$
    •    $i' = \arg\max_{i \neq i^*} U_i(T_i(t), \delta)$
    Update sample mean and confidence interval
    $t \leftarrow t + 2$
**end while**

---

*Theorem 2.5.* For KL-LUCB (Algorithm 5), there exists an instance with a unique best arm where the algorithm never stops with a constant probability.

*Proof.* The proof follows the same line as the proof of Theorem 2.4. We consider an instance with $K = 3$ arms following Gaussian distributions $\mathcal{N}(1,1)$, $\mathcal{N}(0.9,1)$, and $\mathcal{N}(0.9,1)$. We define the following events:

- $E_1 := \left\{ \hat{\mu}_{1,1} \leq \mu_2 - 3b_{\delta,1} \right\}$.
- $E_2 := \left\{ \forall t > 0, \hat{\mu}_{2,t} \in \left( \mu_2 - b_{\delta,t}, \mu_2 + b_{\delta,t} \right) \right\}$.
- $E_3 := \left\{ \forall t > 0, \hat{\mu}_{3,t} \in \left( \mu_3 - b_{\delta,t}, \mu_3 + b_{\delta,t} \right) \right\}$.

We first note that the event $E_1 \cap E_2 \cap E_3$ implies that the algorithm will never pull the best arm after it pulls each arm for once and never stops. To see this, we first show that the best arm has lowest sample mean among the three arms given $E_1 \cap E_2 \cap E_3$. We take the second arm as an example, but same argument for the third arm,

$$\hat{\mu}_{1,1} \leq \mu_2 - 3b_{\delta,1} = \mu_2 - b_{\delta,1} - 2b_{\delta,1} \leq \hat{\mu}_{2,1} - 2b_{\delta,1} < \hat{\mu}_{2,1}.$$

Thereby the algorithm will take $\arg\max_{i=2,3} \hat{\mu}_i$ as the best arm and pull both of arm 2 and 3 in the next round. Given the event $E_2 \cap E_3$, the confidence interval for arm 2 and 3 keep shrinking, while the confidence interval for arm 1 keeps still. Thus arm 1 will never be pulled again. Also given $E_2 \cap E_3$, the confidence interval for arm 2 and 3 will always overlap, thus the algorithm will never stop.

The lower bound of the probabilities of the events $E_1 \cap E_2 \cap E_3$ is same as the proof of Theorem 2.4. □

## B. Doubling Sequential Halving with fixed confidence (FC-DSH)

Throughout this section, acute readers will notice that we may talk about events that happen in phase $m$ without having a condition that the algorithm has not stopped before. To deal with this without notational overload, we take the model where the algorithm has already been run for all phases without stopping, and the user of the algorithm only reveals what happened already and stop when the stopping condition is met. This way, we can talk about events in any phase without adding conditions on whether the algorithm has stopped or not (and this is valid because the samples are independent between phases).

### B.1. Proof of FC-DSH's Correctness

*Theorem 3.1* (Correctness). FC-DSH runs with confidence level $\delta$ is $\delta$-correct.

*Proof.* Following the definition of $\delta$-correction in 2.1, we need to prove that
$$\mathbb{P}\left(\tau < \infty,\, J \neq 1\right) \leq \delta.$$

For each phase $m \in \{1, 2, \ldots\}$ in FC-DSH, for each stage $\ell \in \left[\lceil \log_2(K) \rceil\right]$, for each arm $i \in [K]$, recall the confidence width

$$b_i^{(m)} = \sqrt{\frac{2}{N^{(m,\ell_i)}} \log\left(\frac{6K \lceil \log_2(K) \rceil m^2}{\delta}\right)}$$

and define the following events:

$$G := \left\{\forall i \in [K],\, \forall m \in \{1, 2, \ldots\},\, \forall \ell \in \left[\lceil \log_2(K) \rceil\right],\, \left|\hat{\mu}_i^{(m,\ell)} - \mu_i\right| \leq b_i^{(m,\ell)}\right\},$$

and

$$G^c := \left\{\exists i \in [K],\, \exists m \in \{1, 2, \ldots\},\, \exists \ell \in \left[\lceil \log_2(K) \rceil\right],\, \left|\hat{\mu}_i^{(m,\ell)} - \mu_i\right| > b_i^{(m,\ell)}\right\}.$$

We first claim that the probability of event $G^c$ happens is at most $\delta$. The proof is as follows

$$\mathbb{P}\left(G^c\right)$$

$$\leq \sum_{m=1}^{\infty} \sum_{\ell=1}^{\lceil \log_2(K) \rceil} \sum_{i=1}^{K} \mathbb{P}\left(\left|\hat{\mu}_i^{(m,\ell)} - \mu_i\right| > \sqrt{\frac{2}{N_i^{(m,\ell)}} \log\left(\frac{6K \lceil \log_2(K) \rceil m^2}{\delta}\right)}\right) \quad \text{(use union bound)}$$

$$\leq \sum_{m=1}^{\infty} \sum_{\ell=1}^{\lceil \log_2(K) \rceil} \sum_{i=1}^{K} 2\exp\left(-\frac{N_i^{(m,\ell)}\left(\sqrt{\frac{2}{N_i^{(m,\ell)}} \log\left(\frac{6K \lceil \log_2(K) \rceil m^2}{\delta}\right)}\right)^2}{2}\right) \quad \text{(use Hoeffding's inequality)}$$

$$= \sum_{m=1}^{\infty} \sum_{\ell=1}^{\lceil \log_2(K) \rceil} \sum_{i=1}^{K} \frac{2\delta}{6K \lceil \log_2(K) \rceil m^2}$$

$$= \sum_{m=1}^{\infty} \sum_{\ell=1}^{\lceil \log_2(K) \rceil} \sum_{i=1}^{K} \frac{\delta}{3K \lceil \log_2(K) \rceil m^2}$$

$$= \sum_{m=1}^{\infty} \frac{\delta}{3m^2}$$

$$= \frac{\pi^2}{6} \cdot \frac{\delta}{3} \qquad \text{(use geometric sum } \sum_{m=1}^{\infty} \frac{1}{m^2} = \frac{\pi^2}{6}\text{)}$$
$$\leq \delta.$$

Secondly, we claim that under the event $G$, FC-DSH never outputs a suboptimal arm, formally, $\mathbb{P}\left(J \neq 1, G\right) = 0$. We prove this claim by contradiction.

Suppose FC-DSH outputs arm $J \neq 1$. From the stopping condition $L_J^{(m)} \geq \max_i U_{i \neq J}^{(m)}$,

$$L_J^{(m)} \geq \max_i U_{i \neq J}^{(m)}$$
$$\Rightarrow L_J^{(m)} \geq U_1^{(m)}$$
$$\Rightarrow \hat{\mu}_J^{(m)} - b_J^{(m)} \geq \hat{\mu}_1^{(m)} + b_1^{(m)}.$$

Under event $G$, we have $\mu_J \geq \hat{\mu}_J^{(m)} - b_J^{(m)}$ and $\hat{\mu}_1^{(m)} + b_1^{(m)} \geq \mu_1$, which implies

$$\mu_J \geq \hat{\mu}_J^{(m)} - b_J^{(m)} \geq \hat{\mu}_1^{(m)} + b_1^{(m)} \geq \mu_1.$$

Therefore, we have $\mu_J \geq \mu_1$ which is a contradiction.

Finally, we combine both claims to show

$$\mathbb{P}\left(\tau < \infty, J \neq 1\right)$$
$$= \mathbb{P}\left(\tau < \infty, J \neq 1, G^c\right) + \mathbb{P}\left(\tau < \infty, J \neq 1, G\right)$$
$$\leq \mathbb{P}\left(G^c\right) + \mathbb{P}\left(J \neq 1, G\right)$$
$$\leq \delta + 0 = \delta.$$

$\square$

## B.2. Proof of FC-DSH's Exponential Stopping Tail

*Theorem 3.2* (Exponential stopping tail). Let $T_\delta = 4096 H_2 \log_2(K) \log\left(\frac{6K \log_2(K)}{\delta}\right)$ and $\kappa = 4096 H_2 \log_2(K)$. Then, for all $T \geq 2T_\delta$, FC-DSH satisfies

$$\mathbb{P}\left(\tau \geq T\right) \leq \exp\left(-\frac{T}{2\kappa}\right).$$

*Proof.* To avoid redundancy and for the sake of readability, from now on, we assume $K$ is of a power of 2. Hence $\lceil \log_2(K) \rceil = \log_2(K)$. It is easy to verify the result for any $K$.

Denote by $m_\tau = \min\left\{m \geq 1 : L_{J_m}^{(m)} \geq \max_{i \neq J_m} U_i^{(m)}\right\}$ the stopping phase of FC-DSH. Then, $\tau$ corresponds to the total number of samples at the end of stopping phase $m_\tau$.

It suffices to show for all phase $m$ such that $T_m \geq T_\delta$, $\mathbb{P}\left(\tau \geq T_m\right) \leq \exp\left(-\frac{T_m}{\kappa}\right)$. Then, we can apply Lemma B.1 to obtain for all $T \geq 2T_\delta$, $\mathbb{P}\left(\tau \geq T\right) \leq \exp\left(-\frac{T}{2\kappa}\right)$.

Recall the stopping condition: at the end of phase $m$, FC-DSH outputs an arm $J_m$, and FC-DSH will stop if the condition $L_{J_m}^{(m)} \geq \max_{i \neq J_m} U_i^{(m)}$ is satisfied.

Let $m$ be any phase such that $T_m \geq T_\delta$. We start off by decomposing probability bound according to the stopping condition

$$\mathbb{P}\left(\tau \geq T_m\right)$$
$$\leq \prod_{m'=1}^{m} \mathbb{P}\left(\text{phase } m' \text{ failed to stop}\right)$$
$$\leq \mathbb{P}\left(\text{phase } m \text{ failed to stop}\right)$$

$$= \mathbb{P}\left(L_{J_m}^{(m)} < \max_{i \neq J_m} U_i^{(m)}\right)$$

$$= \mathbb{P}\left(L_{J_m}^{(m)} < \max_{i \neq J_m} U_i^{(m)}, \, J_m \neq 1\right) + \mathbb{P}\left(L_{J_m}^{(m)} < \max_{i \neq J_m} U_i^{(m)}, \, J_m = 1\right)$$

$$\leq \mathbb{P}\left(L_{J_m}^{(m)} < \max_{i \neq J_m} U_i^{(m)}, \, J_m \neq 1\right) + \sum_{i \neq 1} \mathbb{P}\left(L_{J_m}^{(m)} < U_i^{(m)}, \, J_m = 1\right).$$

For the first term, we obtain the following probability bound

$$\mathbb{P}\left(L_{J_m}^{(m)} < \max_{i \neq J_m} U_i^{(m)}, \, J_m \neq 1\right)$$

$$\leq \mathbb{P}\left(J_m \neq 1\right)$$

$$\leq \exp\left(-\frac{T_m}{512 H_2 \log_2(K)}\right). \qquad \text{(use Lemma B.5)}$$

In phase $m$, for each $i \in [K]$, we denote by $\ell_i$ the stage at which arm $i$ is eliminated. We define $\ell_i^*$ to be the largest stage $\ell_i$ such that $\Delta_i \leq \frac{1}{2}\Delta_{\frac{K}{4}\cdot 2^{-\ell_i+1}}$. We observe that there are arms $i$ that have no such $\ell_i^*$ and there are arms $i$ that $\ell_i^*$ exists. For the former case, if an arm $i$ has no such $\ell_i^*$, it implies $\Delta_i > \Delta_{\frac{K}{4}}$. We further split the second term accordingly

$$\sum_{i \neq 1} \mathbb{P}\left(L_{J_m}^{(m)} < U_i^{(m)}, \, J_m = 1\right)$$

$$\leq \sum_{\substack{i \neq 1: \\ \Delta_i > \Delta_{\frac{K}{4}}}} \mathbb{P}\left(L_{J_m}^{(m)} < U_i^{(m)}, \, J_m = 1\right) + \sum_{\substack{i \neq 1: \\ \ell_i^* \text{ exists}}} \mathbb{P}\left(L_{J_m}^{(m)} < U_i^{(m)}, \, J_m = 1\right)$$

$$\leq \sum_{\substack{i \neq 1: \\ \Delta_i > \Delta_{\frac{K}{4}}}} \mathbb{P}\left(L_{J_m}^{(m)} < U_i^{(m)}, \, J_m = 1\right)$$

$$+ \sum_{\substack{i \neq 1: \\ \ell_i^* \text{ exists}}} \left(\mathbb{P}\left(L_{J_m}^{(m)} < U_i^{(m)}, \, J_m = 1, \, \ell_i \geq \ell_i^*\right) + \mathbb{P}\left(L_{J_m}^{(m)} < U_i^{(m)}, \, J_m = 1, \, \ell_i < \ell_i^*\right)\right).$$

For the first subterm, for all arm $i$ that satisfies $i \neq 1$, $\Delta_i > \frac{1}{2}\Delta_{\frac{K}{4}}$, we apply Lemma B.2 to obtain the following probability bound

$$\mathbb{P}\left(L_1^{(m)} < U_i^{(m)}, \, J_m = 1\right) \leq \exp\left(-\frac{T_m}{512 H_2 \log_2(K)}\right).$$

For the second and third subterms, for all arm $i$ that satisfies $i \neq 1$ and $\ell_i^*$ exists, we apply Lemma B.3 to obtain the following probability bound

$$\mathbb{P}\left(L_1^{(m)} < U_i^{(m)}, \, \ell_i \geq \ell_i^*, \, J_m = 1\right) \leq \exp\left(-\frac{T_m}{1024 H_2 \log_2(K)}\right),$$

and we apply Lemma B.4 to obtain the following probability bound

$$\mathbb{P}\left(\ell_i < \ell_i^*, \, J_m = 1\right) \leq \exp\left(-\frac{T_m}{2048 H_2 \log_2(K)}\right).$$

Hence, for the second term, we obtain the following probability bound

$$\sum_{i \neq 1} \left(\mathbb{P}\left(L_1^{(m)} < U_i^{(m)}, \, \ell_i \geq \ell_i^*, \, J_m = 1\right) + \mathbb{P}\left(\ell_i < \ell_i^*, \, J_m = 1\right)\right)$$

$$\leq \sum_{\substack{i \neq 1: \\ \Delta_i > \Delta_{\frac{K}{4}}}} \mathbb{P}\left(L_{J_m}^{(m)} < U_i^{(m)}, \, J_m = 1\right)$$

$$+ \sum_{\substack{i \neq 1: \\ \ell_i^* \text{ exists}}} \left( \mathbb{P}\left( L_{J_m}^{(m)} < U_i^{(m)}, J_m = 1, \ell_i \geq \ell_i^* \right) + \mathbb{P}\left( L_{J_m}^{(m)} < U_i^{(m)}, J_m = 1, \ell_i < \ell_i^* \right) \right)$$

$$\leq \sum_{\substack{i \neq 1: \\ \Delta_i > \Delta_{\frac{K}{4}}}} \exp\left( -\frac{T_m}{512 H_2 \log_2(K)} \right)$$

$$+ \sum_{\substack{i \neq 1: \\ \ell_i^* \text{ exists}}} \left( \exp\left( -\frac{T_m}{1024 H_2 \log_2(K)} \right) + \exp\left( -\frac{T_m}{2048 H_2 \log_2(K)} \right) \right)$$

$$\leq 2 \sum_{i \neq 1} \exp\left( -\frac{T_m}{2048 H_2 \log_2(K)} \right)$$

$$\leq 2(K-1) \exp\left( -\frac{T_m}{2048 H_2 \log_2(K)} \right).$$

Finally, we combine two terms and obtain the probability bound

$$\mathbb{P}\left( L_{J_m}^{(m)} < \max_{i \neq J_m} U_i^{(m)}, J_m \neq 1 \right) + \sum_{i \neq 1} \mathbb{P}\left( L_{J_m}^{(m)} < U_i^{(m)}, J_m = 1 \right)$$

$$= \exp\left( -\frac{T_m}{512 H_2 \log_2(K)} \right) + 2(K-1) \exp\left( -\frac{T_m}{2048 H_2 \log_2(K)} \right)$$

$$\leq 2K \exp\left( -\frac{T_m}{2048 H_2 \log_2(K)} \right)$$

$$= \exp\left( -\frac{T_m}{2048 H_2 \log_2(K)} + \log(2K) \right).$$

The condition $T_m \geq T_\delta = 4096 H_2 \log_2(K) \log\left( \frac{6K \log_2(K)}{\delta} \right)$ implies that

$$T_m \geq 4096 H_2 \log_2(K) \log\left( \frac{6K \log_2(K)}{\delta} \right)$$

$$\Rightarrow T_m \geq 4096 H_2 \log_2(K) \log(2K) \qquad\qquad \text{(use } \delta \leq 1\text{)}$$

$$\Rightarrow \log(2K) \leq \frac{T_m}{4096 H_2 \log_2(K)}$$

$$\Rightarrow -\frac{T_m}{2048 H_2 \log_2(K)} + \log(2K) \leq -\frac{T_m}{4096 H_2 \log_2(K)}.$$

Therefore,

$$\exp\left( -\frac{T_m}{2048 H_2 \log_2(K)} + \log(2K) \right)$$

$$\leq \exp\left( -\frac{T_m}{4096 H_2 \log_2(K)} \right),$$

which concludes the proof of our theorem.

$\square$

**Lemma B.1.** *Suppose for all phase $m$ such that $T_m \geq T_\delta = 4096 H_2 \log_2(K) \log\left( \frac{6K \log_2(K)}{\delta} \right)$, for a constant c, FC-DSH achieves*

$$\mathbb{P}(\tau \geq T_m) \leq \exp\left( -\frac{T_m}{c H_2 \log_2(K)} \right)$$

*then, for all $T \geq 2T_\delta$, FC-DSH achieves*

$$\mathbb{P}(\tau \geq T) \leq \exp\left( -\frac{T}{2c H_2 \log_2(K)} \right)$$

*Proof.* Let $T \geq 2T_\delta$. Then, there exists $m$ such that

$$T_m \leq T < T_{m+1}.$$

By the FC-DSH setup, $2T_m = T_{m+1}$, thus $T < T_{m+1} = 2T_m$

Then,

$$
\begin{aligned}
&\mathbb{P}\left(\tau \geq T\right) \\
&\leq \mathbb{P}\left(\tau \geq T_m\right) && \text{(use } T \geq T_m) \\
&\leq \exp\left(-\frac{T_m}{cH_2 \log_2(K)}\right) \\
&\leq \exp\left(-\frac{T}{2cH_2 \log_2(K)}\right). && \text{(use } T < 2T_m)
\end{aligned}
$$

$\square$

**Lemma B.2.** *Let $u$ be any arm that satisfies $u \neq 1$, $\Delta_u > \frac{1}{2}\Delta_{\frac{K}{4}}$. For all phase $m$ such that $T_m \geq T_\delta = 4096H_2 \log_2(K) \log\left(\frac{6K \log_2(K)}{\delta}\right)$, we obtain*

$$\mathbb{P}\left(L_1^{(m)} < U_u^{(m)}, J_m = 1\right) \leq \exp\left(-\frac{T_m}{512H_2 \log_2(K)}\right).$$

*Proof.* In the following proof, we abbreviate $\log(\cdot) = \log\left(\frac{6K \log_2(K)m^2}{\delta}\right)$ for clarity.

We have

$$
\begin{aligned}
&\mathbb{P}\left(L_1^{(m)} < U_u^{(m)}, J_m = 1\right) \\
&= \mathbb{P}\left(\hat{\mu}_1^{(m)} - \sqrt{\frac{2}{N^{(m,\ell_1)}}\log(\cdot)} < \hat{\mu}_u^{(m)} + \sqrt{\frac{2}{N^{(m,\ell_u)}}\log(\cdot)}, J_m = 1\right) && \text{(by the definition of } L_1^{(m)} \text{ and } U_1^{(m)}) \\
&= \mathbb{P}\left(\hat{\mu}_1^{(m)} - \hat{\mu}_u^{(m)} < \sqrt{\frac{2}{N^{(m,\ell_u)}}\log(\cdot)} + \sqrt{\frac{2}{N^{(m,\ell_1)}}\log(\cdot)}, J_m = 1\right) \\
&\leq \mathbb{P}\left(\hat{\mu}_1^{(m)} - \hat{\mu}_u^{(m)} < 2\sqrt{\frac{2}{N^{(m,\ell_u)}}\log(\cdot)}, J_m = 1\right) && \text{(since } J_m = 1, N^{(m,\ell_1)} \geq N^{(m,\ell_u)}) \\
&= \mathbb{P}\left(\hat{\mu}_1^{(m)} - \mu_1 - \hat{\mu}_u^{(m)} + \mu_u < -\Delta_u + 2\sqrt{\frac{2}{N^{(m,\ell_u)}}\log(\cdot)}, J_m = 1\right).
\end{aligned}
$$

Lemma B.6 states that for all arm $u$ such that $u \neq 1$, $\Delta_u > \frac{1}{2}\Delta_{\frac{K}{4}}$, for all phase $m$ such that $T_m \geq T_\delta$, $N^{(m,\ell_u)} \geq \frac{32}{\Delta_u^2} \log\left(\frac{6K \log_2(K)m^2}{\delta}\right)$. This inequality implies that

$$
\begin{aligned}
N^{(m,\ell_u)} &\geq \frac{32}{\Delta_u^2} \log\left(\frac{6K \log_2(K)m^2}{\delta}\right) \\
\Rightarrow \Delta_u^2 &\geq \frac{32}{N^{(m,\ell_u)}} \log\left(\frac{6K \log_2(K)m^2}{\delta}\right) \\
\Rightarrow \Delta_u &\geq 4\sqrt{\frac{2}{N^{(m,\ell_u)}} \log\left(\frac{6K \log_2(K)m^2}{\delta}\right)} \\
\Rightarrow -\Delta_u + 2\sqrt{\frac{2}{N^{(m,\ell_u)}} \log\left(\frac{6K \log_2(K)m^2}{\delta}\right)} &\leq -\frac{\Delta_u}{2}.
\end{aligned}
$$

Hence,

$$
= \mathbb{P}\left(\hat{\mu}_1^{(m)} - \mu_1 - \hat{\mu}_u^{(m)} + \mu_u < -\Delta_u + 2\sqrt{\frac{2}{N^{(m,\ell_u)}} \log\left(\frac{6K \log_2(K) m^2}{\delta}\right)}, \ J_m = 1\right)
$$

$$
\leq \mathbb{P}\left(\hat{\mu}_1^{(m)} - \mu_1 - \hat{\mu}_u^{(m)} + \mu_u < -\frac{\Delta_u}{2}, \ J_m = 1\right)
$$

$$
= \sum_{z=1}^{L} \mathbb{P}\left(\hat{\mu}_1^{(m)} - \mu_1 - \hat{\mu}_u^{(m)} + \mu_u < -\frac{\Delta_u}{2}, \ \ell_u = z, \ J_m = 1\right)
$$

$$
\leq \sum_{z=1}^{L} \exp\left(-\frac{\frac{\Delta_u^2}{4}}{2\left(\frac{1}{N^{(m,L)}} + \frac{1}{N^{(m,z)}}\right)}\right) \qquad \text{(use Hoeffding's inequality)}
$$

$$
\leq \sum_{z=1}^{L} \exp\left(-\frac{\Delta_u^2 N^{(m,1)}}{16}\right) \qquad \text{(use } L \geq 1 \text{ and } z \geq 1\text{)}
$$

$$
= \sum_{z=1}^{L} \exp\left(-\frac{\Delta_u^2}{16} \cdot \frac{T_m}{K \log_2(K)}\right) \qquad \text{(by def } N^{(m,1)} = \frac{T_m}{K \log_2(K)}\text{)}
$$

$$
\leq \sum_{z=1}^{L} \exp\left(-\frac{\Delta_{\frac{K}{4}}^2}{64} \cdot \frac{T_m}{K \log_2(K)}\right) \qquad \text{(use } \Delta_u > \frac{1}{2}\Delta_{\frac{K}{4}}\text{)}
$$

$$
\leq \sum_{z=1}^{L} \exp\left(-\frac{\Delta_{\frac{K}{4}}^2}{\frac{K}{4}} \cdot \frac{T_m}{256 \log_2(K)}\right)
$$

$$
\leq \sum_{z=1}^{L} \exp\left(-\frac{1}{H_2} \cdot \frac{T_m}{256 \log_2(K)}\right) \qquad \text{(by the definition } H_2 = \max_i i\Delta_i^{-2}\text{)}
$$

$$
= \sum_{z=1}^{L} \exp\left(-\frac{T_m}{256 H_2 \log_2(K)}\right)
$$

$$
\leq \log_2(K) \exp\left(-\frac{T_m}{256 H_2 \log_2(K)}\right)
$$

$$
\leq \exp\left(-\frac{T_m}{256 H_2 \log_2(K)} + \log(\log_2(K))\right).
$$

The condition $T_m \geq T_\delta = 4096 H_2 \log_2(K) \log\left(\frac{6K \log_2(K)}{\delta}\right)$ implies that

$$
T_m \geq 4096 H_2 \log_2(K) \log\left(\frac{6K \log_2(K)}{\delta}\right)
$$

$$
\Rightarrow T_m \geq 512 H_2 \log_2(K) \log\left(\log_2(K)\right) \qquad \text{(use } \delta \leq 1\text{)}
$$

$$
\Rightarrow \log\left(\log_2(K)\right) \leq \frac{T_m}{512 H_2 \log_2(K)}
$$

$$
\Rightarrow -\frac{T_m}{256 H_2 \log_2(K)} + \log\left(\log_2(K)\right) \leq -\frac{T_m}{512 H_2 \log_2(K)}.
$$

Therefore,

$$
\exp\left(-\frac{T_m}{256 H_2 \log_2(K)} + \log(\log_2(K))\right)
$$

$$
\leq \exp\left(-\frac{T_m}{512 H_2 \log_2(K)}\right).
$$

□

**Lemma B.3.** *Let $u \neq 1$ be any non-optimal arm. For all phase $m$ such that $T_m \geq T_\delta = 4096H_2\log_2(K)\log\left(\frac{6K\log_2(K)}{\delta}\right)$, assuming $\ell_u^*$ exists, we obtain*

$$\mathbb{P}\left(L_1^{(m)} < U_u^{(m)}, \ell_u \geq \ell_u^*, J_m = 1\right) \leq \exp\left(-\frac{T_m}{1024H_2\log_2(K)}\right).$$

*Proof.* Recall that $\ell_u^*$ is defined to be the largest stage $\ell$ such that $\Delta_u \leq \frac{1}{2}\Delta_{\frac{K}{4}\cdot 2^{-\ell+1}}$. In the following proof, we abbreviate $\log(\cdot) = \log\left(\frac{6K\log_2(K)m^2}{\delta}\right)$ for brevity.

We have

$$\mathbb{P}\left(L_1^{(m)} < U_u^{(m)}, \ell_u \geq \ell_u^*, J_m = 1\right)$$

$$= \mathbb{P}\left(\hat{\mu}_1^{(m)} - \sqrt{\frac{2}{N^{(m,\ell_1)}}\log(\cdot)} < \hat{\mu}_u^{(m)} + \sqrt{\frac{2}{N^{(m,\ell_u)}}\log(\cdot)}, \ell_u \geq \ell_u^*, J_m = 1\right)$$

$$\text{(by the definition of } L_1^{(m)} \text{ and } U_1^{(m)}\text{)}$$

$$\leq \mathbb{P}\left(\hat{\mu}_1^{(m)} - \sqrt{\frac{2}{N^{(m,\ell_1)}}\log(\cdot)} < \hat{\mu}_u^{(m)} + \sqrt{\frac{2}{N^{(m,\ell_u^*)}}\log(\cdot)}, \ell_u \geq \ell_u^*, J_m = 1\right) \quad \text{(use } \ell_u \geq \ell_u^*\text{)}$$

$$= \mathbb{P}\left(\hat{\mu}_1^{(m)} - \hat{\mu}_u^{(m)} < \sqrt{\frac{2}{N^{(m,\ell_u^*)}}\log(\cdot)} + \sqrt{\frac{2}{N^{(m,\ell_1)}}\log(\cdot)}, \ell_u \geq \ell_u^*, J_m = 1\right)$$

$$\leq \mathbb{P}\left(\hat{\mu}_1^{(m)} - \hat{\mu}_u^{(m)} < 2\sqrt{\frac{2}{N^{(m,\ell_u^*)}}\log(\cdot)}, \ell_u \geq \ell_u^*, J_m = 1\right) \quad \text{(since } J_m = 1, N^{(m,\ell_1)} \geq N^{(m,\ell_i^*)}\text{)}$$

$$= \mathbb{P}\left(\hat{\mu}_1^{(m)} - \mu_1 - \hat{\mu}_u^{(m)} + \mu_u < -\Delta_u + 2\sqrt{\frac{2}{N^{(m,\ell_u^*)}}\log(\cdot)}, \ell_u \geq \ell_u^*, J_m = 1\right).$$

Lemma B.7 states that for any arm $u$ such that $u \neq 1$ and $\ell_u^*$ exists, for all phase $m$ such that $T_m \geq T_\delta$, $N^{(m,\ell_u^*)} \geq \frac{32}{\Delta_u^2}\log\left(\frac{6K\log_2(K)m^2}{\delta}\right)$. This inequality implies that

$$N^{(m,\ell_u^*)} \geq \frac{32}{\Delta_u^2}\log\left(\frac{6K\log_2(K)m^2}{\delta}\right)$$

$$\Rightarrow \Delta_u^2 \geq \frac{32}{N^{(m,\ell_u^*)}}\log\left(\frac{6K\log_2(K)m^2}{\delta}\right)$$

$$\Rightarrow \Delta_u \geq 4\sqrt{\frac{2}{N^{(m,\ell_u^*)}}\log\left(\frac{6K\log_2(K)m^2}{\delta}\right)}$$

$$\Rightarrow -\Delta_u + 2\sqrt{\frac{2}{N^{(m,\ell_u^*)}}\log\left(\frac{6K\log_2(K)m^2}{\delta}\right)} \leq -\frac{\Delta_u}{2}.$$

Hence,

$$= \mathbb{P}\left(\hat{\mu}_1^{(m)} - \mu_1 - \hat{\mu}_u^{(m)} + \mu_u < -\Delta_u + 2\sqrt{\frac{2}{N^{(m,\ell_u^*)}}\log\left(\frac{6K\log_2(K)m^2}{\delta}\right)}, \ell_u \geq \ell_u^*, J_m = 1\right)$$

$$\leq \mathbb{P}\left(\hat{\mu}_1^{(m)} - \mu_1 - \hat{\mu}_u^{(m)} + \mu_u < -\frac{\Delta_u}{2}, \ell_u \geq \ell_u^*, J_m = 1\right)$$

$$= \sum_{z=\ell_u^*}^{L}\mathbb{P}\left(\hat{\mu}_1^{(m)} - \mu_1 - \hat{\mu}_u^{(m)} + \mu_u < -\frac{\Delta_u}{2}, \ell_u = z, J_m = 1\right) \quad (L = \log_2(K))$$

$$\leq \sum_{z=\ell_u^*}^{L} \exp\left(-\frac{\frac{\Delta_u^2}{4}}{2\left(\frac{1}{N^{(m,L)}} + \frac{1}{N^{(m,z)}}\right)}\right) \qquad \text{(use Hoeffding's inequality)}$$

$$\leq \sum_{z=\ell_u^*}^{L} \exp\left(-\frac{\Delta_u^2 N^{(m,\ell_u^*)}}{16}\right) \qquad \text{(use } L \geq \ell_u^* \text{ and } z \geq \ell_u^*)$$

$$= \sum_{z=\ell_u^*}^{L} \exp\left(-\frac{\Delta_u^2}{16} \cdot \frac{T_m}{2^{-\ell_u^*+1} K \log_2(K)}\right) \qquad \text{(by def } N^{(m,\ell_u^*)} = \frac{T_m}{2^{-\ell_u^*+1} K \log_2(K)})$$

There are two cases: (1) $\frac{K}{4} 2^{-\ell_u^*} \geq 1$ and (2) $\frac{K}{4} 2^{-\ell_u^*} < 1$.

In the first case: by the definition of $\ell_u^*$, we have $\frac{1}{2}\Delta_{\frac{K}{4} \cdot 2^{-\ell_u^*}} \leq \Delta_u$. Thus, we have

$$\exp\left(-\frac{\Delta_u^2}{16} \cdot \frac{T_m}{2^{-\ell_u^*+1} K \log_2(K)}\right)$$

$$\leq \exp\left(-\frac{\Delta_{\frac{K}{4} 2^{-\ell_u^*}}^2}{64} \cdot \frac{T_m}{2^{-\ell_u^*+1} K \log_2(K)}\right)$$

$$= \exp\left(-\frac{\Delta_{\frac{K}{4} 2^{-\ell_u^*}}^2}{\frac{K}{4} 2^{-\ell_u^*}} \cdot \frac{T_m}{512 \log_2(K)}\right)$$

$$\leq \exp\left(-\frac{1}{H_2} \cdot \frac{T_m}{512 \log_2(K)}\right) \qquad \text{(by the definition } H_2 = \max_i i\Delta_i^{-2})$$

$$= \exp\left(-\frac{T_m}{512 H_2 \log_2(K)}\right).$$

In the second case $\frac{K}{4} 2^{-\ell_u^*} < 1$, we have

$$\exp\left(-\frac{\Delta_u^2}{16} \cdot \frac{T_m}{2^{-\ell_u^*+1} K \log_2(K)}\right)$$

$$= \exp\left(-\frac{\Delta_u^2}{128} \cdot \frac{T_m}{\frac{K}{4} 2^{-\ell_u^*} \log_2(K)}\right)$$

$$\leq \exp\left(-\frac{\Delta_u^2}{128} \cdot \frac{T_m}{\log_2(K)}\right) \qquad \text{(use } \frac{K}{4} 2^{-\ell_u^*} < 1)$$

$$\leq \exp\left(-\frac{\Delta_u^2}{u} \cdot \frac{T_m}{128 \log_2(K)}\right) \qquad \text{(use } u \geq 1)$$

$$\leq \exp\left(-\frac{1}{H_2} \cdot \frac{uT_m}{128 \log_2(K)}\right) \qquad \text{(by the definition } H_2 = \max_i i\Delta_i^{-2})$$

$$= \exp\left(-\frac{T_m}{128 H_2 \log_2(K)}\right)$$

$$\leq \exp\left(-\frac{T_m}{512 H_2 \log_2(K)}\right).$$

In both cases, we have

$$\exp\left(-\frac{\Delta_u^2}{16} \cdot \frac{T_m}{2^{-\ell_u^*+1} K \log_2(K)}\right)$$

$$\leq \exp\left(-\frac{T_m}{512 H_2 \log_2(K)}\right).$$

Therefore,

$$\sum_{z=\ell_u^*}^{L} \exp\left(-\frac{\Delta_u^2}{16} \cdot \frac{T_m}{2^{-\ell_u^*+1} K \log_2(K)}\right)$$

$$\leq \sum_{z=\ell_u^*}^{L} \exp\left(-\frac{T_m}{512 H_2 \log_2(K)}\right)$$

$$\leq \log_2(K) \exp\left(-\frac{T_m}{512 H_2 \log_2(K)}\right)$$

$$\leq \exp\left(-\frac{T_m}{512 H_2 \log_2(K)} + \log(\log_2(K))\right).$$

The condition $T_m \geq T_\delta = 4096 H_2 \log_2(K) \log\left(\frac{6K \log_2(K)}{\delta}\right)$ implies that

$$T_m \geq 4096 H_2 \log_2(K) \log\left(\frac{6K \log_2(K)}{\delta}\right)$$

$$\Rightarrow T_m \geq 1024 H_2 \log_2(K) \log\left(\log_2(K)\right) \qquad \text{(use } \delta \leq 1)$$

$$\Rightarrow \log\left(\log_2(K)\right) \leq \frac{T_m}{1024 H_2 \log_2(K)}$$

$$\Rightarrow -\frac{T_m}{512 H_2 \log_2(K)} + \log\left(\log_2(K)\right) \leq -\frac{T_m}{1024 H_2 \log_2(K)}.$$

Therefore,

$$\exp\left(-\frac{T_m}{512 H_2 \log_2(K)} + \log(\log_2(K))\right)$$

$$\leq \exp\left(-\frac{T_m}{1024 H_2 \log_2(K)}\right).$$

$\square$

**Lemma B.4.** *Let* $u \neq 1$ *be any arm non-optimal.* *For all phase* $m$ *such that* $T_m \geq T_\delta = 4096 H_2 \log_2(K) \log\left(\frac{6K \log_2(K)}{\delta}\right)$, *assuming* $\ell_u^*$ *exists, we obtain*

$$\mathbb{P}\left(\ell_u < \ell_u^*\right) \leq \exp\left(-\frac{T_m}{2048 H_2 \log_2(K)}\right).$$

*Proof.* Recall that $\ell_u^*$ to be the largest stage $\ell$ such that $\Delta_u \leq \frac{1}{2}\Delta_{\frac{K}{4} \cdot 2^{-\ell+1}}$. We denote by $\mathcal{A}_\ell$ the set of arms at stage $\ell$. In the event arm $u$ does not survive until stage $\ell_u^*$. We have

$$\mathbb{P}\left(\ell_u < \ell_u^*\right)$$

$$= \mathbb{P}\left(\exists \ell \in [\ell_u^* - 1] : u \notin \mathcal{A}_{\ell+1}, u \in \mathcal{A}_\ell\right)$$

$$\leq \sum_{\ell=1}^{\ell_u^*-1} \mathbb{P}\left(u \notin \mathcal{A}_{\ell+1}, u \in \mathcal{A}_\ell\right) \qquad \text{(use union bound)}$$

$$\leq \sum_{\ell=1}^{\ell_u^*-1} \sum_{\substack{a_\ell \subseteq [K]: \\ u \in a_\ell, \\ |a_\ell|=K \cdot 2^{-\ell+1}}} \mathbb{P}\left(u \notin \mathcal{A}_{\ell+1}, u \in \mathcal{A}_\ell, \mathcal{A}_\ell = a_\ell\right) \qquad \text{(use the law of total probability)}$$

$$= \sum_{\ell=1}^{\ell_u^*-1} \sum_{\substack{a_\ell \subseteq [K]: \\ u \in a_\ell, \\ |a_\ell| = K \cdot 2^{-\ell+1}}} \mathbb{P}\left(u \notin \mathcal{A}_{\ell+1}, \, u \in \mathcal{A}_\ell \mid \mathcal{A}_\ell = a_\ell\right) \mathbb{P}\left(\mathcal{A}_\ell = a_\ell\right).$$

By the setup of the FC-DSH, at stage $\ell$, the event arm $u$ is in set $\mathcal{A}_\ell$ but is eliminated next around means that there exists a set $A \subset \mathcal{A}_\ell$ that half the size of $\mathcal{A}_\ell$ such that $\forall i \in A, \, \hat{\mu}_u \leq \hat{\mu}_i$. Formally,

$$\sum_{\ell=1}^{\ell_u^*-1} \sum_{\substack{a_\ell \subseteq [K]: \\ u \in a_\ell, \\ |a_\ell| = K2^{-\ell+1}}} \mathbb{P}\left(u \notin \mathcal{A}_{\ell+1}, \, u \in \mathcal{A}_\ell \mid \mathcal{A}_\ell = a_\ell\right) \mathbb{P}\left(\mathcal{A}_\ell = a_\ell\right)$$

$$= \sum_{\ell=1}^{\ell_u^*-1} \sum_{\substack{a_\ell \subseteq [K]: \\ u \in a_\ell, \\ |a_\ell| = K2^{-\ell+1}}} \mathbb{P}\left(\exists A \subset \mathcal{A}_\ell, \text{ s.t. } |A| = \frac{|\mathcal{A}_\ell|}{2}, \, \forall i \in A, \, \hat{\mu}_u \leq \hat{\mu}_i \mid \mathcal{A}_\ell = a_\ell\right) \mathbb{P}\left(\mathcal{A}_\ell = a_\ell\right)$$

$$= \sum_{\ell=1}^{\ell_u^*-1} \sum_{\substack{a_\ell \subseteq [K]: \\ u \in a_\ell, \\ |a_\ell| = K2^{-\ell+1}}} \mathbb{P}\left(\exists A \subset a_\ell, \text{ s.t. } |A| = \frac{|a_\ell|}{2}, \, \forall i \in A, \, \hat{\mu}_u \leq \hat{\mu}_i \mid \mathcal{A}_\ell = a_\ell\right) \mathbb{P}\left(\mathcal{A}_\ell = a_\ell\right).$$

Let $A \subset a_\ell$ be a set as described in $\mathbb{P}(\cdot)$ above. We denote by $\mathsf{Bot}_j(A)$ a set of arms with $|A| - j$ lowest means in $A$. Formally, $\mathsf{Bot}_j(A)$ satisfies that $\mathsf{Bot}_j(A) \subseteq A, \big|\mathsf{Bot}_j(A)\big| = |A| - j$, and $\forall x \in \mathsf{Bot}_j(A), \, \forall y \in A \backslash \mathsf{Bot}_j(A), \, \mu_x \leq \mu_y$.

We denote $\mu_i(A)$ to be mean of the arm $i$ indexed within the set $A$. Also to make it clear, we denote $\mu_i([K])$ to be mean of the arm $i$ indexed within the whole set $[K]$, i.e., $\mu_i([K]) = \mu_i$

We set $j = \frac{|A|}{2} = \frac{|a_\ell|}{4}$, we obtain the following properties

- $\big|\mathsf{Bot}_j(A)\big| = \frac{|A|}{2} = \frac{|a_\ell|}{4}$.
- $\forall i \in \mathsf{Bot}_j(A), \, \mu_i \leq \mu_{\frac{|A|}{2}}(A) \leq \mu_{\frac{|A|}{2}}([K]) = \mu_{\frac{|a_\ell|}{4}}([K]) \leq \mu_{\frac{|a_\ell|}{4}}$. The first inequality is from the setup of $\mathsf{Bot}_j(A)$. The second inequality uses the fact that if $A \subseteq [K]$, hence $\forall i, \, \mu_i(A) \leq \mu_i([K])$

Within stage $\ell \leq \ell_u^* - 1$ and set $a_\ell \in [K]$ such that $u \in a_\ell$ and $|a_\ell| = K2^{-\ell+1}$, we focus on the probability

$$\mathbb{P}\left(\exists A \subset a_\ell, \text{ s.t. } |A| = \frac{|a_\ell|}{2}, \, \forall i \in A, \, \hat{\mu}_u \leq \hat{\mu}_i \mid \mathcal{A}_\ell = a_\ell\right)$$

$$\leq \mathbb{P}\left(\exists A \subset a_\ell, \text{ s.t. } |A| = \frac{|a_\ell|}{2}, \, \forall i \in \mathsf{Bot}_j(A), \, \hat{\mu}_u \leq \hat{\mu}_i \mid \mathcal{A}_\ell = a_\ell\right) \qquad (\text{use } \mathsf{Bot}_j(A) \subset A)$$

Within the condition $\exists A \subset a_\ell, \text{s.t.} \, |A| = \frac{|a_\ell|}{2}, \, \forall i \in \mathsf{Bot}_j(A)$, we have

$$\hat{\mu}_u \leq \hat{\mu}_i$$

$$\Leftrightarrow \hat{\mu}_u - \mu_u - \hat{\mu}_i + \mu_i \leq -\mu_u + \mu_i$$

$$\Rightarrow \hat{\mu}_u - \mu_u - \hat{\mu}_i + \mu_i \leq -\mu_u + \mu_{\frac{|a_\ell|}{4}} \qquad (\text{use the property } \mu_i \leq \mu_{\frac{|a_\ell|}{4}})$$

$$\Leftrightarrow \hat{\mu}_u - \mu_u - \hat{\mu}_i + \mu_i \leq \Delta_u - \Delta_{\frac{|a_\ell|}{4}}$$

$$\Rightarrow \hat{\mu}_u - \mu_u - \hat{\mu}_i + \mu_i \leq \frac{1}{2}\Delta_{\frac{K}{4}2^{-\ell_u^*+1}} - \Delta_{\frac{|a_\ell|}{4}} \qquad (\text{by the definition } \Delta_u \leq \frac{1}{2}\Delta_{\frac{K}{4}2^{-\ell_u^*+1}})$$

$$\Rightarrow \hat{\mu}_u - \mu_u - \hat{\mu}_i + \mu_i \leq \frac{1}{2}\Delta_{\frac{K}{4}2^{-\ell+1}} - \Delta_{\frac{|a_\ell|}{4}} \qquad (\text{since } \ell < \ell_u^*)$$

$$\Leftrightarrow \hat{\mu}_u - \mu_u - \hat{\mu}_i + \mu_i \leq \frac{1}{2}\Delta_{\frac{|a_\ell|}{4}} - \Delta_{\frac{|a_\ell|}{4}} \qquad (\text{use } |a_\ell| = K2^{-\ell+1})$$

$$\Leftrightarrow \hat{\mu}_u - \mu_u - \hat{\mu}_i + \mu_i \leq -\frac{1}{2}\Delta_{\frac{|a_\ell|}{4}}$$

Thus, we have

$$\mathbb{P}\left(\exists A \subset a_\ell, \text{ s.t. } |A| = \frac{|a_\ell|}{2}, \forall i \in \mathsf{Bot}_j(A), \hat{\mu}_u \leq \hat{\mu}_i \mid \mathcal{A}_\ell = a_\ell\right)$$

$$\leq \mathbb{P}\left(\exists A \subset a_\ell, \text{ s.t. } |A| = \frac{|a_\ell|}{2}, \forall i \in \mathsf{Bot}_j(A), \hat{\mu}_u - \mu_u - \hat{\mu}_i + \mu_i \leq -\frac{1}{2}\Delta_{\lfloor\frac{|a_\ell|}{4}\rfloor} \mid \mathcal{A}_\ell = a_\ell\right)$$

$$\leq \mathbb{P}\left(\exists A \subset a_\ell, \text{ s.t. } |A| = \frac{|a_\ell|}{2}, \forall i \in \mathsf{Bot}_j(A), \left(\hat{\mu}_u - \mu_u \leq -\frac{1}{4}\Delta_{\lfloor\frac{|a_\ell|}{4}\rfloor}\right) \vee \left(\hat{\mu}_i - \mu_i \geq \frac{1}{4}\Delta_{\lfloor\frac{|a_\ell|}{4}\rfloor}\right) \mid \mathcal{A}_\ell = a_\ell\right)$$

$$\leq \mathbb{P}\left(\hat{\mu}_u - \mu_u \leq -\frac{1}{4}\Delta_{\lfloor\frac{|a_\ell|}{4}\rfloor} \mid \mathcal{A}_\ell = a_\ell\right)$$

$$\quad + \mathbb{P}\left(\exists A \subset a_\ell, \text{ s.t. } |A| = \frac{|a_\ell|}{2}, \forall i \in \mathsf{Bot}_j(A), \hat{\mu}_i - \mu_i \geq \frac{1}{4}\Delta_{\lfloor\frac{|a_\ell|}{4}\rfloor} \mid \mathcal{A}_\ell = a_\ell\right) \qquad \text{(use union bound)}$$

$$\leq \mathbb{P}\left(\hat{\mu}_u - \mu_u \leq -\frac{1}{4}\Delta_{\lfloor\frac{|a_\ell|}{4}\rfloor} \mid \mathcal{A}_\ell = a_\ell\right)$$

$$\quad + \sum_{\substack{A \subset a_\ell: \\ |A| = \frac{|a_\ell|}{2}}} \mathbb{P}\left(\forall i \in \mathsf{Bot}_j(A), \hat{\mu}_i \geq \mu_i + \frac{1}{4}\Delta_{\lfloor\frac{|a_\ell|}{4}\rfloor} \mid \mathcal{A}_\ell = a_\ell\right)$$

$$\leq \exp\left(-\frac{N^{(m,\ell)}}{2}\frac{\Delta^2_{\lfloor\frac{|a_\ell|}{4}\rfloor}}{16}\right) + \sum_{\substack{A \subset a_\ell: \\ |A| = \frac{|a_\ell|}{2}}} \exp\left(-\left|\mathsf{Bot}_j(A)\right|\frac{N^{(m,\ell)}}{2}\frac{\Delta^2_{\lfloor\frac{|a_\ell|}{4}\rfloor}}{16}\right) \qquad \text{(use Hoeffding's inequality)}$$

$$= \exp\left(-\frac{N^{(m,\ell)}}{2}\frac{\Delta^2_{\lfloor\frac{|a_\ell|}{4}\rfloor}}{16}\right) + \sum_{\substack{A \subset a_\ell: \\ |A| = \frac{|a_\ell|}{2}}} \exp\left(-\frac{|a_\ell|}{4}\frac{N^{(m,\ell)}}{2}\frac{\Delta^2_{\lfloor\frac{|a_\ell|}{4}\rfloor}}{16}\right) \qquad \text{(use property } \left|\mathsf{Bot}_j(A)\right| = \frac{|a_\ell|}{4}\text{)}$$

$$= \exp\left(-\frac{N^{(m,\ell)}}{2}\frac{\Delta^2_{\lfloor\frac{|a_\ell|}{4}\rfloor}}{16}\right) + \binom{|a_\ell|}{\frac{|a_\ell|}{2}}\exp\left(-\frac{|a_\ell|}{4}\frac{N^{(m,\ell)}}{2}\frac{\Delta^2_{\lfloor\frac{|a_\ell|}{4}\rfloor}}{16}\right)$$

$$\leq \exp\left(-\frac{N^{(m,\ell)}}{2}\frac{\Delta^2_{\lfloor\frac{|a_\ell|}{4}\rfloor}}{16}\right) + (2e)^{\frac{|a_\ell|}{2}}\exp\left(-\frac{|a_\ell|}{4}\frac{N^{(m,\ell)}}{2}\frac{\Delta^2_{\lfloor\frac{|a_\ell|}{4}\rfloor}}{16}\right) \qquad \text{(use Lemma B.9)}$$

$$= \exp\left(-\frac{N^{(m,\ell)}}{2}\frac{\Delta^2_{\lfloor\frac{|a_\ell|}{4}\rfloor}}{16}\right) + \exp\left(-\frac{|a_\ell|}{4}\frac{N^{(m,\ell)}}{2}\frac{\Delta^2_{\lfloor\frac{|a_\ell|}{4}\rfloor}}{16} + \frac{|a_\ell|}{2}\log(2e)\right).$$

We apply Lemma B.8 that shows for all phase $m$ such that $T_m \geq T_\delta$, $N^{(m,\ell)} \geq \frac{256}{\Delta^2_{\lfloor\frac{|a_\ell|}{4}\rfloor}}\log(2e)$. This inequality also means that

$$\frac{256}{\Delta^2_{\lfloor\frac{|a_\ell|}{4}\rfloor}}\log(2e) \leq N^{(m,\ell)}$$

$$\Leftrightarrow 8\log(2e) \leq \frac{N^{(m,\ell)}}{2}\frac{\Delta^2_{\lfloor\frac{|a_\ell|}{4}\rfloor}}{16}$$

$$\Leftrightarrow |a_\ell|\log(2e) \leq \frac{|a_\ell|}{8}\frac{N^{(m,\ell)}}{2}\frac{\Delta^2_{\lfloor\frac{|a_\ell|}{4}\rfloor}}{16}$$

$$\Rightarrow \frac{|a_\ell|}{2}\log(2e) \leq \frac{|a_\ell|}{8}\frac{N^{(m,\ell)}}{2}\frac{\Delta^2_{\lfloor\frac{|a_\ell|}{4}\rfloor}}{16}$$

$$\Rightarrow -\frac{|a_\ell|}{4}\frac{N^{(m,\ell)}}{2}\frac{\Delta^2_{\lfloor\frac{|a_\ell|}{4}\rfloor}}{16} + \frac{|a_\ell|}{2}\log(2e) \le -\frac{|a_\ell|}{8}\frac{N^{(m,\ell)}}{2}\frac{\Delta^2_{\lfloor\frac{|a_\ell|}{4}\rfloor}}{16}.$$

Thus,

$$\exp\left(-\frac{N^{(m,\ell)}}{2}\frac{\Delta^2_{\lfloor\frac{|a_\ell|}{4}\rfloor}}{16}\right) + \exp\left(-\frac{|a_\ell|}{4}\frac{N^{(m,\ell)}}{2}\frac{\Delta^2_{\lfloor\frac{|a_\ell|}{4}\rfloor}}{16} + \frac{|a_\ell|}{2}\log(2e)\right)$$

$$\le \exp\left(-\frac{N^{(m,\ell)}}{2}\frac{\Delta^2_{\lfloor\frac{|a_\ell|}{4}\rfloor}}{16}\right) + \exp\left(-\frac{|a_\ell|}{8}\frac{N^{(m,\ell)}}{2}\frac{\Delta^2_{\lfloor\frac{|a_\ell|}{4}\rfloor}}{16}\right)$$

$$\le \exp\left(-\frac{N^{(m,\ell)}}{2}\frac{\Delta^2_{\lfloor\frac{|a_\ell|}{4}\rfloor}}{16}\right) + \exp\left(-\frac{1}{8}\frac{N^{(m,\ell)}}{2}\frac{\Delta^2_{\lfloor\frac{|a_\ell|}{4}\rfloor}}{16}\right) \qquad \left(\text{use }|a_\ell| \ge 1\right)$$

$$\le 2\exp\left(-\frac{\Delta^2_{\lfloor\frac{|a_\ell|}{4}\rfloor}N^{(m,\ell)}}{256}\right)$$

$$= 2\exp\left(-\frac{\Delta^2_{\frac{K}{4}2^{-\ell+1}}N^{(m,\ell)}}{256}\right) \qquad \left(\text{use }|a_\ell| = K2^{-\ell+1}\right)$$

By the definition $N^{(m,\ell)} = \frac{T_m}{2^{-\ell+1}K\log_2(K)}$, we further have

$$\exp\left(-\frac{\Delta^2_{\frac{K}{4}2^{-\ell+1}}}{256}\cdot\frac{T_m}{2^{-\ell+1}K\log_2(K)}\right)$$

$$= 2\exp\left(-\frac{\Delta^2_{\frac{K}{4}2^{-\ell+1}}}{\frac{K}{4}2^{-\ell+1}}\cdot\frac{T_m}{1024\log_2(K)}\right)$$

$$\le 2\exp\left(-\frac{1}{H_2}\cdot\frac{T_m}{1024\log_2(K)}\right) \qquad \left(\text{by the definition }H_2 = \max_i i\Delta_i^{-2}\right)$$

$$= 2\exp\left(-\frac{T_m}{1024H_2\log_2(K)}\right).$$

To summarize, we have obtained the following probability bound

$$\mathbb{P}\left(\exists A \subset a_\ell,\text{ s.t. }|A| = \frac{|a_\ell|}{2},\ \forall i \in A,\ \hat\mu_u \le \hat\mu_i \mid \mathcal{A}_\ell = a_\ell\right) \le 2\exp\left(-\frac{T_m}{1024H_2\log_2(K)}\right).$$

Therefore, from the beginning derivation, we obtain the following probability bound

$$\mathbb{P}\left(\ell_u < \ell_u^*\right)$$

$$= \sum_{\ell=1}^{\ell_u^*-1}\sum_{\substack{a_\ell \subseteq [K]:\\ u\in a_\ell,\\ |a_\ell|=K\cdot2^{-\ell+1}}} \mathbb{P}\left(u \notin \mathcal{A}_{\ell+1},\ u \in \mathcal{A}_\ell \mid \mathcal{A}_\ell = a_\ell\right)\mathbb{P}\left(\mathcal{A}_\ell = a_\ell\right)$$

$$= \sum_{\ell=1}^{\ell_u^*-1}\sum_{\substack{a_\ell \subseteq [K]:\\ u\in a_\ell,\\ |a_\ell|=K\cdot2^{-\ell+1}}} 2\exp\left(-\frac{T_m}{1024H_2\log_2(K)}\right)\mathbb{P}\left(\mathcal{A}_\ell = a_\ell\right)$$

$$= \sum_{\ell=1}^{\ell_u^*-1} 2\exp\left(-\frac{T_m}{1024H_2\log_2(K)}\right)$$

$$\leq 2\log_2(K)\exp\left(-\frac{T_m}{1024H_2\log_2(K)}\right) \qquad \text{(use } \ell_u^* \leq \log_2(K))$$

$$= \exp\left(-\frac{T_m}{1024H_2\log_2(K)} + \log(2\log_2(K))\right).$$

The condition $T_m \geq T_\delta = 4096H_2\log_2(K)\log\left(\frac{6K\log_2(K)}{\delta}\right)$ implies that

$$T_m \geq 4096H_2\log_2(K)\log\left(\frac{6K\log_2(K)}{\delta}\right)$$

$$\Rightarrow T_m \geq 2048H_2\log_2(K)\log\left(2\log_2(K)\right) \qquad \text{(use } \delta \leq 1)$$

$$\Rightarrow \log\left(2\log_2(K)\right) \leq \frac{T_m}{2048H_2\log_2(K)}$$

$$\Rightarrow -\frac{T_m}{1024H_2\log_2(K)} + \log\left(2\log_2(K)\right) \leq -\frac{T_m}{2048H_2\log_2(K)}.$$

Therefore,

$$\exp\left(-\frac{T_m}{1024H_2\log_2(K)} + \log(2\log_2(K))\right)$$

$$\leq \exp\left(-\frac{T_m}{2048H_2\log_2(K)}\right),$$

which concludes the proof of the Lemma.

$\square$

**Lemma B.5.** *For all phase $m$ such that $T_m \geq T_\delta = 4096H_2\log_2(K)\log\left(\frac{6K\log_2(K)}{\delta}\right)$, the probability that FC-DSH fails to output arm $1$ at the end of phase $m$ satisfies*

$$\mathbb{P}\left(J_m \neq 1\right) \leq \exp\left(-\frac{T_m}{512H_2\log_2(K)}\right).$$

*Proof.* While one can directly use the proof of Karnin et al. (2013), we present here an alternative proof that has an pedagogical value – this proof shows a more fine-grained control of events that reveals that the bottleneck of the guarantee is the bad behavior of the best arm rather than the bad behavior of the bad arms.

We denote by $\mathcal{A}_\ell$ the set of surviving arms at stage $\ell$. We denote by $\ell_1$ the stage at which arm $1$ is eliminated. The event that $J_m \neq 1$ implies that arm $1$ does not survive until the last stage $L$, i.e., $\ell_1 < L$. We have

$$\mathbb{P}\left(\ell_1 < L\right)$$

$$= \mathbb{P}\left(\exists \ell \leq L-1 : 1 \notin \mathcal{A}_{\ell+1}, 1 \in \mathcal{A}_\ell\right)$$

$$\leq \sum_{\ell=1}^{L-1} \mathbb{P}\left(1 \notin \mathcal{A}_{\ell+1}, 1 \in \mathcal{A}_\ell\right) \qquad \text{(use union bound)}$$

$$\leq \sum_{\ell=1}^{L-1} \sum_{\substack{a_\ell \subseteq [K]: \\ 1 \in a_\ell, \\ |a_\ell| = K \cdot 2^{-\ell+1}}} \mathbb{P}\left(1 \notin \mathcal{A}_{\ell+1}, 1 \in \mathcal{A}_\ell, \mathcal{A}_\ell = a_\ell\right) \qquad \text{(use the law of total probability)}$$

$$= \sum_{\ell=1}^{L-1} \sum_{\substack{a_\ell \subseteq [K]: \\ 1 \in a_\ell, \\ |a_\ell| = K \cdot 2^{-\ell+1}}} \mathbb{P}\left(1 \notin \mathcal{A}_{\ell+1}, 1 \in \mathcal{A}_\ell \mid \mathcal{A}_\ell = a_\ell\right) \mathbb{P}\left(\mathcal{A}_\ell = a_\ell\right).$$

By the setup of the FC-DSH, at stage $\ell$, the event where arm $1$ is in set $\mathcal{A}_\ell$ but is eliminated next around means that there

exists a set $A \subset \mathcal{A}_\ell$ that half the size of $\mathcal{A}_\ell$ such that $\forall i \in A$, $\hat{\mu}_1 \leq \hat{\mu}_i$. Formally,

$$\sum_{\ell=1}^{L-1} \sum_{\substack{a_\ell \subseteq [K]: \\ 1 \in a_\ell, \\ |a_\ell| = K2^{-\ell+1}}} \mathbb{P}\left(u \notin \mathcal{A}_{\ell+1}, 1 \in \mathcal{A}_\ell \mid \mathcal{A}_\ell = a_\ell\right) \mathbb{P}\left(\mathcal{A}_\ell = a_\ell\right)$$

$$= \sum_{\ell=1}^{L-1} \sum_{\substack{a_\ell \subseteq [K]: \\ 1 \in a_\ell, \\ |a_\ell| = K2^{-\ell+1}}} \mathbb{P}\left(\exists A \subset \mathcal{A}_\ell, \text{ s.t. } |A| = \frac{|\mathcal{A}_\ell|}{2}, \forall i \in A, \hat{\mu}_1 \leq \hat{\mu}_i \mid \mathcal{A}_\ell = a_\ell\right) \mathbb{P}\left(\mathcal{A}_\ell = a_\ell\right)$$

$$= \sum_{\ell=1}^{L-1} \sum_{\substack{a_\ell \subseteq [K]: \\ 1 \in a_\ell, \\ |a_\ell| = K2^{-\ell+1}}} \mathbb{P}\left(\exists A \subset a_\ell, \text{ s.t. } |A| = \frac{|a_\ell|}{2}, \forall i \in A, \hat{\mu}_1 \leq \hat{\mu}_i \mid \mathcal{A}_\ell = a_\ell\right) \mathbb{P}\left(\mathcal{A}_\ell = a_\ell\right).$$

Let $A \subset a_\ell$ be a set as described inside $\mathbb{P}(\cdot)$ above. We denote by $\mathsf{Bot}_j(A)$ be a set of arms with $|A| - j$ lowest means in $A$. Formally, $\mathsf{Bot}_j(A)$ satisfies that $\mathsf{Bot}_j(A) \subseteq A$, $\left|\mathsf{Bot}_j(A)\right| = |A| - j$, and $\forall x \in \mathsf{Bot}_j(A)$, $\forall y \in A \backslash \mathsf{Bot}_j(A)$, $\mu_x \leq \mu_y$.

We denote $\mu_i(A)$ to be mean of the arm $i$ indexed within the set $A$. Also to make it clear, we denote $\mu_i([K])$ to be mean of the arm $i$ indexed within the whole set $[K]$, i.e., $\mu_i([K]) = \mu_i$.

We set $j = \frac{|A|}{2} = \frac{|a_\ell|}{4}$, we obtain the following properties

- $\left|\mathsf{Bot}_j(A)\right| = \frac{|A|}{2} = \frac{|a_\ell|}{4}$.
- $\forall i \in \mathsf{Bot}_j(A)$, $\mu_i \leq \mu_{\lfloor \frac{|A|}{2} \rfloor}(A) \leq \mu_{\lfloor \frac{|A|}{2} \rfloor}([K]) = \mu_{\lfloor \frac{|a_\ell|}{4} \rfloor}([K]) \leq \mu_{\lfloor \frac{|a_\ell|}{4} \rfloor}$. The first inequality is from the definition of $\mathsf{Bot}_j(A)$. The second inequality uses the fact that if $A \subseteq [K]$, hence $\forall i$, $\mu_i(A) \leq \mu_i([K])$

Within stage $\ell \leq L - 1$ and set $a_\ell \subseteq [K]$, $1 \in a_\ell$, $|a_\ell| = K2^{-\ell+1}$, we focus on the probability

$$\mathbb{P}\left(\exists A \subset a_\ell, \text{ s.t. } |A| = \frac{|a_\ell|}{2}, \forall i \in A, \hat{\mu}_1 \leq \hat{\mu}_i \mid \mathcal{A}_\ell = a_\ell\right)$$

$$\leq \mathbb{P}\left(\exists A \subset a_\ell, \text{ s.t. } |A| = \frac{|a_\ell|}{2}, \forall i \in \mathsf{Bot}_j(A), \hat{\mu}_1 \leq \hat{\mu}_i \mid \mathcal{A}_\ell = a_\ell\right)$$

$$= \mathbb{P}\left(\exists A \subset a_\ell, \text{ s.t. } |A| = \frac{|a_\ell|}{2}, \forall i \in \mathsf{Bot}_j(A), \hat{\mu}_1 - \mu_1 - \hat{\mu}_i + \mu_i \leq -\mu_1 + \mu_i \mid \mathcal{A}_\ell = a_\ell\right)$$

$$\leq \mathbb{P}\left(\exists A \subset a_\ell, \text{ s.t. } |A| = \frac{|a_\ell|}{2}, \forall i \in \mathsf{Bot}_j(A), \hat{\mu}_1 - \mu_1 - \hat{\mu}_i + \mu_i \leq -\mu_1 + \mu_{\lfloor \frac{|a_\ell|}{4} \rfloor} \mid \mathcal{A}_\ell = a_\ell\right)$$

$$\text{(use the property } \mu_i \leq \mu_{\lfloor \frac{|a_\ell|}{4} \rfloor})$$

$$= \mathbb{P}\left(\exists A \subset a_\ell, \text{ s.t. } |A| = \frac{|a_\ell|}{2}, \forall i \in \mathsf{Bot}_j(A), \hat{\mu}_1 - \mu_1 - \hat{\mu}_i + \mu_i \leq -\Delta_{\lfloor \frac{|a_\ell|}{4} \rfloor} \mid \mathcal{A}_\ell = a_\ell\right)$$

$$\leq \mathbb{P}\left(\exists A \subset a_\ell, \text{ s.t. } |A| = \frac{|a_\ell|}{2}, \forall i \in \mathsf{Bot}_j(A), \left(\hat{\mu}_1 - \mu_1 \leq -\frac{1}{2}\Delta_{\lfloor \frac{|a_\ell|}{4} \rfloor}\right) \vee \left(\hat{\mu}_i - \mu_i \geq \frac{1}{2}\Delta_{\lfloor \frac{|a_\ell|}{4} \rfloor}\right) \mid \mathcal{A}_\ell = a_\ell\right)$$

$$\leq \mathbb{P}\left(\hat{\mu}_1 - \mu_1 \leq -\frac{1}{2}\Delta_{\lfloor \frac{|a_\ell|}{4} \rfloor} \mid \mathcal{A}_\ell = a_\ell\right)$$

$$+ \mathbb{P}\left(\exists A \subset a_\ell, \text{ s.t. } |A| = \frac{|a_\ell|}{2}, \forall i \in \mathsf{Bot}_j(A), \hat{\mu}_i - \mu_i \geq \frac{1}{2}\Delta_{\lfloor \frac{|a_\ell|}{4} \rfloor} \mid \mathcal{A}_\ell = a_\ell\right) \qquad \text{(use union bound)}$$

$$\leq \mathbb{P}\left(\hat{\mu}_1 - \mu_1 \leq -\frac{1}{2}\Delta_{\lfloor \frac{|a_\ell|}{4} \rfloor} \mid \mathcal{A}_\ell = a_\ell\right)$$

$$+ \sum_{\substack{A \subset a_\ell: \\ |A| = \frac{|a_\ell|}{2}}} \mathbb{P}\left(\forall i \in \mathsf{Bot}_j(A), \hat{\mu}_i \geq \mu_i + \frac{1}{2}\Delta_{\lfloor \frac{|a_\ell|}{4} \rfloor} \mid \mathcal{A}_\ell = a_\ell\right)$$

$$\leq \exp\left(-\frac{N^{(m,\ell)}}{2}\frac{\Delta^2_{\lfloor a_\ell \rfloor}}{4}\right) + \sum_{\substack{A \subset a_\ell: \\ |A| = \frac{|a_\ell|}{2}}} \exp\left(-\left|\mathsf{Bot}_j(A)\right|\frac{N^{(m,\ell)}}{2}\frac{\Delta^2_{\lfloor a_\ell \rfloor}}{4}\right) \qquad \text{(use Hoeffding's inequality)}$$

$$= \exp\left(-\frac{N^{(m,\ell)}}{2}\frac{\Delta^2_{\lfloor a_\ell \rfloor}}{4}\right) + \sum_{\substack{A \subset a_\ell: \\ |A| = \frac{|a_\ell|}{2}}} \exp\left(-\frac{|a_\ell|}{4}\frac{N^{(m,\ell)}}{2}\frac{\Delta^2_{\lfloor a_\ell \rfloor}}{4}\right) \qquad \text{(use property } \left|\mathsf{Bot}_j(A)\right| = \frac{|a_\ell|}{4})$$

$$= \exp\left(-\frac{N^{(m,\ell)}}{2}\frac{\Delta^2_{\lfloor a_\ell \rfloor}}{4}\right) + \binom{|a_\ell|}{\frac{|a_\ell|}{2}}\exp\left(-\frac{|a_\ell|}{4}\frac{N^{(m,\ell)}}{2}\frac{\Delta^2_{\lfloor a_\ell \rfloor}}{4}\right)$$

$$\leq \exp\left(-\frac{N^{(m,\ell)}}{2}\frac{\Delta^2_{\lfloor a_\ell \rfloor}}{4}\right) + (2e)^{\frac{|a_\ell|}{2}}\exp\left(-\frac{|a_\ell|}{4}\frac{N^{(m,\ell)}}{2}\frac{\Delta^2_{\lfloor a_\ell \rfloor}}{4}\right) \qquad \text{(use Lemma B.9)}$$

$$= \exp\left(-\frac{N^{(m,\ell)}}{2}\frac{\Delta^2_{\lfloor a_\ell \rfloor}}{4}\right) + \exp\left(-\frac{|a_\ell|}{4}\frac{N^{(m,\ell)}}{2}\frac{\Delta^2_{\lfloor a_\ell \rfloor}}{4} + \frac{|a_\ell|}{2}\log(2e)\right).$$

We apply Lemma B.8 that shows for all phase $m$ such that $T_m \geq T_\delta$, $N^{(m,\ell)} \geq \frac{256}{\Delta^2_{\lfloor a_\ell \rfloor}}\log(2e)$. This inequality also means that

$$\frac{256}{\Delta^2_{\lfloor a_\ell \rfloor}}\log(2e) \leq N^{(m,\ell)}$$

$$\Leftrightarrow 32\log(2e) \leq \frac{N^{(m,\ell)}}{2}\frac{\Delta^2_{\lfloor a_\ell \rfloor}}{4}$$

$$\Leftrightarrow 4|a_\ell|\log(2e) \leq \frac{|a_\ell|}{8}\frac{N^{(m,\ell)}}{2}\frac{\Delta^2_{\lfloor a_\ell \rfloor}}{4}$$

$$\Rightarrow \frac{|a_\ell|}{2}\log(2e) \leq \frac{|a_\ell|}{8}\frac{N^{(m,\ell)}}{2}\frac{\Delta^2_{\lfloor a_\ell \rfloor}}{4}$$

$$\Rightarrow -\frac{|a_\ell|}{4}\frac{N^{(m,\ell)}}{2}\frac{\Delta^2_{\lfloor a_\ell \rfloor}}{4} + \frac{|a_\ell|}{2}\log(2e) \leq -\frac{|a_\ell|}{8}\frac{N^{(m,\ell)}}{2}\frac{\Delta^2_{\lfloor a_\ell \rfloor}}{4}.$$

Thus,

$$\exp\left(-\frac{N^{(m,\ell)}}{2}\frac{\Delta^2_{\lfloor a_\ell \rfloor}}{4}\right) + \exp\left(-\frac{|a_\ell|}{4}\frac{N^{(m,\ell)}}{2}\frac{\Delta^2_{\lfloor a_\ell \rfloor}}{4} + \frac{|a_\ell|}{2}\log(2e)\right)$$

$$\leq \exp\left(-\frac{N^{(m,\ell)}}{2}\frac{\Delta^2_{\lfloor a_\ell \rfloor}}{4}\right) + \exp\left(-\frac{|a_\ell|}{8}\frac{N^{(m,\ell)}}{2}\frac{\Delta^2_{\lfloor a_\ell \rfloor}}{4}\right)$$

$$\leq \exp\left(-\frac{N^{(m,\ell)}}{2}\frac{\Delta^2_{\lfloor a_\ell \rfloor}}{4}\right) + \exp\left(-\frac{1}{8}\frac{N^{(m,\ell)}}{2}\frac{\Delta^2_{\lfloor a_\ell \rfloor}}{4}\right) \qquad \left(\text{use } |a_\ell| \geq 1\right)$$

$$\leq 2\exp\left(-\frac{\Delta^2_{\lfloor a_\ell \rfloor}N^{(m,\ell)}}{64}\right)$$

$$= 2\exp\left(-\frac{\Delta_{\frac{K}{4}2^{-\ell+1}}^2 N^{(m,\ell)}}{64}\right) \qquad \text{(use } |a_\ell| = K2^{-\ell+1})$$

By the definition $N^{(m,\ell)} = \frac{T_m}{2^{-\ell+1}K\log_2(K)}$, we further have

$$\exp\left(-\frac{\Delta_{\frac{K}{4}2^{-\ell+1}}^2}{64} \cdot \frac{T_m}{2^{-\ell+1}K\log_2(K)}\right)$$

$$= 2\exp\left(-\frac{\Delta_{\frac{K}{4}2^{-\ell+1}}^2}{\frac{K}{4}2^{-\ell+1}} \cdot \frac{T_m}{256\log_2(K)}\right)$$

$$\leq 2\exp\left(-\frac{1}{H_2} \cdot \frac{T_m}{256\log_2(K)}\right) \qquad \text{(by the definition } H_2 = \max_i i\Delta_i^{-2})$$

$$= 2\exp\left(-\frac{T_m}{256 H_2 \log_2(K)}\right).$$

To summarize, we have obtained the following probability bound

$$\mathbb{P}\left(\exists A \subset a_\ell, \text{ s.t. } |A| = \frac{|a_\ell|}{2}, \forall i \in A, \hat{\mu}_1 \leq \hat{\mu}_i \mid \mathcal{A}_\ell = a_\ell\right) \leq 2\exp\left(-\frac{T_m}{256 H_2 \log_2(K)}\right).$$

Therefore, from the beginning derivation, we obtain the following probability bound

$$\mathbb{P}\left(\ell_u < L\right)$$

$$= \sum_{\ell=1}^{L-1} \sum_{\substack{a_\ell \subseteq [K]: \\ 1 \in a_\ell, \\ |a_\ell| = K \cdot 2^{-\ell+1}}} \mathbb{P}\left(u \notin \mathcal{A}_{\ell+1}, u \in \mathcal{A}_\ell \mid \mathcal{A}_\ell = a_\ell\right) \mathbb{P}\left(\mathcal{A}_\ell = a_\ell\right)$$

$$= \sum_{\ell=1}^{L-1} \sum_{\substack{a_\ell \subseteq [K]: \\ 1 \in a_\ell, \\ |a_\ell| = K \cdot 2^{-\ell+1}}} 2\exp\left(-\frac{T_m}{256 H_2 \log_2(K)}\right) \mathbb{P}\left(\mathcal{A}_\ell = a_\ell\right)$$

$$= \sum_{\ell=1}^{L-1} 2\exp\left(-\frac{T_m}{256 H_2 \log_2(K)}\right)$$

$$\leq 2\log_2(K)\exp\left(-\frac{T_m}{256 H_2 \log_2(K)}\right) \qquad \text{(use } L = \log_2(K))$$

$$= \exp\left(-\frac{T_m}{256 H_2 \log_2(K)} + \log(2\log_2(K))\right).$$

The condition $T_m \geq T_\delta = 4096 H_2 \log_2(K) \log\left(\frac{6K\log_2(K)}{\delta}\right)$ implies that

$$T_m \geq 4096 H_2 \log_2(K) \log\left(\frac{6K\log_2(K)}{\delta}\right)$$

$$\Rightarrow T_m \geq 2048 H_2 \log_2(K) \log\left(2\log_2(K)\right) \qquad \text{(use } \delta \leq 1)$$

$$\Rightarrow T_m \geq 512 H_2 \log_2(K) \log\left(2\log_2(K)\right)$$

$$\Rightarrow \log\left(2\log_2(K)\right) \leq \frac{T_m}{512 H_2 \log_2(K)}$$

$$\Rightarrow -\frac{T_m}{256 H_2 \log_2(K)} + \log\left(2\log_2(K)\right) \leq -\frac{T_m}{512 H_2 \log_2(K)}.$$

Therefore,

$$\exp\left(-\frac{T_m}{256 H_2 \log_2(K)} + \log(2\log_2(K))\right)$$

$$\leq \exp\left(-\frac{T_m}{512H_2\log_2(K)}\right),$$

which concludes the proof of the Lemma.

$\square$

**Lemma B.6.** *Let $u \neq 1$ be any non-optimal that satisfies $\Delta_u > \frac{1}{2}\Delta_{\frac{K}{4}}$. For all phase $m$ such that $T_m \geq T_\delta = 4096H_2\log_2(K)\log\left(\frac{6K\log_2(K)}{\delta}\right)$, the number of samples used at stage $\ell_u$ when arm $u$ is eliminated, satisfies*

$$N^{(m,\ell_u)} \geq \frac{32}{\Delta_u^2}\log\left(\frac{6K\log_2(K)m^2}{\delta}\right)$$

*Proof.* We start from the condition $T_m \geq 4096H_2\log_2(K)\log\left(\frac{6K\log_2(K)}{\delta}\right)$ and obtain the inequality as follows

$$T_m \geq 4096H_2\log_2(K)\log\left(\frac{6K\log_2(K)m^2}{\delta}\right)$$

$$\Rightarrow T_m \geq 1024\frac{\frac{K}{4}}{\Delta_{\frac{K}{4}}^2}\log_2(K)\log\left(\frac{6K\log_2(K)m^2}{\delta}\right) \qquad \text{(by the definition } H_2 = \max_{i\geq 2}i\Delta_i^{-2})$$

$$\Rightarrow T_m \geq 1024\frac{\frac{K}{4}2^{-\ell_u}}{\Delta_{\frac{K}{4}}^2}\log_2(K)\log\left(\frac{6K\log_2(K)m^2}{\delta}\right) \qquad \text{(use } 1 \geq 2^{-\ell_u})$$

$$\Leftrightarrow \frac{T_m}{K2^{-\ell_u+1}\log_2(K)} \geq \frac{128}{\Delta_{\frac{K}{4}}^2}\log\left(\frac{6K\log_2(K)m^2}{\delta}\right)$$

$$\Leftrightarrow N^{(m,\ell_u)} \geq \frac{128}{\Delta_{\frac{K}{4}}^2}\log\left(\frac{6K\log_2(K)m^2}{\delta}\right) \qquad \text{(by the definition of } N^{(m,\ell)})$$

$$\Rightarrow N^{(m,\ell_u)} \geq \frac{128}{4\Delta_u^2}\log\left(\frac{6K\log_2(K)m^2}{\delta}\right) \qquad \text{(use } \Delta_u > \frac{1}{2}\Delta_{\frac{K}{4}})$$

$$\Leftrightarrow N^{(m,\ell_u)} \geq \frac{32}{\Delta_u^2}\log\left(\frac{6K\log_2(K)m^2}{\delta}\right).$$

$\square$

**Lemma B.7.** *Let $u \neq 1$ be any non-optimal arm. For all phase $m$ such that $T_m \geq T_\delta = 4096H_2\log_2(K)\log\left(\frac{6K\log_2(K)}{\delta}\right)$, assuming $\ell_u^*$ exists, the number of samples used at stage $\ell_u^*$ satisfies*

$$N^{(m,\ell_u^*)} \geq \frac{32}{\Delta_u^2}\log\left(\frac{6K\log_2(K)m^2}{\delta}\right)$$

*Proof.* Recall that $\ell_u^*$ is defined to be the largest stage $\ell$ such that $\Delta_u \leq \frac{1}{2}\Delta_{\frac{K}{4}\cdot2^{-\ell+1}}$. For any arm $u \in [2,K]$, since it survives until stage $\ell_u^*$, by the definition, it satisfies $\frac{1}{2}\Delta_{\frac{K}{4}\cdot2^{-\ell_u^*}} \leq \Delta_i \leq \frac{1}{2}\Delta_{\frac{K}{4}\cdot2^{-\ell_u^*+1}}$.

We start from the condition $T_m \geq 4096H_2\log_2(K)\log\left(\frac{6K\log_2(K)}{\delta}\right)$ and obtain the inequality as follows

$$T_m \geq 4096H_2\log_2(K)\log\left(\frac{6K\log_2(K)m^2}{\delta}\right)$$

$$\Rightarrow T_m \geq 1024\frac{\frac{K}{4}2^{-\ell_u^*}}{\Delta_{\frac{K}{4}\cdot2^{-\ell_u^*}}^2}\log_2(K)\log\left(\frac{6K\log_2(K)m^2}{\delta}\right) \qquad \text{(by the definition } H_2 = \max_{i\geq 2}i\Delta_i^{-2})$$

$$\Leftrightarrow \frac{T_m}{K2^{-\ell_u^*+1} \log_2(K)} \geq \frac{128}{\Delta_{\frac{K}{4} \cdot 2^{-\ell_u^*}}^2} \log \left( \frac{6K \log_2(K)m^2}{\delta} \right)$$

$$\Leftrightarrow N^{(m,\ell_u^*)} \geq \frac{128}{\Delta_{\frac{K}{4} \cdot 2^{-\ell_u^*}}^2} \log \left( \frac{6K \log_2(K)m^2}{\delta} \right) \qquad \text{(by the definition of } N^{(m,\ell)})$$

$$\Rightarrow N^{(m,\ell_u^*)} \geq \frac{128}{4\Delta_u^2} \log \left( \frac{6K \log_2(K)m^2}{\delta} \right) \qquad \text{(use } \Delta_u \geq \tfrac{1}{2}\Delta_{\frac{K}{4} \cdot 2^{-\ell_u^*}})$$

$$\Leftrightarrow N^{(m,\ell_u^*)} \geq \frac{32}{\Delta_u^2} \log \left( \frac{6K \log_2(K)m^2}{\delta} \right).$$

$\square$

**Lemma B.8.** *For all phase $m$ such that $T_m \geq T_\delta = 4096H_2 \log_2(K) \log \left( \frac{6K \log_2(K)}{\delta} \right)$, the number of samples used at stage $\ell$ where $|a_\ell| = K2^{-\ell+1}$, satisfies*

$$N^{(m,\ell)} \geq \frac{256}{\Delta_{\frac{|a_\ell|}{4}}^2} \log \left( 2e \right).$$

*Proof.* We start from the condition $T_m \geq 4096H_2 \log_2(K) \log \left( \frac{6K \log_2(K)}{\delta} \right)$ and obtain the inequality as follows

$$T_m \geq 4096H_2 \log_2(K) \log \left( \frac{6K \log_2(K)}{\delta} \right)$$

$$\Rightarrow T_m \geq 1024 H_2 \log_2(K) \log \left( 2e \right) \qquad \text{(since } \delta \leq 1)$$

$$\Rightarrow T_m \geq 1024 \frac{\frac{K}{4}2^{-\ell+1}}{\Delta_{\frac{K}{4}2^{-\ell+1}}^2} \log_2(K) \log \left( 2e \right) \qquad \text{(by the definition } H_2 = \max_{i\geq 2} i\Delta_i^{-2})$$

$$\Leftrightarrow \frac{T_m}{K2^{-\ell+1} \log_2(K)} \geq \frac{256}{\Delta_{\frac{K}{4}2^{-\ell+1}}^2} \log \left( 2e \right)$$

$$\Leftrightarrow N^{(m,\ell)} \geq \frac{256}{\Delta_{\frac{K}{4}2^{-\ell+1}}^2} \log \left( 2e \right) \qquad \text{(by the definition of } N^{(m,\ell)})$$

$$\Leftrightarrow N^{(m,\ell)} \geq \frac{256}{\Delta_{\frac{|a_\ell|}{4}}^2} \log \left( 2e \right). \qquad (|a_\ell| = K2^{-\ell+1})$$

$\square$

**Lemma B.9** (Stirling's formula (Das, 2016))**.** *Let $k, d$ be two positive integers such that $1 \leq k \leq n$, then,*

$$\left( \frac{n}{k} \right)^k \leq \binom{n}{k} \leq \left( \frac{en}{k} \right)^k.$$

## C. BrakeBooster

### C.1. Proof of BrakeBooster's Correctness

*Theorem 4.1* (Correctness). Let an algorithm $\mathcal{A}$ be $\delta_0$-correct and have a sample complexity of $T^*_{\delta_0}(\mathcal{A})$. Suppose we run BrakeBooster (Algorithm 2) denoted by $\mathcal{M}$ with input $\mathcal{A}$, $\delta$, $L_1 = \lceil \frac{4 \log(1 + \frac{2}{\delta})}{\log \frac{1}{4e\delta_0}} \rceil$, $T_1 \geq 1$, and $\delta_0 \leq \frac{1}{(2e)^2}$. Then,

$$\mathbb{P}\left(\tau(\mathcal{M}) < \infty, J(\mathcal{M}) \neq 1\right) \leq \delta.$$

In this proof, acute readers will notice that we often talk about events that happen in stage $(r, c)$ without having a condition that the algorithm has not stopped before. To deal with this without notational overload, we take the model where the algorithm has already been run for all stages without stopping, and the user of the algorithm only reveals what happened already and stops when the stopping condition is met. This way, we can talk about events in any stage without adding conditions on whether the algorithm has stopped or not (and this is valid because the samples are independent between stages).

Define

$$Q_{r,c} := \{\ell \in [L_{r,c}] : \ell\text{-th trial}$$
$$\text{self-terminates and output incorrect arm}\}.$$

Note that

$$\mathbb{P}\left(\tau(\mathcal{M}) < \infty, J(\mathcal{M}) \neq 1\right)$$
$$= \sum_{r=1}^{\infty} \sum_{c=1}^{r} \mathbb{P}\left(\mathcal{E}_1, \mathcal{E}_2\right) \cdot \mathbb{P}\left(J_{r,c} \notin \{0, 1\}\right)$$
$$\leq \sum_{r=1}^{\infty} \sum_{c=1}^{r} \mathbb{P}\left(J_{r,c} \notin \{0, 1\}\right),$$

where, $\mathcal{E}_1, \mathcal{E}_2$ are shorthand for

$$\mathcal{E}_1 := \{\forall u \in [r-1],\ v \in [u],\ \hat{J}_{u,v} = 0\},$$
$$\mathcal{E}_2 := \{\forall w \in [c-1], \hat{J}_{r,w} = 0\}.$$

A stage outputting an incorrect arm index means that

1. More than half of the trials of $\mathcal{A}$ has self-terminated; i.e., $v_0 \leq \lfloor L_{r,c}/2 \rfloor$ and thus $\sum_{i \in [K]} v_i \geq L_{r,c} - \lfloor L_{r,c}/2 \rfloor$.

2. The majority vote of those terminated trials is an incorrect arm index; i.e., $\sum_{i \in \{2,\ldots,K\}} v_i \geq \lceil \frac{1}{2} \sum_{i \in [K]} v_i \rceil$.

Thus, we have

$$|Q_{r,c}| = \sum_{i \in \{2,\ldots,K\}} v_i \geq \left\lceil \frac{1}{2}(L_{r,c} - \lfloor L_{r,c}/2 \rfloor) \right\rceil \geq \left\lceil \frac{L_{r,c}}{4} \right\rceil,$$

which implies that

$$\sum_{r=1}^{\infty} \sum_{c=1}^{r} \mathbb{P}\left(J_{r,c} \notin \{0, 1\}\right) \leq \sum_{r=1}^{\infty} \sum_{c=1}^{r} \mathbb{P}\left(|Q_{r,c}| \geq \left\lceil \frac{L_{r,c}}{4} \right\rceil\right).$$

Algorithm $\mathcal{A}$ being $\delta_0$-correct implies,

$$\mathbb{P}(\text{a trial self-terminates and outputs an incorrect arm})$$
$$\leq \delta_0.$$

Hence by Lemma C.3 in the appendix with $\delta = \delta_0$ whose requirement $\delta_0 \leq \alpha = \frac{1}{4}$ is satisfied by the assumption of the theorem,

$$\sum_{r=1}^{\infty} \sum_{c=1}^{r} \mathbb{P}\left(|Q_{r,c}| \geq \left\lceil \frac{L_{r,c}}{4} \right\rceil\right)$$

$$\leq \sum_{r=1}^{\infty} \sum_{c=1}^{r} \exp\left(-\frac{L_{r,c}}{4} \log(\frac{1}{4e\delta_0})\right)$$

$$= \sum_{r=1}^{\infty} \sum_{c=1}^{r} \exp\left(-\frac{r \cdot 2^{(r-c)} L_1}{4} \log(\frac{1}{4e\delta_0})\right)$$

$$= \sum_{r=1}^{\infty} \sum_{k=0}^{r-1} \exp\left(-\frac{r \cdot 2^{k} L_1}{4} \log(\frac{1}{4e\delta_0})\right) .$$

Since $L_1 = \lceil \frac{4 \log(1+\frac{2}{\delta})}{\log \frac{1}{4e\delta_0}} \rceil$, by evaluating an infinite sum (see Lemma C.2), we conclude the proof:

$$\sum_{r=1}^{\infty} \sum_{k=0}^{r-1} \exp\left(-\frac{r \cdot 2^{k} L_1}{4} \log(\frac{1}{4e\delta_0})\right)$$

$$\leq \sum_{r=1}^{\infty} \frac{3}{2} \exp\left(-r \log(1+\frac{2}{\delta})\right)$$

$$= \frac{3}{2} \cdot \frac{\exp\left(-\log(1+\frac{2}{\delta})\right)}{1 - \exp\left(-\log(1+\frac{2}{\delta})\right)} \qquad \text{(geometric sum)}$$

$$\leq \frac{3}{2} \cdot \frac{\delta}{2} \leq \delta.$$

One can see from above the reason why we set $L_{r,c}$ to be $\propto r2^{r-1}$ rather than $\propto 2^{r-1}$ in the algorithm – without the extra factor of $r$, the sum of will not be controlled.

## C.2. Proof of BrakeBooster's Exponential Stopping Tail

We prove a slightly more general version of Theorem 4.2.

**Theorem C.1** (Exponential stopping tail; full version). *Let an algorithm $\mathcal{A}$ be $\delta_0$-correct and have a high probability sample complexity of $T_{\delta_0}^*(\mathcal{A})$. Suppose we run BrakeBooster (Algorithm 2) with input $\mathcal{A}$, $\delta$, $L_1 = \lceil \frac{4 \log(1+\frac{2}{\delta})}{\log \frac{1}{4e\delta_0}} \rceil$, $T_1 \geq 1$, and $\delta_0 \leq (\frac{1}{2e})^2$. Let $T_0 = 24\bar{T}^* \log_2^2(\frac{16\bar{T}^*}{T_1})L_1$ where $\bar{T}^* = T_{\delta_0}^*(\mathcal{A}) \vee T_1$. Then,*

$$\forall T \geq T_0, \ \mathbb{P}\left(\tau(\mathcal{M}) \geq T\right) \leq \exp\left(-\frac{T}{768 T_{\delta_0}^*(\mathcal{A}) \log_2 T} \ln \frac{1}{\delta_0}\right)$$

*Proof.* Let $r^* := \min\{r \in \mathbb{N}_+ : T_{r,r} \geq T_{\delta_0}^*(\mathcal{A})\}$. Let $\bar{T}_{r,c}$ be the total amount samples consumed up to and including stage $(r,c)$. Define $R_{r,c} := \{\ell \in [L_{r,c}] : \ell\text{-th trial does not self-terminate }\}$.

Let $r > r^*$. Then,

$$\mathbb{P}\left(\tau(\mathcal{M}) \geq \bar{T}_{r,c}\right) \leq \mathbb{P}(J_{r-1,r^*} = 0)$$

Note that the fact that a stage returns 0 implies that more than a half of the trials did not self-terminate. Thus

$$\mathbb{P}(J_{r-1,r^*} = 0) \leq \mathbb{P}\left(\exists \text{ at least } (\lfloor L_{r-1,r^*}/2 \rfloor + 1) \text{ trials that do not self-terminate }\right)$$

$$\leq \mathbb{P}\left(|R_{r-1,r^*}| \geq \frac{L_{r-1,r^*}}{2}\right)$$

Since the trials in stage $(r-1, r^*)$ use samples more than $T_{\delta_0}^*(\mathcal{A})$, by the sample complexity guarantee of $\mathcal{A}$ and Lemma C.3, we have

$$\mathbb{P}\left(|R_{r-1,r^*}| \geq \frac{L_{r-1,r^*}}{2}\right) \leq \exp\left(-\frac{L_{r-1,r^*}}{2} \log \frac{1}{2e\delta_0}\right)$$

$$\leq \exp\left(-\frac{2^{r-1-r^*}(r-1)L_1}{2} \ln \sqrt{\frac{1}{\delta_0}}\right) \qquad (\sqrt{\delta_0} \leq \frac{1}{2e})$$

$$= \exp\left(-\frac{2^{r-1-r^*}(r-1)L_1}{4}\ln\frac{1}{\delta_0}\right)$$

Note that by Lemma C.5, we have, $\forall r \geq 2$,

$$\bar{T}_{r,c} \geq \frac{T_1 L_1}{2} \cdot 2^{r-1} \cdot r^2 \geq T_1 L_1 2^r \implies r \leq \log_2\left(\frac{\bar{T}_{r,c}}{L_1 T_1}\right)$$

Then,

$$\mathbb{P}\left(\tau(\mathcal{M}) \geq \bar{T}_{r,c}\right) = \exp\left(-\frac{2^{r-r^*-1}L_1}{4}(r-1)\ln\frac{1}{\delta_0}\right)$$

$$\leq \exp\left(-\frac{2^{r-r^*-1}L_1}{8}r\ln\frac{1}{\delta_0}\right) \qquad (r \geq 2 \implies r-1 \geq \tfrac{r}{2})$$

$$= \exp\left(-\frac{2^{r-1}L_1}{8 \cdot 2^{r^*}}r\ln\frac{1}{\delta_0}\right)$$

$$= \exp\left(-\frac{2^{r-1}T_1 L_1 r^2}{8 T_1 r \cdot 2^{r^*}}\ln\frac{1}{\delta_0}\right)$$

$$\leq \exp\left(-\frac{\bar{T}_{r,c}/3}{8 T_1 r \cdot 2^{r^*}}\ln\frac{1}{\delta_0}\right) \qquad (\text{Theorem C.5})$$

$$\leq \exp\left(-\frac{\bar{T}_{r,c}/3}{8 T_1 r \cdot \frac{8\bar{T}^*}{T_1}}\ln\frac{1}{\delta_0}\right) \qquad (\text{Theorem C.4}; \bar{T}^* := T^*_{\delta_0}(\mathcal{A}) \vee T_1)$$

$$= \exp\left(-\frac{\bar{T}_{r,c}/3}{64 r \cdot \bar{T}^*}\ln\frac{1}{\delta_0}\right)$$

$$\leq \exp\left(-\frac{\bar{T}_{r,c}/3}{64\bar{T}^* \log_2 \bar{T}_{r,c}}\ln\frac{1}{\delta_0}\right) \qquad (r \geq 2, r \leq \log_2 \tfrac{\bar{T}_{r,c}}{L_1 T_1} \leq \log_2 \bar{T}_{r,c})$$

Hence,

$$\forall r > r^* \quad \forall c \leq r \quad \mathbb{P}\left(\tau(\mathcal{M}) \geq \bar{T}_{r,c}\right) \leq \exp\left(-\frac{\bar{T}_{r,c}}{192\bar{T}^* \log_2 \bar{T}_{r,c}}\ln\frac{1}{\delta_0}\right)$$

Next, in order to obtain an exponential stopping tail guarantee, we need to upper bound $\mathbb{P}(\tau(\mathcal{M}) \geq T)$ for every $T$ that is sufficiently large instead of those particular $\bar{T}_{r,c}$'s.

Let $T \geq \bar{T}_{r^*+1,1}$. We consider two cases.

**Case 1.** $T \in [\bar{T}_{r,c}, \bar{T}_{r,c+1})$ for some $r > r^*$ and $c \leq r-1$.
In this case, we have

$$\bar{T}_{r,c+1} = \bar{T}_{r,c} + T_1 L_1 2^{r-1} \cdot r$$

$$\frac{\bar{T}_{r,c+1}}{\bar{T}_{r,c}} = 1 + \frac{T_1 L_1 2^{r-1} \cdot r}{\bar{T}_{r,c}}$$

$$\leq 1 + \frac{T_1 L_1 2^{r-1} r}{\frac{T_1 L_1}{2} \cdot 2^{r-1} \cdot r^2} \qquad (\bar{T}_{r,c} \geq \tfrac{T_1 L_1}{2} \cdot 2^{r-1} \cdot r^2)$$

$$= 1 + \frac{2}{r}$$

$$\overset{(d)}{\leq} 2 \qquad (r \geq r^* + 1 \geq 2)$$

This implies $\bar{T}_{r,c} \leq T < 2\bar{T}_{r,c}$. Then,

$$\mathbb{P}\left(\tau(\mathcal{M}) \geq T\right) \leq \mathbb{P}\left(\tau(\mathcal{M}) \geq \bar{T}_{r,c}\right)$$

$$\leq \exp\left(-\frac{\bar{T}_{r,c}}{192\bar{T}^* \log_2 \bar{T}_{r,c}} \ln \frac{1}{\delta_0}\right)$$

$$\leq \exp\left(-\frac{T}{384\bar{T}^* \log_2 T} \ln \frac{1}{\delta_0}\right) \qquad (\bar{T}_{r,c} \leq T < 2\bar{T}_{r,c})$$

**Case 2.** $T \in [\bar{T}_{r,r}, \bar{T}_{r+1,1})$ for some $r > r^*$.
In this case,

$$\bar{T}_{r+1,1} = \bar{T}_{r,r} + T_1 L_1 2^r (r+1)$$

$$\frac{\bar{T}_{r+1,1}}{\bar{T}_{r,r}} = 1 + \frac{T_1 L_1 2^r (r+1)}{\bar{T}_{r,r}}$$

$$\leq 1 + \frac{T_1 L_1 2^r (r+1)}{\frac{T_1 L_1}{2} \cdot 2^{r-1} \cdot r^2} \qquad (\bar{T}_{r,r} \geq \frac{T_1 L_1}{2} \cdot 2^{r-1} \cdot r^2)$$

$$= 1 + \frac{4(r+1)}{r^2}$$

$$= 4 \qquad (r \geq 2)$$

Thus, we have $\bar{T}_{r,r} \leq T < 4\bar{T}_{r,r}$ and

$$\mathbb{P}\left(\tau(\mathcal{M}) \geq T\right) \leq \mathbb{P}\left(\tau(\mathcal{M}) \geq \bar{T}_{r,r}\right)$$

$$\leq \exp\left(-\frac{\bar{T}_{r,r}}{192\bar{T}^* \log_2 \bar{T}_{r,r}} \ln \frac{1}{\delta_0}\right)$$

$$\leq \exp\left(-\frac{T}{768\bar{T}^* \log_2 T} \ln \frac{1}{\delta_0}\right) \qquad (\bar{T}_{r,r} \leq T < 4\bar{T}_{r,r})$$

Therefore, in either case, we have

$$T \geq \bar{T}_{r^*+1,1} \implies \mathbb{P}\left(\tau(\mathcal{M}) \geq T\right) \leq \exp\left(-\frac{T}{768\bar{T}^* \log_2 T} \ln \frac{1}{\delta_0}\right)$$

We conclude the proof by working out an explicit upper bound on $\bar{T}_{r^*+1,1}$ as follows:

$$\bar{T}_{r^*+1,1} \leq \bar{T}_{r^*+1,r^*+1}$$

$$\leq 2^{r^*} \cdot 3(r^*+1)^2 L_1 T_1 \qquad \text{(Theorem C.5)}$$

$$\leq \frac{8\bar{T}^*}{T_1} \cdot 3(\log_2 \frac{16\bar{T}^*}{T_1})^2 L_1 T_1 \qquad \text{(Theorem C.4)}$$

$$\leq 24\bar{T}^* \log_2^2\left(\frac{16\bar{T}^*}{T_1}\right) L_1$$

$\square$

### C.3. Utility Lemmas

**Lemma C.2.** *Let $\beta > 0$, $a \geq 1$, $\delta \in (0,1)$, and $\delta_0 \in (0,1)$. If, $L_1 \geq \alpha \frac{\ln\left(1+\frac{2}{\delta}\right)}{\ln\left(\frac{1}{\beta\delta_0}\right)}$, then*

$$\sum_{s=1}^{\infty} \exp\left(-a \cdot \frac{2^{s-1} L_1}{\alpha} \ln\left(\frac{1}{\beta\delta_0}\right)\right) \leq \frac{3}{2} \exp\left(-a \cdot \ln(1+\frac{2}{\delta})\right)$$

*Proof.*

$$\sum_{s=1}^{\infty} \exp\left(-a \cdot \frac{2^{s-1}L_1}{\alpha} \ln\left(\frac{1}{\beta\delta_0}\right)\right)$$

$$= \exp\left(-a \cdot \frac{L_1}{\alpha} \ln\left(\frac{1}{\beta\delta_0}\right)\right) + \exp\left(-a \cdot \frac{2L_1}{\alpha} \ln\left(\frac{1}{\beta\delta_0}\right)\right) + \exp\left(-a \cdot \frac{4L_1}{\alpha} \ln\left(\frac{1}{\beta\delta_0}\right)\right) + \cdots$$

$$\leq \exp\left(-a \cdot \frac{L_1}{\alpha} \ln\left(\frac{1}{\beta\delta_0}\right)\right) + \exp\left(-a \cdot \frac{2L_1}{\alpha} \ln\left(\frac{1}{\beta\delta_0}\right)\right) + \exp\left(-a \cdot \frac{3L_1}{\alpha} \ln\left(\frac{1}{\beta\delta_0}\right)\right) + \cdots$$

$$= \frac{\exp\left(-a \cdot \frac{L_1}{\alpha} \ln\left(\frac{1}{\beta\delta_0}\right)\right)}{1 - \exp\left(-a \cdot \frac{L_1}{\alpha} \ln\left(\frac{1}{\beta\delta_0}\right)\right)} \qquad \text{(geometric sum)}$$

Let $L_1 \geq \frac{\alpha \ln\left(1+\frac{2}{\delta}\right)}{\ln \frac{1}{\beta\delta_0}}$. Then,

$$1 - \exp\left(-a \cdot \frac{L_1}{\alpha} \ln\left(\frac{1}{\beta\delta_0}\right)\right) = 1 - \exp\left(-a \ln\left(1 + \frac{2}{\delta}\right)\right)$$

$$= 1 - \frac{1}{\left(1 + \frac{2}{\delta}\right)^a}$$

$$\geq 1 - \frac{1}{\left(1 + \frac{2}{\delta}\right)} \qquad (a \geq 1)$$

$$\geq 1 - \frac{1}{\left(1 + \frac{2}{1}\right)} \qquad (\delta \leq 1)$$

$$= \frac{2}{3}$$

Thus,

$$\frac{\exp\left(-a \cdot \frac{L_1}{\alpha} \ln\left(\frac{1}{\beta\delta_0}\right)\right)}{1 - \exp\left(-a \cdot \frac{L_1}{\alpha} \ln\left(\frac{1}{\beta\delta_0}\right)\right)} \leq \frac{3}{2} \exp\left(-a \cdot \ln(1 + \frac{2}{\delta})\right),$$

which completes the proof. $\qquad\square$

**Lemma C.3.** *Let $\mathcal{E}$ be an event from a random trial such that $\mathbb{P}(\mathcal{E}) \leq \delta$. Let $\alpha$ satisfy $\delta < \alpha < 1$. Let $N$ be the number of trials where $\mathcal{E}$ holds true out of $L$ independent trials. Then,*

$$\mathbb{P}(N \geq \alpha L) \leq \exp\left(-\alpha L \log\left(\frac{1}{e\delta}\right)\right)$$

*Proof.* Recall the standard KL divergence based concentration inequality where $\hat{\mu}_n$ is the sample mean of $n$ Bernoulli i.i.d. random variables with head probability $\mu$:

$$\forall \varepsilon \geq 0, \mathbb{P}(\hat{\mu}_n - \mu \leq \varepsilon) \leq \exp(-n\mathbf{KL}(\mu + \varepsilon, \mu)) \,.$$

Note that $N/L$ can be viewed as the sample mean of Bernoulli trials with $\mu := \mathbb{P}(\mathcal{E})$. Then,

$$
\begin{aligned}
\mathbb{P}(N \geq \alpha L) &= \mathbb{P}(\frac{N}{L} \geq \alpha) \\
&= \mathbb{P}(\frac{N}{L} - \mu \geq \alpha - \mu) \\
&\leq \exp(-L\mathbf{KL}(\alpha, \mu)) && (\alpha > \delta \geq \mu) \\
&= \exp\left(-L\left(\alpha \ln(\frac{\alpha}{\mu}) + (1-\alpha)\ln\frac{1-\alpha}{1-\mu}\right)\right) \\
&\overset{(a)}{\leq} \exp\left(-L\left(\alpha \ln(\frac{\alpha}{\mu}) - \alpha\right)\right) \\
&= \exp\left(-L\alpha \ln(\frac{\alpha}{e\mu})\right)
\end{aligned}
$$

where $(a)$ is by the following derivation:

$$
\begin{aligned}
(1-\alpha)\ln\frac{1-\alpha}{1-\mu} &= -(1-\alpha)\ln\frac{1-\mu}{1-\alpha} \\
&= -(1-\alpha)\ln\left(1 + \frac{\alpha - \mu}{1-\alpha}\right) \\
&\geq -(1-\alpha) \cdot \frac{\alpha - \mu}{1-\alpha} && (\forall x, \ln(1+x) \leq x) \\
&= -(\alpha - \mu) \\
&\geq -\alpha
\end{aligned}
$$

Hence,

$$\mathbb{P}(N \geq \alpha L) \leq \exp\left(-L\alpha \ln(\frac{\alpha}{e\mu})\right) \leq \exp\left(-L\alpha \ln(\frac{\alpha}{e\delta})\right) \qquad (\mu = \mathbb{P}(\mathcal{E}) \leq \delta)$$

$\square$

**Lemma C.4.** *Let $T_{r,r} = 2^{r-1}T_1$ for some $T_1 \geq 1$ and define $\bar{T}^* := T^* \vee T_1$. Define $r^* := \min\{r \in \mathbb{N}_+ : T_{r,r} \geq T^*\}$. Then,*

$$r^* \leq \log_2 \frac{8\bar{T}^*}{T_1}$$

*Proof.* Consider two cases:

(i) $r^* \geq 2$: In this case,

$$T^* > T_{r^*-1, r^*-1} = 2^{r^*-2}T_1$$
$$\implies r^* < \log_2(\frac{4T^*}{T_1})$$

(ii) $r^* \leq 1$: Nothing to do here.

Together, we have $r^* \leq 1 \vee \log_2 \frac{4T^*}{T_1} \leq 1 \vee \log_2 \frac{4\bar{T}^*}{T_1} \leq \log_2 \frac{8\bar{T}^*}{T_1}$ where we use the fact that $\forall a, b \geq 0, a \vee b \leq a + b$ and $\log_2(4\bar{T}^*/T_1) \geq 2 \geq 0$. $\qquad \square$

**Lemma C.5.** *Let $\bar{T}_{r,c}$ be the total number of samples used in Algorithm 2 up to and including stage $(r, c)$. If $r \geq 2, c \in [r]$, then*

$$\frac{1}{2} \leq \frac{\bar{T}_{r,c}}{r^2 2^{r-1} T_1 L_1} \leq 3 .$$

*Proof.* For the upper bound,

$$
\begin{aligned}
\bar{T}_{r,c} \leq \bar{T}_{r,r} &\leq \sum_{u=1}^{r} \sum_{c=1}^{u} u \cdot 2^{u-1} L_1 T_1 \\
&= (r^2 2^{r-1} - r 2^{r+1} + 3 \cdot 2^r - 3) L_1 T_1 \\
&\leq 2^{r-1}(r^2 + 6) L_1 T_1 \\
&\leq 2^{r-1} \cdot 3 r^2 L_1 T_1 \qquad\qquad (6 \leq 2r^2)
\end{aligned}
$$

For the lower bound,

$$
\begin{aligned}
\bar{T}_{r,c} &= \sum_{u=1}^{r-1} \sum_{c=1}^{u} T_1 L_1 \cdot u \cdot 2^{u-1} + \sum_{v=1}^{c} T_1 L_1 \cdot r \cdot 2^{r-1} \\
&= T_1 L_1 \sum_{u=1}^{r-1} u^2 \cdot 2^{u-1} + T_1 L_1 \cdot c \cdot r \cdot 2^{r-1} \\
&= T_1 L_1 \cdot \left( 2^{r-1} \left( r^2 - 4r + 6 \right) - 3 \right) + T_1 L_1 \cdot c \cdot r \cdot 2^{r-1} \\
&\geq T_1 L_1 \cdot \left( 2^{r-1} \left( r^2 - 4r + 6 \right) - 3 \right) + T_1 L_1 \cdot r \cdot 2^{r-1} \\
&= T_1 L_1 \cdot \left( 2^{r-1} \left( r^2 - 3r + 6 \right) - 3 \right) \\
&= \frac{T_1 L_1}{2} \cdot \left( 2^{r-1} \left( 2r^2 - 6r + 12 \right) - 6 \right) \\
&= \frac{T_1 L_1}{2} \cdot 2^{r-1} \cdot r^2 + \frac{T_1 L_1}{2} \cdot \left( 2^{r-1} \left( r^2 - 6r + 12 \right) - 6 \right) \\
&\geq \frac{T_1 L_1}{2} \cdot 2^{r-1} \cdot r^2 + \frac{T_1 L_1}{2} \cdot 2^{r-1} \left( r^2 - 6r + 9 \right) \qquad\qquad (r \geq 2) \\
&= \frac{T_1 L_1}{2} \cdot 2^{r-1} \cdot r^2 + \frac{T_1 L_1}{2} \cdot 2^{r-1}(r - 3)^2 \\
&\geq \frac{T_1 L_1}{2} \cdot 2^{r-1} \cdot r^2
\end{aligned}
$$

$\qquad \square$

# D. Empirical studies

## D.1. Empirical Evaluation of Successive Elimination (SE)

In this section we present the study of stopping time distribution of Successive Elimination (SE) (Even-Dar et al., 2006) algorithm.

**Experimental setup:** We implemented SE in two different configurations, 1) Original version 2) Version with $\varepsilon$-slack added to the stopping condition. We set the number of arms 3, with mean rewards $\{1.0, 0.9, 0.9\}$. Noise follows $\mathcal{N}(0, 1)$. We set $\delta = 0.01$. We conduct 1000 trials and observe the stopping times of those trials. We forcefully terminated the trials that do not stop until 30,000 time steps or 1,000,000 time steps. In Figure 5, 6 we have plotted the histograms of observed stopping times for all the trials.

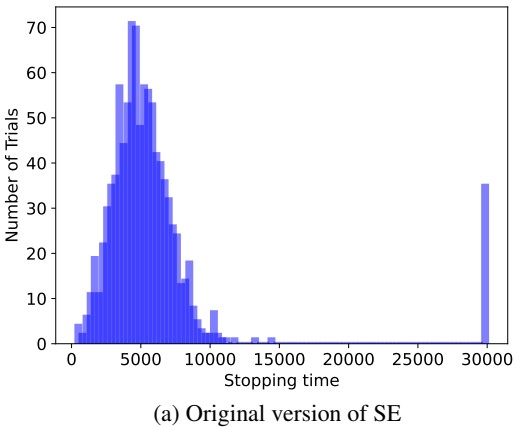

(a) Original version of SE

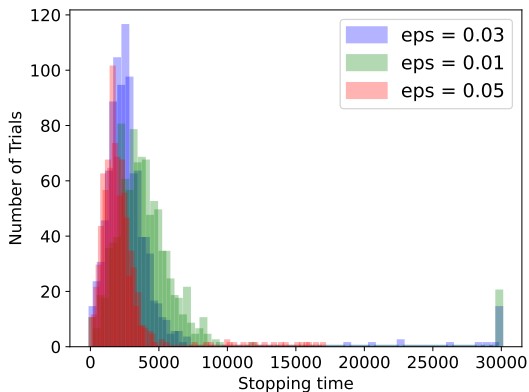

(b) SE with $\varepsilon$-slack added to the stopping condition

*Figure 5.* Histogram of stopping times (force stop after 30K rounds)

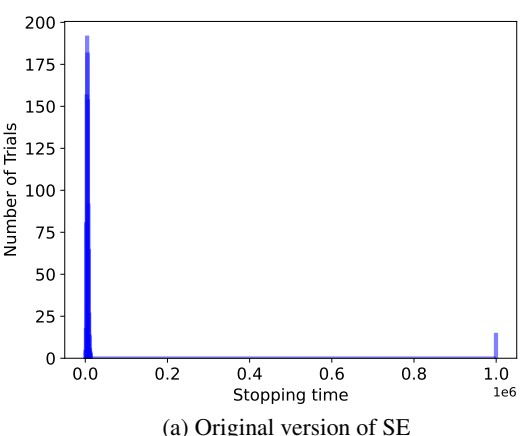

(a) Original version of SE

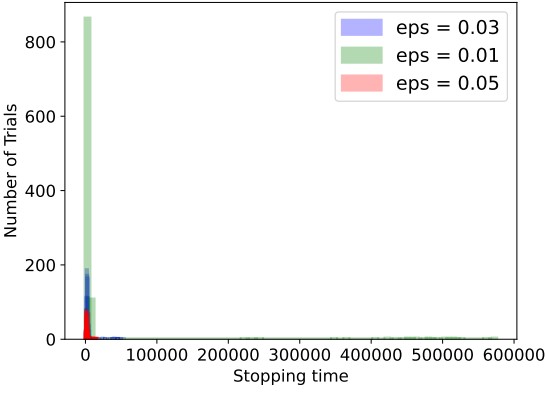

(b) SE with $\varepsilon$-slack added to the stopping condition

*Figure 6.* Histogram of stopping times (force stop after 1M rounds)

**Observations:** In the case of *original version of SE*, a considerable number of trials have not been terminated even after 30,000 time steps (Figure 5a). We have observed that all these trials have already eliminated the best arm, and thus we expect that many of them will never stop. This is also confirmed by Figure 6a where most of the trials that did not stop after 30,000 time steps are still running after $1,000,000$ time steps. This shows that a high probability stopping time bound does not guarantee that all the trials will stop.

In the case of *SE with $\varepsilon$-slack added to the stopping condition*, first, we can see that for $\varepsilon = 0.01, 0.03$, still some of the

trials are not terminated even after 30,000 time steps. It takes up to $600,000$ time steps for all the trials to terminate. This is significantly higher as discussed in the main part of the paper.

Moreover, once the best arm is eliminated, the rest of the procedure can be considered as uniform sampling when $\Delta_2 = \varepsilon$. Hence, as we have discussed in the main part of this paper, the stopping time distribution will follow $(T_\delta, \kappa)$-exponential stopping tail with $T_\delta = \tilde{\Theta}(K\varepsilon^{-2}\ln(1/\delta))$ and $\kappa = \Theta(K\varepsilon^{-1})$. The same goes for more general cases as well. However our proposed algorithms achieve a better $(\tilde{\Theta}(H_1 \ln(1/\delta)), H_1)$-exponential stopping tail.

### D.2. Empirical Evaluation of SE with Brakebooster

In this section, we examine the impact of incorporating the Brakebooster algorithm into the Successive Elimination (SE) framework, where Brakebooster operates by taking SE as its input.

**Experimental Setup:** We consider a bandit problem with four arms, each associated with mean rewards of $\{1.0, 0.6, 0.6, 0.6\}$. The reward noise is drawn from a normal distribution $\mathcal{N}(0, 1)$. The confidence parameter is set to $\delta = 0.01$. We perform 1000 independent trials and record the stopping times for each. Trials that do not terminate within 1,000,000 time steps are forcefully stopped at that point. Figures 7a and 7b present histograms of the stopping times observed across all trials for SE and SE augmented with BrakeBooster, respectively. Additionally, Figure 7c shows a comparative cumulative distribution function (CDF), scaled over 1000 trials, for both SE and SE with BrakeBooster. We also conduct the same set of experiments using a four-arm bandit problem with mean rewards $\{1.0, 0.9, 0.9, 0.9\}$. Furthermore, In these experiments, we employ a $1.2\times$ growth factor for both the per-trial budget and the number of trials, in contrast to the conventional doubling scheme, speculating that the exponential guarantee will still be preserved. The corresponding results are presented in Figure 8.

**Observations:** The results show that the BrakeBooster mechanism ensures that all the instances stop. Furthermore Figure 7c and 8c shows that applying BrakeBooster on Successive Elimination helps to stop all the trials without sacrificing too many samples (the CDF curve of BrakeBooster+SE catches up with that of SE very fast because the crossover point is at the stopping time of $\sim 2 \times 10^4$ for Figure 7c and $\sim 0.05 \times 10^6$ for Figure 8c).

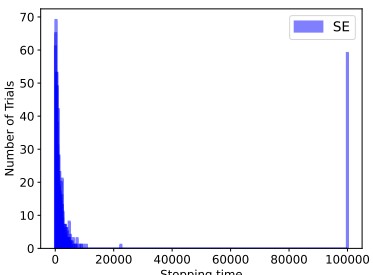 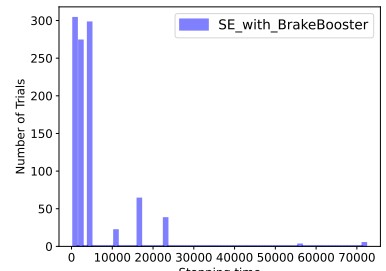 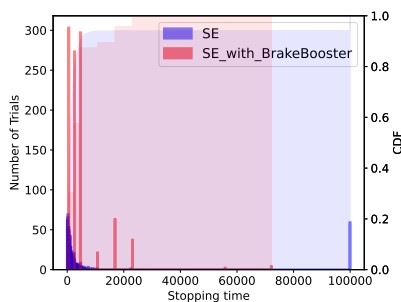

(a) Successive Elimination (SE): Some trials fail to stop.  (b) BrakeBooster + SE: All trials successfully stop.  (c) Both plots with CDF of stopping times shaded with respective colors

*Figure 7.* Histogram of stopping times for problem instance $\mathcal{A} = \{1.0, 0.6, 0.6, 0.6\}$ (force stop after 0.1M samples). Results from 1K trials.

### D.3. Empirical Evaluation of LUCB1, TS-TCI, and FC-DSH

In this section, we analyze the stopping times of LUCB1 (Kalyanakrishnan et al., 2012), TS-TCI (Jourdan et al., 2022), and FC-DSH (ours).

**Experimental Setup:** For the implementation of FC-DSH algorithm, we deviate from the theoretical version presented in the paper by reusing samples across rounds. We opt to implement this practical version to reflect a more efficient use of data in empirical settings. We remark that the theoretical analysis can still be done by taking a union bound over the arms just like how Successive Rejects (Audibert et al., 2010) analysis can be done with sample reuse. Furthermore, instead of *doubling* the budget after each phase, we generalize the growth schedule by introducing a scaling parameter $b$, such that $T_m = b^{m-1}T_1$, for phase $m \geq 2$. While our analysis focuses on the case $b = 2$, we speculate that the exponential guarantee may hold for any value of $b$. In our experiments, we set $b = 1.01$ to allow for finer budget increment. Additionally, we adopt

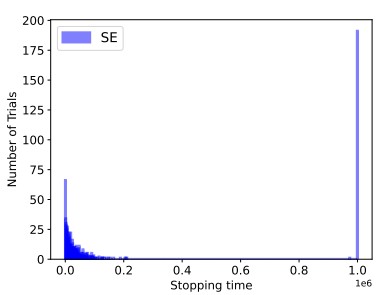 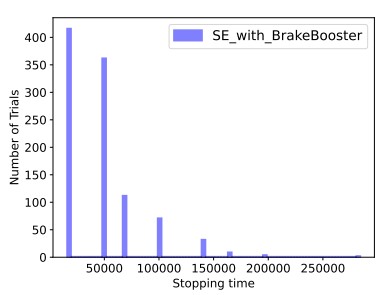 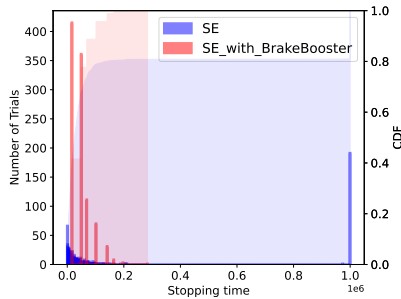

(a) Successive Elimination (SE): Some trials fail to stop.

(b) BrakeBooster + SE: All trials successfully stop.

(c) Both plots with CDF of stopping times shaded with respective colors

*Figure 8.* Histogram of stopping times for problem instance $\mathcal{A} = \{1.0, 0.9, 0.9, 0.9\}$ (force stop after 1M samples). Results from 1K trials.

the stopping rule proposed by (Jourdan et al., 2022) across all algorithms to ensure fair comparison.

In this experiment, we consider multiple bandit instances with varying number of arms $K \in \{4, 8, 16\}$. For each instance, the mean reward of the optimal arm is set to $1.0$, while all sub-optimal arms have mean rewards of $0.6$. For an instance, a $4$-armed bandit instance is associated with mean rewards of $\{1.0, 0.6, 0.6, 0.6\}$. The reward noise is drawn from a normal distribution $\mathcal{N}(0, 1)$. The confidence parameter is set to $\delta = 0.05$. We perform 1000 independent trials and record the stopping times for each. Trials that do not terminate within 1,000,000 time steps are forcefully stopped after that time step. Figure 9, 10, and 11 illustrate the histograms and cumulative distribution functions (CDFs) of the stopping times observed for LUCB1, TS-TCI, and FC-DSH across instances with $K \in \{4, 8, 16\}$.

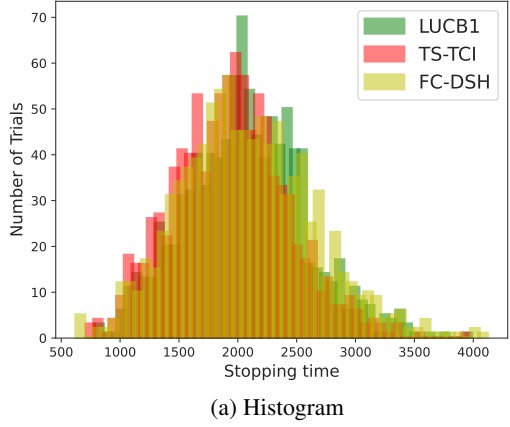 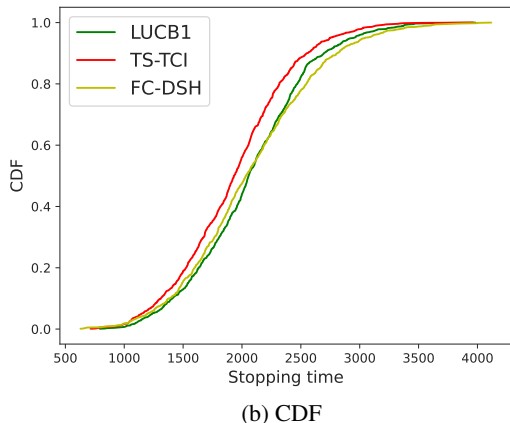

(a) Histogram

(b) CDF

*Figure 9.* Histogram and CDF of stopping time of LUCB1, TS-TCI, and FC-DSH on the instance with $K = 4$.

**Observations:** First, the plots confirm that LUCB1, TS-TCI, and FC-DSH all successfully stops, in contrast to the Successive Elimination (SE), which may fail to stop (Figure 7a). Secondly, unsurprisingly, the stopping time increase as the number of $K$ grows. Among the methods, TS-TCI exhibits the best performance, for $K = 8$ and $K = 16$, whereas FC-DSH yields the longest stopping times.

**Additional Experimental Setup:** To further investigate the tail behavior of the stopping time distribution, we conduct an additional experiment focusing on the tail probability $P(X > x)$ as shown in Figure 12. Anticipating that the tail behavior would become more apparent with a significantly larger number of trials, we increase the number of trials from 1000 to 1,000,000. The experiment is performed on a 4-armed bandit instance with mean rewards of $\{1.0, 0.6, 0.6, 0.6\}$ with all other settings remain identical to those described earlier in this section.

**Observations:** Figure 12 confirms that FC-DSH exhibits a exponential tail stopping time. Unexpectedly, it does not provide evidence that LUCB1 exhibits a polynomial tail. If LUCB1 follows polynomial tail, we would expect to observe a linear

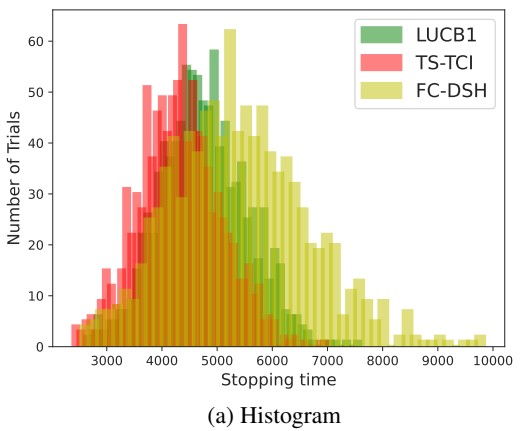
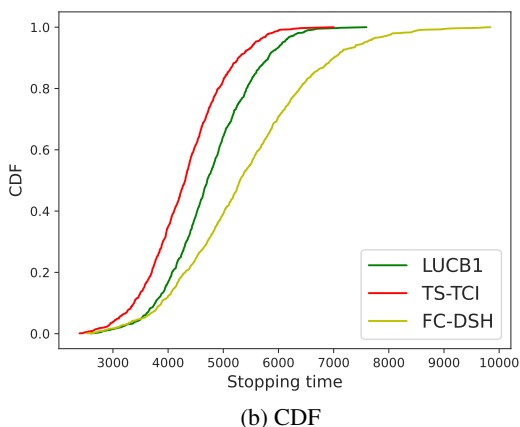

(a) Histogram

(b) CDF

*Figure 10.* Histogram and CDF of stopping time of LUCB1, TS-TCI, and FC-DSH on the instance with $K = 8$.

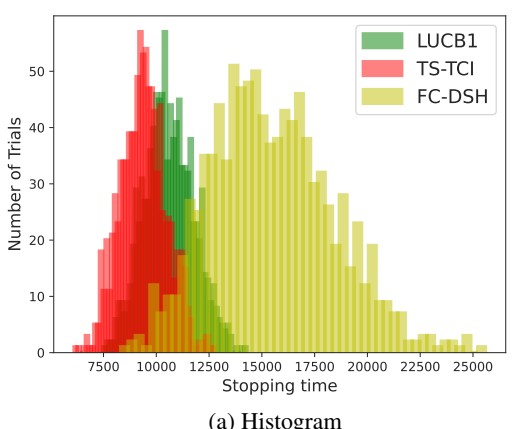
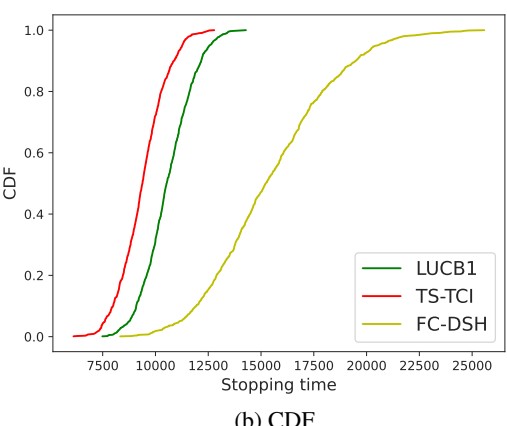

(a) Histogram

(b) CDF

*Figure 11.* Histograms and CDFs of stopping times of LUCB1, TS-TCI, and FC-DSH on the instance with $K = 16$.

decay trend in the plot; however, such a pattern does not emerge. There are two plausible interpretations: (i) LUCB1 may exhibit an exponential tail, suggesting that existing theoretical guarantees are loose and warrant tighter analysis, (ii) LUCB1 indeed has a polynomial tail, but a much larger number of trials may be required to empirically verify it.

Furthermore, TS-TCI shows intriguing behavior. In the regime where stopping times fall between approximately 2,500 and 10,000, TS-TSC exhibits a roughly linear decay in $\log(P(X > x))$. We conjecture that a trade-off exists: the aggressive sampling approach of TS-TCI likely results in heavier-tailed stopping time distributions. In addition, the more aggressive in sampling strategy, the faster the algorithm enters the linear decay regime. Obtaining a positive/negative answer to this phenomenon is left as future work.

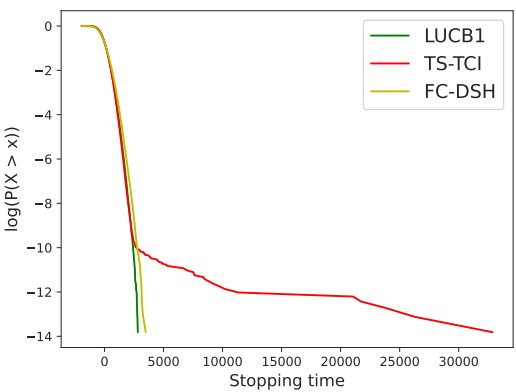

*Figure 12.* Log of tail probability $\log(P(X > x))$ curve on the instance with $K = 4$. Results from 1,000,000 trials.

