# OpenReview forum: "Fixing the Loose Brake: Exponential-Tailed Stopping Time in Best Arm Identification"
_ICML.cc/2025/Conference — ICML 2025 poster_

### Official Review · Reviewer_YUrK · 2025-02-26

**Overall Recommendation:** 3

**Summary:**

This paper is related to the best arm identification (BAI) problem in multi-armed bandits, where the objective is to recommend the (unique) arm with highest expected reward at a fixed error rate $\delta$ ($\delta$-correctness property) after collecting as few observations as possible.  The stopping time of a run of a best arm identification bandit algorithm is the number of observations collected up to the recommendation time. This paper focuses on the problem of guaranteeing that the distribution of stopping times is light-tailed and emphasizes on the fact that few papers from the literature try to tackle the problem of sometimes very long or infinitely long runs, even though the authors show that some of the most well-known algorithms from prior works exhibit such an undesirable behavior. The authors advocate for a stronger type of guarantee on the sample complexity than what is proposed in the literature so far, named exponential tail stopping time. The authors introduce a $\delta$-correct algorithm named FC-DSH which exhibits an exponential tail stopping time, and a meta-algorithm called BrakeBooster which takes as input any $\delta_0$-correct algorithm with weaker guarantees on the stopping time and returns a $\delta$-algorithm with exponential-tail stopping time.

## update after rebuttal

I keep the score to 3. The problem of non-terminating runs seems limited to older algorithms, so I am still on the fence regarding the significance of the problem. But the strengths of the paper still hold so I would suggest acceptance.

**Claims And Evidence:**

Most of the theoretical and practical claims are convincing and clear. However:
- While I understand the point of disentangling correctness and sample complexity, I disagree with the following claim: Lines 153-155, Page 3 “In practice, one may desire to be loose on the correctness (large δ) yet want to ensure that the stopping time is small with very high confidence (small δ).” If this were true, then one would usually turn to fixed-budget or anytime settings (where an error bound is provided at each sampling time) rather than to a fixed-confidence setting.
- Lines 98-99, Page 2 “The computational efficiency of FC-DSH’s is not worse (orderwise) than other algorithms.” As there are no experiments comparing the performance of FC-DSH and other fixed-confidence BAI algorithms, and as there is an increasing sampling budget for phases, it is unclear to me if this sentence is true.
- I am still not convinced about the importance of the problem of heavy-tailed stopping time distributions in fixed-confidence BAI. As demonstrated by experiments (Figures 1 and 3-4), not stopping at all or long stopping times are extremely rare events, even in carefully crafted bandit instances such as the one used in Theorems 2.4-2.5 (e.g., $1/(8\sqrt{\pi})(\delta/3.3)^ 118 \approx 10^{-99}$ for $\delta=0.5$ from Theorem 2.4). Moreover, in practice, one actually performs somehow the approach of BrakeBooster: stop the bandit algorithm after some time ($10^ 6$ for instance) and restart the algorithm with a new random seed for synthetic data / with new observations for real-life data (discarding prior observations in the process). But I agree that this approach only allows for correctness to hold and does not provide very strong guarantees on the stopping time. But is it worth it to sacrifice global performance for an event of small probability?
- Table 1: LUCB has an upper bound in high probability on the sample complexity (their Corollary 7).

**Essential References Not Discussed:**

None.

**Experimental Designs Or Analyses:**

The experiments of counting the number of forcefully terminated runs in Successive Elimination look correct. However, no code is provided (supplementary zip file or anonymous code repository) to check the reproducibility of the results.

**Methods And Evaluation Criteria:**

The authors focus on a single aspect of multi-armed bandits for fixed-confidence BAI, which is the tail of the distribution of stopping times, whereas most papers from the literature focus on the performance in high probability or in (asymptotic) expectation (as shown in Table 1). As such, especially as there is no experiments comparing to baselines (both in terms of quality of the recommendation, average stopping time or even a comparison of the distribution of stopping times of FC-DSH compared to the ones plotted for Successive Elimination in Figures 3-4, or better, more recent algorithms as those listed in Table 1), it makes the assessment of the theoretical improvement incurred by this paper difficult.

**Other Comments Or Suggestions:**

- Caption of Figure 1: “Historgram” should be “histogram”.
- Throughout Section A in Appendix: “⊥” instead of the symbol for probability.
- Experiments in the supplementary material (Section D): the exact number of forcefully stopped runs should be given for Figures 3-4, it is hard to estimate the proportion of these runs among the 1,000 trials from the histograms alone.

**Other Strengths And Weaknesses:**

Strengths
- The idea of a meta-algorithm converting any “base” fixed-confidence BAI algorithm into an algorithm with stronger guarantees on the stopping time is interesting and well-executed.
- The paper is well-written and attracts attention on a not-so-investigated problem in the BAI literature.
- Disentangling correctness and stopping time probabilities is a good idea.

Weaknesses
- The importance of the problem is not obvious to me.

**Questions For Authors:**

The main weakness of this paper and the reason why I rated it 3 is because I am not convinced by the importance of getting an exponential-tail stopping time, especially as logarithmic factors and large constants are present in the analysis of the algorithmic contributions (as mentioned in the discussion). To clarify this:
- Isn’t it better to get a polynomial bound on the stopping time or a non-asymptotic bound on the expected stopping time and stronger guarantees on the sample complexity in high probability than to get an exponential-tail stopping time and a less good “recommendation performance” constant (derived from fixed-budget algorithms, where it is unclear whether such constants would match (even with logarithmic factors) the lower bounds on the minimal sample complexity for $\delta$-correct algorithms [1-2])?
[1] Kaufmann, E., Cappé, O., & Garivier, A. (2016). On the complexity of best-arm identification in multi-armed bandit models. The Journal of Machine Learning Research, 17(1), 1-42.
[2] Degenne, R. (2023, July). On the existence of a complexity in fixed budget bandit identification. In The Thirty Sixth Annual Conference on Learning Theory (pp. 1131-1154). PMLR.
- Could you perform the same kind of experiments run on Successive Elimination on FC-DSH or on BrakeBooster with Successive Elimination?
- Can you compare computationally speaking the cost of FC-DSH compared to other algorithms from the literature?

**Relation To Broader Scientific Literature:**

The authors propose a new type of guarantee on the sample complexity of a bandit algorithm, which diverges from traditional approaches (high probability upper bound [1], upper bound in expectation [2]). The algorithmic contributions are based on the doubling trick (which “almost” doubles the sampling budget in a trial) which is very used in the bandit literature [3] and on a well-known fixed-budget algorithm named Sequential Halving [4-5]. The technical tools (concentration inequalities, correctness analysis by contradiction) are standard in the bandit literature (see prior citations).

[1] Kalyanakrishnan, S., Tewari, A., Auer, P., & Stone, P. (2012, June). PAC subset selection in stochastic multi-armed bandits. In ICML (Vol. 12, pp. 655-662).

[2] Garivier, A., & Kaufmann, E. (2016, June). Optimal best arm identification with fixed confidence. In Conference on Learning Theory (pp. 998-1027). PMLR.

[3] Besson, L., & Kaufmann, E. (2018). What doubling tricks can and can't do for multi-armed bandits. arXiv preprint arXiv:1803.06971.

[4] Karnin, Z., Koren, T., & Somekh, O. (2013, May). Almost optimal exploration in multi-armed bandits. In International conference on machine learning (pp. 1238-1246). PMLR.

[5] Zhao, Y., Stephens, C., Szepesvári, C., & Jun, K. S. (2023, July). Revisiting simple regret: Fast rates for returning a good arm. In International Conference on Machine Learning (pp. 42110-42158). PMLR.

**Theoretical Claims:**

I have checked the supplementary material for the correctness and exponential tail stopping time guarantees for FC-DSH and BrakeBooster (not the technical lemmas in Section C.2) and they seem all correct to me.

---

> ### Author Rebuttal · Authors · 2025-04-01
>
> We thank the reviewer for recognizing the significance of our problem, its novelty in bandit literature, and the strength of an exponentially decaying stopping time over high-probability bounds. The detailed description and anonymous link of the additional experiments we did are in **Additional empirical evidences** section from the rebuttal for reviewer **94Yb**. Please take a look. We address the comments below.
>
> **I am still not convinced about the importance of the problem of heavy-tailed stopping time distributions in fixed-confidence BAI ... ?**
>
> A light-tailed distribution offers several benefits:
>
> 1) Even-though the event of non stopping happens with a low probability, it leads to an unknown expected stopping time, rendering the user clueless. We think, a little bit of inflation in the expected stopping time is better than a totally unknown expected stopping time in an implementation perspective.
>
> 2) High probability stopping time can be considered as a single point guarantee. In contrast our results provide a broader guarantee, where we can predict what happens to the stopping time when the requirements are modified.
>
> 3) The algorithms that provide a light tailed guarantee can be easily adapted as anytime algorithms.
>
>
>
> Furthermore, we think our contribution is theoretical, that is, we want to prove that it is possible to obtain a exponentially decaying distribution of stopping time theoretically, rather than proposing a practically efficient algorithm. Moreover, our work is a first step of showing that it is possible. There is no evidence we are aware of that indicates that achieving exponential stopping time necessarily will inherently sacrifice the performance.
>
>
> **While I understand the point of disentangling correctness and sample complexity, I disagree with the following claim ... ?**
>
> That is good a point. If we be less rigorous and consider the fixed budget settings have $\delta$-correctness ($\delta$ specified) and $0$-stopping time (stopping time = budget and $\delta = 0$ here) and anytime settings have $\delta$-correctness ($\delta$ non-specified) and $0$-stopping time (stopping time = whenever the user stops and $\delta = 0$ here), even then we do not have the full freedom to choose different $\delta$. Hence, even though this is a good strategy, it does not exactly serve our intentions.
>
> **The computational efficiency of FC-DSH’s is not worse (orderwise) than other algorithms ....  ?**
>
> The computational complexity we have mentioned here is not sample complexity. It is just processor time needed for algorithm implementation
>
>
> **Table 1: LUCB has an upper bound in high probability on the sample complexity (their Corollary 7)**
>
> Thanks for pointing it out. We will correct it in the final version.
>
> **Isn’t it better to get a polynomial bound on the stopping time or a non-asymptotic bound on the expected stopping time ..... ?**
>
> This is an important topic for discussion. We do not view the exponential tail as an exclusive guarantee such that to achieve it, we always have to incur some inflation in the sample complexity. There could be some algorithms (TS-TCI could be a strong candidate) which can achieve the best of both worlds. Our paper prevails as the early attempt to introduce the importance of exponential tail and taking a first step to achieve it. This will lead to more imminent research to come up with algorithms that can achieve the best of both worlds.
>
> Furthermore the experiments (Figure 1b and 2b) we have done indicates that LUCB1 which achieves polynomial tail exhibits a worse performance compared to our FC-DSH which achieves an exponential tail.
>
> **Could you perform the same kind of experiments ... ?**
>
> We did some additional experiments comparing FC-DSH to TS-TCI, LUCB1 and SE (Figure 1 and 2). Also analyzing the effect of Brakebooster on SE (Figure 3b). The detailed description and anonymous link are in Additional empirical evidences section from the rebuttal for reviewer 94Yb. Please take a look.
>
> **Can you compare computationally speaking the cost of FC-DSH compared to other algorithms from the literature?**
>
> We could do that. However we have observed that orderwise there is no difference in the computational cost of our algorithm and the other algorithms in the literature. We also speculate that, since the trials can be implemented in parallel ($L_{r,c}$ trials used for voting are independent, hence can be parallelized), computational cost can even be improved.
>
> We’d be more than happy to address any other questions and concerns.

---

> > ### Comment · Reviewer_YUrK · 2025-04-01
> >
> > Thanks for your rebuttal.
> >
> > **I am still not convinced about the importance of the problem of heavy-tailed stopping time distributions in fixed-confidence BAI ... ?**
> >
> > Thanks for your reply. I understand the point now. However, the impact of your work would have been perhaps more significant if it showed indeed that the constraint on the stopping time still allows to (nearly) match known lower bounds on sample complexity. The samples are independent across stages in the BrakeBooster algorithm, so it seems likely that any BAI algorithm wrapped with BrakeBooster would perform worse sample-complexity-wise than its counterpart without the meta-algorithm.
> >
> > **While I understand the point of disentangling correctness and sample complexity, I disagree with the following claim ... ?**
> >
> > OK, thanks for your answer, it is convincing.
> >
> > **Isn’t it better to get a polynomial bound on the stopping time or a non-asymptotic bound on the expected stopping time ..... ?**
> >
> > Thanks for your reply. I guess it is the same concern as the first question, and then showing that there is an algorithm which is known to be good sample-complexity-wise and also with a good bound on the stopping time would make this work perhaps more significant.
> >
> > **Could you perform the same kind of experiments ... ?**
> >
> > Thanks for running those experiments. The problem of non-terminated runs seems to be more prevalent in Successive Elimination than in more recent algorithms, which makes the problem of non-terminated runs perhaps less significant than I expected.
> >
> > **Can you compare computationally speaking the cost of FC-DSH compared to other algorithms from the literature?**
> >
> > I am not sure that you can consider the number of trials L and the sampling budget at each trial T at each stage as constants, especially if you set L1 as in Theorem 4.1 and in Corollary 4.3 (e.g., $L_1 \approx \frac{4\log(1+2/0.05)}{\log\frac{1}{4e \times 0.05}} \approx15$ which is at least a multiplicative factor of the min(sample complexity of the base algorithm,T)). But I understand that your algorithm is primarily a theoretical contribution.
> >
> > I am still not entirely sold on the impact of this specific problem (see my replies above). However, the strengths of the paper listed above still hold, and, as such, I will keep the score as it is.

---

> > > ### Author Response · Authors · 2025-04-07
> > >
> > > Thank you very much for the discussion. We clarify that we do not claim our algorithm is currently the best choice for practical use; rather, it represents the first attempt to address a previously overlooked problem in theoretical BAI research. While acknowledging the imperfections of our algorithm, we argue that these should not affect evaluating the importance of the problem itself. Our goal was not optimality in sample complexity but rather demonstrating that our algorithm's complexity remains close to the base algorithm, differing only by logarithmic factors.
> > >
> > > We believe the direction of obtaining exponentially-decaying stopping time is worth exploring regardless of whether it ends up leading to practical algorithms or not from the beginning.

---

### Official Review · Reviewer_H9DE · 2025-03-03

**Overall Recommendation:** 3

**Summary:**

This paper studies the fixed-confidence best-arm identification problem for $1$-sub-Gaussian distributions. The authors remark that asymptotic guarantees on the expected sample complexity doesn’t prevent a large tail of the empirical stopping time. Worse, high probability guarantees of the sample complexity doesn’t prevent the algorithm from never stopping with a nonnegligible probability. The latter statement is highlighted both theoretically (Theorems 2.4 and 2.5) and empirically (Figure 1 and Appendix D). Therefore, the authors introduce the $(T_{\delta}, \kappa)$-exponential stopping tail property (Definition 2.8), which captures the fact that the tail of the stopping time is exponentially decreasing for large enough time. This condition is sufficient to prove both high probability bound and expected sample complexity bounds (Proposition 2.9). The authors introduce FC-DSH. This is a variant of the anytime algorithm DSH, which is an anytime version of the fixed-budget algorithm Sequential Halving. The algorithm proceeds in phases whose length doubles. Within each phase, SH runs for the current budget, i.e., uniform sampling on the set of active arms which is halved at the end of each of the $\log_2 K$ stages, and a gap-based stopping rule is evaluated at the end of the phase. The observations between phases $m$ and stages $l$ are not shared. FC-DSH is $\delta$-correct  (Theorem 3.1) and satisfies an exponential stopping tail property (Theorem 3.2). The authors propose the BrakeBooster meta-algorithm, which runs a base algorithm with a 2D doubling trick, both on the budget and on the number of independent runs for each base algorithms. The observations between phases $(r,c)$ and runs $l$ are not shared. Given a $\delta_0$-correct base algorithm that have high probability upper bound on its sample complexity, BrakeBooster is $\delta$-correct (Theorem 4.1) and satisfies an exponential stopping tail property (Theorem 4.2).

**## update after rebuttal**
The discussions between the authors and the different reviewers provided a more detailed and nuanced perspective on the research question tackled in this paper. Including those insightful discussions will improve the paper in its revised version. Two primary practical concerns remains for me: (1) the degradation of the empirical performance when using BrakeBooster and (2) the probability of non-termination might be negligible in modern BAI algorithms. However, I recognize the theoretical contributions of the paper that study more sophisticated guarantees on the stopping time. Personally, I would be excited to see more works on this direction, e.g., matching lower/upper bounds on the higher moments (i.e., variance) of the stopping time. Therefore, I decided to raise my score to weak accept.

**Claims And Evidence:**

**On Theorems 2.4 and 2.5.** As currently stated in the main, I would argue that Theorems 2.4 and 2.5 are misleading/false given what is proven in Appendix A. On the specific considered instance, the probability that the algorithms never stop is lower bounded by $\Omega(\delta^{118})$. Therefore, it is far from being an absolute constant bounded away from $0$ for all $\delta$: it goes fast to $0$ when $\delta \to 0$.

**On Theorem 2.5.** Algorithm 5 seems quite far from lil-KL-LUCB of Tanczos et al., (2017). Could the authors precisely describe what is meant by “adapted and simplified for sub-Gaussian distribution” ? In particular, why those differences are not introducing undesirable behavior compared to the original algorithm ?

**On Theorem 2.7 and Theorem 2.5.** Theorem 2.7 shows that LUCB1 has a polynomial tail guarantee. Given the proof of Theorem 2.5, I am not sure to understand why a similar result cannot be shown for LUCB1, even though it would seem to contradict Theorem 2.7. Could the authors discuss the differences between Algorithm 5 and LUCB1 ? They seem awfully close. Is it solely a difference of the bonuses ? If so, it doesn’t alter the argument in the proof of Theorem 2.5 since it only controls the bad event that the initial draw of the best arm is not a good one, then argue that it will never be sample again. The same phenomenon hold for LUCB1 as it pulls both the empirical best arm and a distinct arm with the largest UCB.

**Targeted base algorithms for BrakeBooster.** BrakeBooster is tailored to improve the guarantees of algorithms that are $\delta$-correct and achieve high probability sample complexity guarantees. However, the authors themselves argue that “the high probability sample complexity [are] weak and rather unnatural”. Therefore, it seems unclear what is the benefit of constructing a meta-algorithm to “boost” the guarantees of algorithms with weak guarantee that suffers from large tail of stopping time. It would have been great if BrakeBooster was adaptive to improved theoretical guarantees of base algorithms. For example, when given a base algorithm with asymptotic guarantees, it could (1) show exponential stopping tail or (2) obtain non-asymptotic upper bound on the expected sample complexity. In other words, designing a meta-algorithm to improve the “best” known algorithms seems more promising than a meta-algorithm that improves algorithms with “poor” guarantees or performance.

**Discarding samples in BrakeBooster.** Crucially, BrakeBooster doesn’t share the observations across the different runs of the base algorithms. While this independence is key to derive theoretical guarantees, it seems wasteful in terms of sample complexity. This phenomenon is most likely blown out of proportion due to the large number of independent runs before stopping. Therefore, it seems legitimate to conjecture that the “lighter” tail of the empirical stopping time comes at a cost of a significant increase of the average empirical stopping time. If this conjecture is true, it would be a significant limitation of the usefulness of BrakeBooster. An empirical study of BrakeBooster seems necessary to understand the impact of this contribution.

**On FC-DSH.** While shown to be $\delta$-correct and having exponential stopping tail, an empirical study of FC-DSH seems necessary to understand whether it is a practical algorithm that performs well compared to existing algorithms.

**Essential References Not Discussed:**

To the best of my knowledge, there is no essential reference that is missing.

**Experimental Designs Or Analyses:**

**Empirical results for FC-DSH.** At the moment, there is no empirical evaluation of the performance of FC-DSH. In particular, it would be interesting to understand what is the empirical impact for FC-DSH of keeping the observations between each phase $m$ or/and each stage $l$. Given the rich literature in terms of sampling rules for BAI, it would be also relevant to understand what is the impact of using uniform sampling rule within each phase/stage by comparing FC-DSH to more adaptive sampling rules. It would allow comparing the proven exponential tail bound of FC-DSH with the empirical tail behavior of other algorithms, hence allowing to conjecture which other sampling rules might enjoy this property.

**Empirical results for BrakeBooster.** BrakeBooster is specifically designed for algorithms such as SE, i.e. $\delta$-correct and high probability upper bound on the sample complexity. Figure 1 and Appendix D show the limitation of SE. Those limitations are supposed to be solved by using BrakeBooster on top of SE. Therefore, it seems rather natural to empirically confirm the usefulness of BrakeBooster when applied to SE, especially since the authors state Corollary 4.3 for SE explicitly. Without empirical evidence of the benefits of using BrakeBooster, it seems rather unclear that BrakeBooster is actually helpful in practice.

**Appendix D and Figure 1.** To the best of my reading, the value of the $\delta$ parameter used in the experiments is missing. What is this value ? It is difficult to understand how “bad” are the “stopping failure” without putting them into perspective with the confidence level that is being targeted.

**Suggestions on the current setup.**
- It would be interesting to compare the targeted confidence $\delta$ with the empirical proportions of runs for which there is a “stopping failure”. Inherently, it should be smaller, since those algorithms are $\delta$-correct. Is it of the same order of magnitude or several order of magnitudes lower ?
- It would be interesting to add a line in the plots to give a proxy for the lower bound, e.g., $H_1 \log(1/\delta)$.

**Methods And Evaluation Criteria:**

See “Experimental Designs Or Analyses” section for details on the empirical evaluation.

**Other Comments Or Suggestions:**

**Theorem 3.2.** In the main, it would be better to write the explicit statement proved in Appendix B.2. This gives a better understanding of the actual dependency in $K$, $H_2$ and multiplicative constants.

**Theorem 4.2.** In the main, it would be better to write the explicit statement proved in Appendix C.1. This gives a better understanding of the actual dependency in $\delta$, $\delta_0$ and $T^\star_{\delta_0}(\mathcal A)$. For example, the term $\log(1/\delta_0)$ should be left in the upper bound instead of being swallowed by the $O(\log (T))$ notation. When $\delta_0\to 0$, it allows seeing that the $\log(1/\delta_0)$ term “cancels out” the dependency in $\delta_0$ from $T^\star_{\delta_0}(\mathcal A)$, which is likely also in  $\log(1/\delta_0)$.

- Appendix A uses the notation $\perp$ to denote the probability. To be consistent with the rest of the paper, it would be better to use $\mathbb P$.

- Lines 342-345. The event $E_1$ and $E_2$ are introduced, but there are not used.

- Lines 268-270. “The complexity of best-arm identification problems is often characterized by an instance-dependent quantity $H_2$”. I would argue that this sentence is slightly incorrect. $H_2$ is an instance-dependent quantity that appears in the analysis of fixed-budget BAI. However, for fixed-confidence BAI, the instance-dependent quantity $H_1$ is closer to the true characteristic time $T^\star$ for the asymptotic regime, i.e., $H_1 \le T^\star \le 2 H_1$. As highlighted by equation (2), the quantity $H_2$ seems to be sub-optimal by a multiplicator factor $\log_2 K$, which gets looser for larger instances.

**Other Strengths And Weaknesses:**

**Theorem 4.1.** The proof of $\delta$-correctness appears convoluted and doesn’t provide lots of insights despite taking almost one page of the main content. It would be better to allocate space to understand the theoretical novelty in the analysis of FC-DSH or BrakeBooster.

**Seemingly loose upper bounds.** In the Appendices, the proofs seem to use loose upper bounding in order to swallow the second-order terms in the first-order term by worsening its dependency. This is obfuscated by the use of $O(\cdot)$ notation in the final results. It would be interesting to write a tighter analysis with a smaller first-order term, and only argue in the end that the second-order term can be “removed” with the $O(\cdot)$ notation.

**Questions For Authors:**

1. **Improved analysis of FC-DSH.** Could the authors discuss whether their analysis of FC-DSH can be adapted to account for the recent improved analysis of FB-DSH ? For example, Zhao et al. (2023) show an improved rate compared to the rate $H_2$ used in this paper. Moreover, Kone et al. (2024, Bandit pareto set identification: the fixed budget setting) show that it is possible to keep past observations instead of discarding them at the end of each phase. It would be especially interesting to allow for keeping the samples, since it is known to have a large impact on the empirical performance of DSH.

2. **Majority voting.** Is it clear that majority voting is the best aggregation strategy based on $L$ independent runs of a given algorithm ? Would it be possible to define an aggregation strategy that leverage additional information on the independent runs, such as the empirical stopping time, to reweight each vote ?

3. Could the authors highlight precisely what are the theoretical novelties in the analysis of FC-DSH or BrakeBooster ?

Several other questions have been asked in the previous sections.

**Relation To Broader Scientific Literature:**

To the best of my understanding, the authors discuss relevant literature adequately.

**Theoretical Claims:**

I checked the correctness of the theoretical claims. To the best of my understanding, there is no major issue.

**On Lemma B.1.** There seems to be a minor error. The statement in Line 872 should read as “for all $T \ge 2T_{\delta}$”. In the current proof, the author requires that $T_m > T_{\delta}$ in Line 886 in order to use the assumption from the Lemma. However, this condition is not true, yet it could be shown if $T \ge 2T_{\delta}$”.

---

> ### Author Rebuttal · Authors · 2025-04-01
>
> We thank the reviewer for recognizing the value of our proposed exponential stopping tail property and finding it helpful. Please see empirical results in rebuttal to 94Yb. The maximum characters limit our rebuttal content, will add more in discussion.
>
> **On Theorems 2.4 / 2.5**: We agree that the lower bound $\Omega(\delta^{118})$ is not an absolute constant. However, the exponent $\delta^{118}$ can be regarded as a constant in the sense that it is independent of $H_1$, $K$, and $T$. Since $\delta$ is a user-specified input, it is reasonable to expect this bound to reflect a small constant probability relative to $\delta$. We will clarify this in our final version.
>
> **On Theorem 2.7 / 2.5**: Algorithm 5, lil-KLUCB (Tanczos et al., 2017), and LUCB1 share similar sampling and stopping rules but differ in confidence bound construction. Algorithm 5 uses a simplified bound, $\log(t^2/\delta)$, aligning with Algorithm 4. In contrast, lil-KLUCB employs $\log(\log_2(t)/\delta)$ with a refined $\delta$, while LUCB1 uses $\log(t^4/\delta)$, achieving a polynomial tail guarantee. We chose the simpler bound for analysis, but both lil-KLUCB and Algorithm 5 can fail to stop with a small, non-negligible probability, unlike LUCB1. We believe LUCB1’s polynomial tail lower bound is provable, yet our algorithms and non-stopping results reveal overlooked issues, showing that LUCB1’s guarantee is not the strongest possible.
>
> **Base algorithms**: Our algorithm doesn’t need a sample complexity bound as input; it works with any $\delta$-correct algorithm and improves if the base algorithm does. Feel free to pair it with $\delta$-correct algorithms boasting asymptotic sample guarantees-likely have strong finite-time guarantees too, though they may be tough to analyze (eg, TS-TCI/EB-TCI). The meta-algorithm’s strength is enabling exponential stopping times by leveraging existing algorithms with weaker guarantees. These base algorithms balance sample complexity and adaptive sampling rounds differently: elimination algorithms need few adaptive decisions, suiting batch sampling or limited adaptivity, while fully adaptive settings favor TS-TCI. Thanks for suggesting adaptation to the best algorithm—it’s a promising direction to explore.
>
> **Discarding samples**: We recognize the practical downside of discarding samples. However, our meta-algorithm’s key contribution is demonstrating, for the first time in the literature, a strategy that transforms a base algorithm with a weak guarantee into one with a stronger guarantee. While avoiding speculation, we think it’s feasible to refine the approach to eliminate or greatly reduce sample discarding. For insight, see Minsker (2023), “U-statistics of growing order and sub-Gaussian mean estimators with sharp constants,” where the author eliminated sample abandonment in the median-of-means algorithm, improving performance bounds using U-statistics.
>
> **On Lemma B.1**: We believe the reviewer may have misread this section of our proof. From Lines 884–886, we apply Lemma B.1’s assumption that FC-DSH satisfies $\mathbb{P}(\tau \geq T_m) \leq \exp\left(-\frac{T_m}{cH_2\log_2(K)}\right)$. Then, from Lines 886–888, we rely on the property $T < T_m$, established in Line 880.
>
> **App D / Fig 1**: We set $\delta$ to 0.05. In our experience, empirical failure rates are typically several times lower than the chosen $\delta$, even with stringent stopping conditions (e.g., from the track-and-stop paper). This pattern isn’t unique to our work but is common across fixed-confidence studies.
>
> **Theorem 4.1**: Many algorithms lack an exponentially decaying stopping time tail due to hard elimination steps that may discard the optimal arm without recovery. FC-DSH Novelty: The innovation lies in its tail probability analysis, a complex task not required for FB-DSH’s fixed-budget guarantee. These proofs involve event divisions absent in prior work, unlike FB-DSH’s simpler analysis (Karnin et al. 2013, Zhao et al. 2023).
>
> **Q1**:This falls outside our scope, but to our knowledge, the accelerated rate from Zhao et al. (2023) is unlikely to apply in the fixed-confidence setting. Fixed-confidence requires decision correctness, necessitating hypothesis testing. For your second point: Analyzing DSH without discarding samples is feasible—it swaps a $\log\log K$ factor in sample complexity for $\log K$. This involves a union bound over $K$ arms, as in the successive reject algorithm, which avoids sample discarding. We discarded samples to streamline the analysis.
>
> **Q2**: If you’re referring to improving sample complexity, our aggregation approach is generally optimal, barring constant or logarithmic factors, as further improvement would contradict established lower bounds. Reducing the number of voters while maintaining the same effect is a potential avenue, though its feasibility remains uncertain to us. Still, your suggestion could trim logarithmic or constant factors, and we’re keen to explore it in future work.

---

> > ### Comment · Reviewer_H9DE · 2025-04-02
> >
> > I thank the authors for their thorough and detailed answers, as well as the additional experiments. At the time being, I am inclined to keep my negative score.
> >
> > For the sake of discussion, I detailed some follow-up comments.
> >
> > **Lemma B.1**. I understood the current proof of Lemma B.1 as outlined in your answer. The authors might have misread my comment. By definition of the assumption within Lemma B.1, the inequality only holds for all phase $m$ such that $T_m \ge T_{\delta}$. To the best of my understanding, the authors do not show that $T_m \ge T_{\delta}$. Therefore, they cannot use their assumption without an additional argument. Taking $T \ge 2 T_{\delta}$ and using that $T_{m} > T/2$ would imply that $T_m \ge T_{\delta}$. This allows to use the assumption of Lemma B.1 and conclude the proof with a slightly modified statement (e.g., multiplicative factor two).
> >
> > **Lower bounds in Theorems 2.4 and 2.5**. By taking $\delta = 0.05$ as done in your experiments, the constant would be of the order $\Omega (3. 10^{-154})$. Therefore, from an empirical perspective, it seems computationally challenging to test that “the probability of not stopping is positive”. Relative to $\delta$, the story is unchanged as it yields $\Omega (\delta^{117})$.
> >
> > **Comments on additional experiments**. I completely agree with the three points raised in the Rebuttal Comment by Reviewer 94Yb.
> > - An empirical comparison with FC-DSH that doesn’t reuse the previous samples would be valuable to observe the impact of dropping observations. While the modification that reuses the sample perform well empirically, it lacks theoretical guarantees for now. Could the authors detail what are the technical challenges to study this improved algorithm ?
> > - Figure 3(b) on SE seems to corroborate the intuition that BrakeBooster might be wasteful in terms of samples. For an easy instance, “terminat[ing] within 350K rounds” and exhibiting a larger bulk of stopping times seems to be a mild evidence of empirical success. Since LUCB1 perform better than SE, it would be interesting to see empirically how BrakeBooster + LUCB1 perform. This would be insightful to see what is the cost of BrakeBooster for the sample complexity of a “relatively good” algorithm.
> >
> > **Practical significance of heavy-tail distributions in modern BAI algorithms**.
> > I tend to agree with Reviewer YUrK. Empirically, it would be great to exhibit a recent or widely-used BAI algorithm having heavy-tailed stopping time. While being introduced in a seminal paper, SE is not considered as a competitive BAI algorithm. To the best of my knowledge, most recent papers do not even include it in their experimental results due to its known empirical/theoretical shortcomings.

---

> > > ### Author Response · Authors · 2025-04-07
> > >
> > > Thank you very much for the discussions:
> > >
> > > - **Lemma B.1.**: We acknowledge the issue pointed out and will correct it following your suggestion. Thank you for bringing this to our attention.
> > >
> > > - **Lower bounds**: First of all, the message we wanted to convey with that theorem is that the probability of not stopping after $t$ does not decay as a function of time step $t$. That is, as $t \rightarrow \infty$, the probability of not stopping $\geq \text{Const} \neq 0$. In practice, $\delta$ is never too small, so we believe it is fair to consider $\delta$ as a constant. Moreover, the large exponent arises due to looseness in the analysis, reflecting a trade-off between achieving absolute tightness and avoiding overly complicated derivations. We believe our plot demonstrates clearly that the "no-stopping" case occurs with a noticeably higher probability in practice compared to the theoretical analysis. We conducted an additional experiment to show the percentage of failed (non-stopping) trials as a function of the confidence parameter. Please see version 2 in the same link at https://zenodo.org/records/15164857. Specifically, we considered a problem instance with 4 arms having mean rewards {1.0, 0.6, 0.6, 0.6}, and varied $\delta$ across the range $[10^{-5},\ldots,10^{-1}]$. We ran over 100K independent trials. As shown in Figure 4 (in the new anonymous link), the failure rate appears to scale approximately log-linearly with $\delta$. Notably, even when the confidence level is set as low as $\delta = 10^{-5}$, the SE algorithm still exhibits a non-stopping rate of 2.7\%. While our presented lower bound of $\Omega (\delta^{118})$ may appear loose, it serves its purpose of demonstrating that SE fails to stop at all with non-negligible probability.
> > >     Frankly, we are confused why the numerical evaluation of the equation appearing in theory is can be problematic -- this is extremely common in theory work and particularly not an issue in our context in our opinion. We repeatedly emphasize that the main contribution is theory, and ICML is a venue that values theory work, too.
> > >
> > > - **Practical significance**:
> > >     We disagree with the reviewer that SE is not considered as a competitive BAI algorithm. The answer depends on whether or not fully-adaptive decision-making is possible.
> > >     For example, a very recent study by Jin et al. (NeurIPS 2024) compared algorithms including Successive Elimination and highlighted practical limitations of fully sequential algorithms like Track-and-Stop, despite their optimal theoretical properties:
> > >
> > >     "The well-known Track-and-Stop algorithm solves the BAI problem with asymptotically optimal sample complexity. However, it is a fully sequential algorithm, which is hard to be implemented in parallel. The learner in such an algorithm receives immediate feedback for each arm pull, and adjusts the strategy for the next arm selection based on the previous observations. Unfortunately, this sequential approach may not be feasible in many real-world applications. For instance, in medical trials, there is typically a waiting time before the efficacy of drugs becomes observable, making it impossible to conduct all tests sequentially...."
> > >
> > >     - Optimal Batched Best Arm Identification, Jin et al, NeurIPS 2024

---

### Official Review · Reviewer_94Yb · 2025-03-11

**Overall Recommendation:** 3

**Summary:**

This paper considers the distribution of the stopping time in the fixed confidence BAI problem. It discovers that while most existing algorithms only have stopping time bounds in expectation or in high probability, which fail to achieve exponential decreasing rate (in time step $T$) for the misidentification probability.

To address this, the authors propose FC-DSH, an algorithm that guarantees an exponential-tailed stopping time. Additionally, a meta algorithm BrakeBooster is introduced, which can transform any fixed confidence BAI algorithm into one with an exponentially decaying stopping time.

**Claims And Evidence:**

Yes, the authors provide theoretical proofs in section 2 to indicate the some current algorithms cannot stop with a constant probability. The proposed FC-DSH and the meta algorithm BrakeBooster are introduced in section 3 and 4 respectively, accompanied by theoretical guarantees.

**Essential References Not Discussed:**

The references look good to me.

**Ethical Review Concerns:**

None.

**Experimental Designs Or Analyses:**

This paper focuses on the theoretical side.
Only one experiment about successive elimination is provided. No empirical experiments are provided to illustrate the proposed algorithms.

**Methods And Evaluation Criteria:**

The paper is mainly on the theoretical side of the existing BAI algorithms.

The proposed methods make sense from the theoretical standpoint. It makes use of and develops the doubling trick in the bandits literature to get the exponential stopping tail.

While the authors emphasize the theoretical contributions, the empirical performance of the proposed algorithms is not presented. Although the use of doubling trick can be beneficial in terms of theoretical analysis, the empirical performance of the algorithms with doubling trick usually have poor performance. It is expected that the authors can provide more discussions regarding this issue.

**Other Comments Or Suggestions:**

As said in previous sections, it would be great if the authors can provide some empirical results on the performance, even if the results may not be as good as those algorithm without exponentially decaying error probability.

In particular, empirical comparisons between the proposed works and the existing works listed in Table 1 is suggested.

**Other Strengths And Weaknesses:**

**Strengths**

This paper further develops the doubling trick to a “two-dimensional” case and proposes the meta algorithm BrakeBooster. The choices of the hyper parameters are also well explained.

**Weaknesses**:

The proposed FC-DSH shares a similar design as the algorithm in Zhao et al. (2023) and the only modification is only the stopping rule.

**Questions For Authors:**

None.

**Relation To Broader Scientific Literature:**

This paper lies in the field of Bandits Algorithms, in particular, it is related to Best Arm Identification with fixed confidence in Bandits. It identifies the flaws in previous algorithms and propose two algorithms to fix the problem.

**Theoretical Claims:**

I skimmed through the analysis and the proofs look reasonable to me.

---

> ### Author Rebuttal · Authors · 2025-04-01
>
> We thank the reviewer for understanding our theoretical contribution and your interest in additional empirical studies. We address your comments below.
>
> Regarding the reviewer's comment that "FC-DSH is a lot like the algorithm in Zhao et al. (2023), with just a change to the stopping rule," we would like to clarify that, while FC-DSH shares some design similarities, its novelty lies in the analysis. Since bounding the tail probability is, in general, a nontrivial task, we had to prove statements that were not necessary when developing a fixed budget guarantee for FB-DSH. For example, we had to bound the probability that DSH fails to satisfy the stopping condition even if the best arm is chosen as the estimated best arm (i.e., $\mathbb{P}(\exists i\neq 1: L_1^{(m)} \le U_i^{(m)}, J_m = 1)$). To bound this, we need to bound the probability of a suboptimal arm $i$ reaching the expected-to-reach stage $\ell_i^*$ (defined in Lemma 4), i.e., $\mathbb{P}(L_1^{(m)} \le U_i^{(m)}, \ell_i \ge \ell_i^*, J_m = 1)$ and $\mathbb{P}(\ell_i < \ell_i^*)$ presented in Lemma 4 and Lemma 5, respectively. Proofs for these require a careful division of the events that is not found in prior work, to our knowledge. In contrast, the guarantee of FB-DSH requires analyzing just the probability of the optimal arm failing to reach the final stage $\mathbb{P}(J_m \ne 1)$, which can be done by our Lemma 6 or by existing proofs from Karnin et al. (2013) or Zhao et al. (2023).
>
> **Additional empirical studies**
>
> Anonymous link for empirical results: https://zenodo.org/records/15117826
>
> We sincerely thank the reviewer for your interest in additional empirical studies. In response, we have conducted and will include the following experiments in the final version of the paper: (1) a study demonstrating that our FC-DSH algorithm (Algorithm 1) consistently terminates, exhibits light-tailed stopping-time behavior, and performs comparably to two widely used fixed-confidence best-arm identification (FC-BAI) algorithms - LUCB1 (Kalyanakrishnan et al., 2012) and TS-TCI (Jourdan et al., 2022) - as presented in Table 1; and (2) an empirical validation of the proposed meta-algorithm, BrakeBooster, confirming its ability to mitigate the stopping-time issues encountered by the Successive Elimination (SE) algorithm.
>
> We implement a similar experimental setup as in our paper, but introduce a variation with 4 arms having of mean rewards of {1.0, 0.6, 0.6, 0.6}. We set the confidence level to $\delta = 0.05$ across all experiments. We conduct 1,000 trials and record the stopping times for each. Trials that did not terminate within $1$ million steps were forcefully stopped. Although this setup features a larger reward gap - making it an easier problem instance - the Successive Elimination (SE) algorithm still fails to stop in a significant number of trials (92 out of 1000 trials), as shown in Figure 1a.
>
> Figure 1b shows that our FC-DSH algorithm consistently stops, alongside two other fixed-confidence best-arm identification (FC-BAI) algorithms: LUCB1 and TS-TCI. As the reviewer noted, the use of the doubling trick may degrade practice performance. To mitigate this issue, we modify our FC-DSH to reuse all samples collected in previous phases and stages. The modified FC-DSH outperforms LUCB1 and performs comparably to TS-TCI as shown in Figure 1b and Figure 2 (Left). This modified version highlights the practical potential of FC-DSH, and we believe that developing a theoretical understanding of sample reuse would be an interesting direction for future work. To highlight the differences in tail behavior, Figure 2 (Right) presents the empirical CDF of the stopping times. As expected, LUCB1 exhibits a slightly heavier tail - consistent with its theoretical polynomial tail guarantee discussed in the paper.
>
> Moreover, we apply BrakeBooster on top of the Successive Elimination (SE) algorithm and, as expected, observe that all trial runs successfully terminate within 350K rounds, as illustrated in Figure 3b - a clear contrast to the behavior of SE alone. While BrakeBooster is not yet optimized for efficiency, it serves as an important first step toward developing a general-purpose meta-algorithm. Its primary value lies in demonstrating the feasibility of such an approach, which has the potential to extend exponential tail stopping-time guarantees to a broad class of base algorithms.
>
> We’d be more than happy to address any other questions and concerns.

---

> > ### Comment · Reviewer_94Yb · 2025-04-02
> >
> > Thank the authors for the detailed reply! I do not have any question for the theoretical part.
> > It is great to see the empirical results of the algorithm. I have a few additional points that I hope the authors can clarify:
> > 1. In terms of Figures 1(b) and 2, the empirical performance of a variant of FC-DSH is good **by reusing the previous samples**, outperforming LUCB1. However, the algorithm reuses the previous samples, so it **does not** enjoy the theoretical guarantees, including the $\delta$-PAC property and exponentially decaying rate of the misidentification probability. As the authors indicated, this method requires further investigation. Since this paper targets at devising algorithms which enjoy finite stopping time guarantees, this variant of FC-DSH is irrelevant to the goal and the experiments do not support the theoretical findings. Therefore, it would be convincing if the authors can implement the proposed algorithm following its **exact theoretical design without reusing the previous samples** (so that it enjoys the proposed theoretical guarantees). I believe most papers in the bandits community implement their algorithms follow the exact design, including LUCB1.
> > 2. Regarding Figure 2(b), while I acknowledge that the authors wish to show the CDF of (the variant of) FC-DSH is light-tailed and that of LUCB1 is polynomial-tailed, the comparison should be refined. Although the data is mean-centered, the **variance** can also influence the shape of the curve of the CDF.
> > 3. For Figure 3, can the authors please provide the $\alpha$-quantile (for $\alpha=0.1\times k,k\in[10]$)? It seems SE stops quite early most of the time, and its histogram concentrates around 0. Based on Figure 3, it indicates the proposed Brakebooster does mitigate the stopping time issue observed in previous algorithms (at least SE) as the theorems indicate. But it is obtained at the cost of (much) higher sample complexity, even under an easy instance with a moderate $\delta=0.05$. In addition, the stopping time performance of SE can be improved by using a smaller $\delta$, e.g., $0.01$ or $0.005$, which can be more practical and easy to be implemented.
> >
> > Given the empirical results, I am still concerned about the practical side of the proposed method and hope the authors can clarify.

---

> > > ### Author Response · Authors · 2025-04-07
> > >
> > > Thank you very much for the discussion.
> > >
> > > - **Q1**: Our choice of not reusing sample was purely for a cleaner presentation as our focus was theoretical. The change in the analysis is just to use a union bound over all the arms, which would introduce a factor of $K$ in place of a $\log_2(K)$ outside $\exp(-(\cdots))$, which replaced a $\log\log$ factor into a $\log$ factor. What matters is ensuring that the sample is above certain count, and that's it. Note that successive rejects (Audibert et al. 2010) reuses samples from its original form and enjoys guarantees in the same mechanism we described. For this reason, it is common to develop theory with no sample reuse and experiment with sample reuse; e.g., Jun and Nowak, 2016, Baharav and Tse, 2019; Zhao et al, 2023 etc.
> > >     - Jun and Nowak, "Anytime exploration for multiarmed bandits," ICML 2016.
> > >     - Baharav and Tse, "Ultra fast medoid identification," NeurIPS 2019.
> > >     - Zhao et al., "Revisiting simple regret," ICML 2023.
> > >
> > > - **Q2**:  We agree with the reviewer that variance can affect the shape of the CDF curve. To investigate this further, we conducted an additional experiment and included a new plot showing the tail probability $P(X > x)$. Please see version 2 in the same link at https://zenodo.org/records/15164857. We used the same problem instance with 4 arms having mean \{1.0, 0.6, 0.6, 0.6\} but increased the number of trials from 1K to 1 million. As shown in Figure 3a and Figure 3b, we are unable to clearly confirm whether LUCB1 exhibits a polynomial tail (if it does, it should ultimately show a linearly decaying trend). Note that the LUCB1's paper does not include any experimental studies to we don't know for sure how it behaves. There are two plausible interpretations. Firstly, LUCB1 may exhibit an exponential tail, which would mean that the current theoretical guarantee of LUCB1 is loose. It is an interesting research direction for tigher analysis. Secondly, LUCB1 might indeed have a polynomial tail but it may take a lot more number of simulations to verify. That said, our results show that LUCB1 is much worse than FC-DSH. Furthermore, as shown in Figure 3b, we are surprised to observe that TS-TCI exhibits interesting behavior. This result reinforces our belief that exponential tail guarantees are far from obvious, even for well-studied algorithms, and merit further attention. We welcome any additional suggestions for experimental validation to better understand and confirm the nature of the tail behavior in these algorithms. We will add these experiments to the final version.
> > >
> > > - **Q3**: We apologize for the mistake in our experimental results. In fact we accidentally ran, brake booster + SE ($\mathcal{A} = \{1,0.9, 0.9, 0.9\}$ ) and SE ($\mathcal{A} = \{1,0.6, 0.6, 0.6\}$ ) on different settings. Here are the refined results: Experiment 1 (Figure 1) $\mathcal{A} = \{1,0.6, 0.6, 0.6\}$ and Experiment 2 (Figure 2) $\mathcal{A} = \{1,0.9, 0.9, 0.9\}$: Considering the space for the rebuttals, we have plotted the CDF instead of giving the $\boldsymbol{\alpha}$-quantiles here.
> > >
> > >
> > > Finally, it seems the reviewer is mainly not satisfied with the practical perspective. We believe getting a rejection due to lack of practically-interesting algorithms while the contribution is theory would be harmful to the community. We believe a healthy research community will be formed when **raising issues from theoretical viewpoint** and **validating if it leads to practical algorithms** can be recognized as two separate contributions (and thus each can be a standalone paper). That said, we agree that it is important to keep an eye on practical aspects, and we will include experiments showing the limitations of the proposed algorithms in the final version for the benefit of the readers.

---

### Official Review · Reviewer_mZ4M · 2025-03-11

**Overall Recommendation:** 3

**Summary:**

This paper address the best arm identification problem under fixed confidence, emphasizing the importance of exponential-tailed stopping time guarantees. Unlike existing algorithms prone to heavy tails or indefinite stopping, the authors propose two novel theoretical results: (1) an algorithm with exponential-tailed stopping time superior to prior methods, and (2) a meta-algorithm transforming any high-probability stopping algorithm into one with exponential-tailed guarantees, highlighting room for improvement in current methods.

**Claims And Evidence:**

I do not identify the claims of the paper as problematic.

**Essential References Not Discussed:**

Some non-aymtotic upper bounds in the family of track-and-stop algorihtms needs to be compared. For example, theorem 2 of "Fast Pure Exploration via Frank-Wolfe, NeurIPS 2021".

**Experimental Designs Or Analyses:**

There is no experimental design for proposed Algorithm 1 and 2.

**Methods And Evaluation Criteria:**

The evaluation criteria is the comparison of the theoretical upperbound. I did identify any big problem in the theoretical criteria. However, It would be better to explicitly write down the theoretical comparison; see in Q1 of "Questions For Authors"

Update a typo; sorry for any inconvenience. "I did not identify any big problem  in the theoretical criteria"

**Other Comments Or Suggestions:**

See in "Other Strengths And Weaknesses" and "Questions For Authors"

**Other Strengths And Weaknesses:**

## Strengths:
The research problem addressed in this paper is significant and has broader implications in the existing literature. Specifically, the observation that successful elimination algorithms are not guaranteed to be $\delta$-correct has largely been overlooked by the community. This paper addresses this gap by proposing a general algorithm to resolve this issue.

## Weaknesses:
1. The paper lacks experimental validation for the proposed Algorithms 1 and 2, even on synthetic datasets. Compared to existing methods such as the track-and-stop and top-two algorithms, the proposed algorihtms may be dramatically less efficient in simluations, because the proposed algorithms require significantly more arm pulls to maintain tail-bound guarantees. Providing empirical justification for the effectiveness of these algorithms through experiments would significantly strengthen the paper. Please clarify if this understanding is incorrect.

2.  I am not satisfied with the information that Table 1 conveys. For example, the column indicating whether an algorithm has exponential-tailed behavior appears not meaningful if the algorithms are already guaranteed to be $\delta$-correct.

3. Similarly to 2, the track-and-stop and top two algorithms are asymptotically optimal, while the proposed FC-DSH and brakebookster are not. Hence, it is not a fair comparison to just give a tick mark in the column of “asymptotic expected sample complexity” without demonstrating the optimality.

4. The statement on line 131, "the value of $\liminf_{\delta \to 0} E[\tau]\/ \ln(1/\delta)$ will be independent of $B$ even if $B$ is very large," is not accurate. For any asymptotically optimal algorithm (such as track-and-stop), it must hold that:  $\liminf_{\delta \to 0} E[\tau] \/ \ln(1/\delta)=\limsup_{\delta \to 0} E[\tau] \/ \ln(1/\delta) = \text{a instance-depedent constant}$

**Questions For Authors:**

1. Could the authors explicitly state the high-probability sample complexity and asymptotic expected sample complexity for the Successive Elimination algorithm enhanced with Brakebooster? Specifically, it would be helpful to specify the "polylog" term mentioned in Proposition 2.9 and explicitly compare the high-probability sample complexities of algorithms with and without Brakebooster. Similarly, an explicit comparison between FC-DSH and DSH would be beneficial.

2. Regarding Algorithm 2, what are the considerations involved in selecting the parameter $T_1$? Could the authors justify why $T_1$ is introduced as a parameter rather than fixing it to $T_1 = 1$?

Overall, for the valueable research problem and interesting results introduced in this paper, I currently recommend "weak acceptance".  I remain open to increasing or decreasing the rating in the discussion stage.

**Relation To Broader Scientific Literature:**

The research problem is very interesting and has a broader imact in the literature. It has been ignored by the community that the family of successful elimination algorithms are not $\delta$-correct. This paper aims to fix this problem.

**Theoretical Claims:**

I did not check the correctness of any proofs

---

> ### Author Rebuttal · Authors · 2025-04-01
>
> We thank the reviewer for recognizing the significance and broader impact of our work in identifying an intriguing problem. We address your comments below. Please see empirical results in rebuttal to 94Yb.
>
> 1. Our goal was to highlight a surprisingly overlooked issue in the bandit literature and provide theoretical evidence demonstrating that an exponentially decaying tail bound is indeed achievable. Before us, no one even realized it was possible. We do not claim empirical contributions but offer an initial theoretical exploration, hoping to stimulate further discussion in this direction.
>
> 2. When comparing our method to track-and-stop or top-two algorithms, we acknowledge that it is less efficient in terms of sample complexity. As demonstrated in the paper, using BrakeBooster results in additional sample complexity, up to logarithmic factors. However, our contribution lies not in minimizing sample complexity but in providing an additional safeguard. By sacrificing a logarithmic factor in sample complexity, our method ensures that the algorithm does not run indefinitely.
>
> 3. Thank you for the reference "Fast Pure Exploration via Frank-Wolfe." We will incorporate it into our paper.
>
> 4. On weakness 2, we clarify that being $\delta$-correct does not necessarily imply an exponential tail. For instance, Successive Elimination is $\delta$-correct yet lacks an exponential tail. Thus, it remains meaningful to explore this property even when an algorithm is $\delta$-correct.
>
> 5. On weakness 3, we agree that simply placing a checkmark in the “asymptotic expected sample complexity” column without proving optimality is not a fair comparison. We will revise this to ensure our contribution is not misleading.
>
> 6. On weakness 4, we agree that the statement could be misleading and is somewhat irrelevant. We will remove it from our revision. Thank you for pointing this out.
>
> **Regarding your questions**
>
> On question 1: We would like to clarify that we do not claim BrakeBooster provides a guarantee of asymptotic expected sample complexity. On your comment mentioning "enhanced with BrakeBooster", we did not intend to imply an improvement in this aspect. Instead, our goal is to highlight that BrakeBooster addresses the often-overlooked issue of stopping tail behavior while aiming to preserve sample complexity as much as possible. For high-probability sample complexity, the Successive Elimination algorithm without BrakeBooster yields a sample complexity of $\mathcal{O}\left(\tau_{\text{SE}}:=\sum_{i=2}^{K} \frac{\ln\left(\frac{K}{\delta\Delta_i}\right)}{\Delta_i^2}\right)$. With BrakeBooster, this shifts to $\mathcal{O}(\tau_{\text{SE}}\log^2(\tau_{\text{SE}}))$. More broadly, regarding the polylog term in Proposition 2.9, as long as an algorithm conforms to the specific form outlined in Definition 2.8 for its exponential tail, we can always address it with the following general approach. Suppose an algorithm satisfies, for all $T \geq T_\delta$,
>
> \begin{align}
> \mathbb{P}\left( \tau \geq T \right) \leq \exp\left(-\frac{T}{\kappa \cdot \log^b(T)}\right)
> \end{align}
> where $b$ is any positive integer. By setting the right-hand side to be less than $\delta$, we obtain
> \begin{align}
> \frac{T}{\log^b(T)} > \kappa \log(1/\delta).
> \end{align}
> Determining a sufficient condition to ensure this inequality holds can be intricate, but it is always feasible to solve for $T$ using the following. Our objective is to establish a high-probability bound through contraposition. To do so, we explore a necessary condition for $T$, such as $T \leq c \log^b(T)$,
> $$ T \le c \log^b(T) $$
>
> $$ \leftrightarrow T^{1/b} \le c^{1/b} \log(T) $$
>
> $$ \leftrightarrow T^{1/b} \le b c^{1/b} \log(T^{1/b}) $$
>
> $$ \leftrightarrow T^{1/b} \le b c^{1/b} \log\left(\frac{T^{1/b}}{2 b c^{1/b}} 2 b c^{1/b}\right) $$
>
> $$ \rightarrow T^{1/b} \le b c^{1/b} \left(\frac{T^{1/b}}{2 b c^{1/b}} - 1 + \log(2 b c^{1/b})\right) \tag*{$\ln(T) \le T - 1$} $$
>
> $$ \leftrightarrow T^{1/b} \le 2 b c^{1/b} \left(\log(2 b c^{1/b}) - 1\right) $$
>
> $$ \leftrightarrow T \le c \left(2 b \left(\log(2 b c^{1/b}) - 1\right)\right)^b $$
>
> Thus $T> c(2b(\log(2bc^{1/b})-1))^b$ is a sufficient condition to say $T> c\log^b(T)$ and we therefore solve the polylog factors.
>
> In comparing FC-DSH and DSH, FC-DSH is a refined version of DSH, enhanced with a stopping condition. DSH, an anytime algorithm, can theoretically run indefinitely without such a limit.
>
> On question 2: Conceptually, setting $T_1 = 1$ throughout does not impact the core results. Our theorems demonstrate that the findings hold for all $T_1 \geq 1$. Including $T_1$ as a parameter provides flexibility in certain cases, as a meaningful minimum number of samples is necessary to run the algorithm effectively. Typically, we require the sample size to exceed the number of arms. On the other hand, if we happen to know the base algorithm's high probability stopping time, we can just set it as $T_1$ directly, as an efficient start of BrakeBooster.

---

> > ### Comment · Reviewer_mZ4M · 2025-04-03
> >
> > Thank you very much for the detailed response. I still have one thought as follows:
> >
> >
> > I did not realize that your definition of $\delta$-correct is different from the definition of $\delta$-PAC in Garivier and Kaufmann (2016). Hence, can we claim that Successive Elimination (SE) is not $\delta$-PAC (as its stopping time may not be finite with probability 1), while SE with your BrakeBoost can be $\delta$-PAC? If this claim is true, I would suggest the authors include it explicitly in the Introduction section of the paper.

---

> > > ### Author Response · Authors · 2025-04-03
> > >
> > > Yes this claim is correct. Thank you very much for making this contribution clear. We will include it explicitly in the introduction. Is there anything we can do to help you consider raising the score? We will do our best to address it.

---

### Decision · Program_Chairs · 2025-05-01

**Decision:**

Accept (poster)

**Comment:**

This paper studies the best arm identification problem in the fixed confidence setting. The main contribution is a meditation on the standard delta-PAC correct stopping time, focusing on the idea that it may have heavy tails, indicating that standard results may be misleading. The paper desires exponential tails for the stopping time and demonstrates a framework to achieve successful results for any input algorithm under standard conditions. During the review process, there was a vibrant discussion about the practicality of the approach that left some reviewers wanting. However, all agreed the paper was well executed otherwise and provided a clear technical contribution to the best arm identification literature.